# Nonlocal Fractional Quantum Field Theory and Converging Perturbation Series

Nikita A. Ignatyuk [1,2,*,†] ⓘ, Stanislav L. Ogarkov [1,*,†] ⓘ and Daniel V. Skliannyi [3,*,†] ⓘ

1   Moscow Institute of Physics and Technology (MIPT), Institutskiy Pereulok 9, 141701 Dolgoprudny, Russia
2   Skolkovo Institute of Science and Technology, Bolshoi Boulevard 30, bld. 1, 121205 Moscow, Russia
3   Weizmann Institute of Science, Herzl St 234, Rehovot 7610001, Israel
*   Correspondence: ignatyuk.na@phystech.edu (N.A.I.); ogarkov.sl@phystech.edu (S.L.O.); daniil.skliannyi@weizmann.ac.il (D.V.S.)
†   These authors contributed equally to this work.

**Abstract:** The main purpose of this paper is to derive a new perturbation theory (PT) that has converging series. Such series arise in the nonlocal scalar quantum field theory (QFT) with fractional power potential. We construct a PT for the generating functional (GF) of complete Green functions (including disconnected parts of functions) $\mathcal{Z}[j]$ as well as for the GF of connected Green functions $\mathcal{G}[j] = \ln \mathcal{Z}[j]$ in powers of coupling constant $g$. It has infrared (IR)-finite terms. We prove that the obtained series, which has the form of a grand canonical partition function (GCPF), is dominated by a convergent series—in other words, has majorant, which allows for expansion beyond the weak coupling $g$ limit. Vacuum energy density in second order in $g$ is calculated and researched for different types of Gaussian part $S_0[\phi]$ of the action $S[\phi]$. Further in the paper, using the polynomial expansion, the general calculable series for $\mathcal{G}[j]$ is derived. We provide, compare and research simplifications in cases of second-degree polynomial and hard-sphere gas (HSG) approximations. The developed formalism allows us to research the physical properties of the considered system across the entire range of coupling constant $g$, in particular, the vacuum energy density.

**Keywords:** quantum field theory (QFT); nonlocal QFT; scalar QFT; nonpolynomial QFT; Euclidean QFT; generating functional (GF); green functions; functional (path) integral; Gaussian measure; grand canonical partition function (GCPF); perturbation theory (PT); converging perturbation series

**MSC:** 28C20; 26E15; 46N50; 46N55; 81-08; 81T08; 81T10; 81T18; 81U20

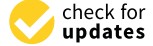



## 1. Introduction

Why do we need a nonlocal quantum field theory (QFT for short) in physics? Nonlocal QFT is a perturbatively, in each order of the perturbation theory (PT for short) with respect to the small coupling constant $g$, nonrenormalizable theory. The set of such theories is much wider than the set of perturbatively renormalizable ones. In particular, a nonlocal QFT can be formulated in a spacetime of arbitrary dimension $d$. Hence, this research is in some sense "orthogonal" to two-dimensional $d = 2$ conformal and integrable QFTs. Thus, nonlocal QFT was created precisely in order to consider the spacetime dimension $d$ as an arbitrary parameter [1–25].

Further, nonlocal QFT makes it possible to describe the high-energy physics of scalar particles (scalar nonlocal QFT) and in quantum electrodynamics (QED for short), as well as low-energy physics in quantum chromodynamics (QCD for short). For example, to describe the low-energy physics of light hadrons in QCD and quark confinement, the so-called Virton-Quark Model and the Quark Confinement Model of Hadrons are qualitatively successful [3,6–8]. Nowadays, research in the field of nonlocal quantum theory of gravity and nonlocal QFT in curved spacetime is also gaining popularity [14,16,23–25].

Another reason to research nonlocal QFT in Euclidean spacetime (the metric signature is all pluses) comes from the so-called grand canonical partition function/nonlocal, **nonpolynomial** Euclidean QFT scattering matrix ($\mathcal{S}$-matrix for short) duality [1–5,19–22,26–32]. This duality is well known, for example, in statistical physics and plasma physics, and its essence is as follows: $\mathcal{S}$-matrix of nonlocal nonpolynomial Euclidean QFT, originally formulated in terms of the functional (path) integral, can be rewritten in terms of the grand canonical partition function (GCPF for short) of some "gas with interaction". This duality is used in both directions. On the one hand, this makes it possible to apply the methods of statistical physics in QFT, in particular, to use the cluster expansion when calculating the $\mathcal{S}$-matrix. On the other hand, this makes it possible to apply QFT methods in statistical physics, in particular, to use the saddle-point method to calculate the functional (path) integral equal to the GCPF. Here, let us note that this duality is used in the research of sin-Gordon and sinh-Gordon models, more precisely, their nonlocal versions.

Finally, within the framework of nonlocal QFT, it is possible to research the strong coupling limit in interaction constant $g$ [3,19–22]. The method used to research the strong coupling limit in nonlocal QFT originated in the theory of polarons (in polaronics). This method is based on finding the minorant (lower estimate) and majorant (upper estimate) for the quantity of interest to us, as functions of the coupling constant $g$. The Jensen and Hölder inequalities are often used to find these estimates. If both estimates tend to each other in the limit $g \to +\infty$, then each of them coincides with the quantity of interest to us (in this limit). In fact, this is the squeeze theorem. For example, the vacuum energy $\mathcal{E}(g)$ was obtained in the strong coupling limit in [3,19,20].

Why do we need a nonlocal quantum field theory in mathematics? The rigorous definition of the functional integral in a nonlocal QFT, the proof of the definition correctness (the existence and uniqueness of such an object), as well as the calculation of the latter, is a complicated but interesting mathematical problem [33–47]. Let us note that even the calculation of such an object needs some definition. In what terms is the functional integral considered calculated? As quoted above, there is extensive mathematical literature devoted to the theory of integration in separable Hilbert and Banach spaces (HS and BS for short) with respect to Gaussian measures. At the same time, the "list of integrals" given in such literature is hardly sufficient for practical applications: results are usually given in cases where the integrand is a polynomial or again a Gaussian exponent, which corresponds to a Gaussian (free, without interaction) QFT. Our paper aims to narrow this gap slightly by adding another result to the "list of integrals".

Another complicated but interesting mathematical problem is the rigorous definition of the Dyson–Schwinger, Schwinger–Tomonaga and functional (nonperturbative, exact) renormalization group flow (FRG flow for short) equations in nonlocal QFT [42,44,45,48–53]. These equations are formulated in terms of variational derivatives, therefore, they are differential. Again, as quoted above, there is extensive mathematical literature devoted to differential equations in infinite-dimensional spaces (separable HS and BS). However, as in the case of integrals, the "list of solutions" given in such literature is hardly sufficient for practical applications. Obtaining solutions to such equations in terms of functional Taylor series is associated with solving infinite hierarchies (systems) of integro-differential equations for various families of Green functions. Such hierarchies are usually solved approximately. Let us note that FRG flow equations formulated as nonlocal QFT appear in many physical sciences: QFT itself, condensed matter physics, critical behaviour theory, stochastic theory of turbulence, etc.

Finally, in the framework of nonlocal QFT research in mathematics, let us note one possible generalization to the case of locally convex topological vector space (LCTVS for short) [54]. Such a seemingly mathematical generalization can lead to far-reaching consequences in physics. The topology of LCTVS is given by a family (set of arbitrary cardinality) of seminorms, generalizing separable HS and BS. In fact, instead of one scalar product or norm, a family of such arises. Such a family, hypothetically, can be associated with an infinite set of different ultraviolet (UV for short) scales, which makes it possible

to deselect one of them. It turns out, a theory in which there is no distinguished UV scale. Additionally, since the selected UV scale is the main problem of nonlocal QFT, such a hypothetical theory may already be a fundamental theory of nature. At the time of writing this paper, such a study seems to be the subject of the future.

There are still a few questions left to answer in the Introduction section. Why a scalar field? On the one hand, from the theory of groups and Lie algebras' point of view, the quantum theory of a scalar field has the smallest number of symmetries. For this reason, in such a theory there are no Ward–Takahashi identities that can simplify the $\mathcal{S}$-matrix calculation. In this sense, the scalar theory is the most complicated. On the other hand, to construct such a theory, it is the measure theory in infinite-dimensional spaces that is needed. In this sense, the scalar theory is the simplest, no additional symmetry relations need to be taken into account.

Why nonpolynomial self-interaction? In nonpolynomial theories that satisfy some general principles, the $\mathcal{S}$-matrix in terms of the GCPF is a convergent series in the interaction constant $g$. Let us note that for polynomial theories, for example, for the theories $\phi^4$ and $\phi^6$, such a series, being a PT series in the interaction constant $g$, is asymptotic. An asymptotic series arises as a consequence of the incorrectness of the mathematical transformations performed in the process of its derivation. In particular, the corollary of the monotone convergence theorem (MCT for short) for alternating series is violated. In addition, although the latter is only a sufficient condition, but not necessary and sufficient, nevertheless, the emergence of an asymptotic series means that the permutation of summation and functional integration does not really take place. In nonpolynomial theories of a certain class, this problem does not arise.

Why fractional power self-interaction $|\phi|^\alpha$, where $\alpha \in (1, 2)$? The research of such a theory seems natural for a number of reasons. The first reason is that such $\alpha$ is greater than the power in the source term and less than the power in the Gaussian theory. Therefore, in such a theory, it is possible to apply the corollary of the MCT for alternating series. The second (less obvious) reason is that such a theory is implicitly related to all the polynomial theories at once. This will be demonstrated in our paper. Finally, it can be expected that the method of calculating the $\mathcal{S}$-matrix in such a theory can be applied in the future to a wider class of theories. In this sense, in addition to the research of the theory itself, the result of the paper is a new method.

Finally, why $\mathcal{S}$-matrix and vacuum energy $\mathcal{E}$? The problem of QFT is considered solved if an explicit expression for the $\mathcal{S}$-matrix is obtained. Here, again, it is important to note that the very notion of an "explicit" expression needs to be defined. It is generally assumed that the corresponding functional integral must be calculated. The $\mathcal{S}$-matrix contains complete information about scattering: the variational derivatives of the $\mathcal{S}$-matrix determine the Green functions due to the interaction of particles. The Green functions make it possible to reconstruct the corresponding scattering cross-sections, which are experimentally observable quantities. For this reason, the research of the $\mathcal{S}$-matrix is the most important. Further, the simplest quantity contained in the $\mathcal{S}$-matrix is the vacuum energy $\mathcal{E}$ of the theory. It is this value that is most easily calculated. Let us note that the calculation of the functional integral leads to results that depend on the UV scale of the theory. In some cases, this dependency may be undesirably "strong". In this case, one has to invent some kind of subtraction scheme or change the original formulation of the theory. In this paper, we aim to calculate the functional integral in nonlocal QFT for given values of the theory parameters and to research how the results obtained depend on the corresponding parameters.

This paper has the following structure. In Section 2 we present a summary of the problem and obtained results. This section can be read independently of the rest of the paper. All the necessary definitions and notations are given within the section. Section 3 is devoted to physical background and motivation. Section 4 is devoted to:

1. an explanation of the Gaussian measure concept;
2. the derivation of PT series and proof of its convergence;

3. the exponentiation of the obtained series.

Section 5 is devoted to the calculation of PT series terms using polynomial approximation. Sections 6 and 7 are devoted to the calculation and research of vacuum energy density, namely:

1. the exact computation of first and second order in coupling constant $g$;
2. the computation with hard-sphere gas (HSG for short) approximation;
3. the expression for the second-degree polynomial approximation.

In the Section 8, we give a final discussion of all the results obtained in the paper. In the Section 9, we highlight further possible areas of research.

## 2. Summary of Problem and Obtained Results

In this paper, Euclidean quantum theory of scalar field $\phi$ in $\mathbb{R}^d$ with action $S[\phi]$:

$$S[\phi] = S_0[\phi] + S_I[\phi] = \frac{1}{2} \int_{\mathbb{R}^d} \int_{\mathbb{R}^d} dx \, dy \, L(x,y) \phi(x) \phi(y)$$
$$+ \int_{\mathbb{R}^d} dx \, g(x) |\phi(x)|^\alpha, \quad \alpha \in (1,2), \tag{1}$$

is considered. Here, $\phi : \mathbb{R}^d \to \mathbb{R}$—real-valued function (scalar field), $g : \mathbb{R}^d \to \mathbb{R}$—real-valued function (IR coupling constant), $j : \mathbb{R}^d \to \mathbb{R}$—real-valued function (source, appearing in the expression below), $L = G^{-1}$—inverse propagator, defined on the space of scalar fields $\Phi$. We do not specify this space in the present section, but in the following sections we will assume that $\Phi$ is a separable HS. **Convention**: hereafter we use the same letter for an operator and its integral kernel, for example, $L$ and $L(x,y)$, and we also use the same physical term for an operator and its integral kernel, for example, the term "propagator" for $G$ and $G(x,y)$. It is always clear from the context what exactly we are talking about. Additionally, in the paper we will assume that $L(x,y) = L(y,x)$, hence $G(x,y) = G(y,x)$. Finally, the propagator $G$ is a trace class operator and in all the final calculations, where possible, we will assume the translational invariance of the Gaussian theory, which implies $G(x,y) = G(x-y)$.

Complete (including disconnected parts) Green functions GF $\mathcal{Z}$ in terms of functional integral over primary (opposite to composite) field $\phi$ in such a theory:

$$\mathcal{Z}[j] = \int_\Phi \mathcal{D}[\phi] \, \exp\left( -\frac{1}{2} \int_{\mathbb{R}^d} \int_{\mathbb{R}^d} dx \, dy \, L(x,y) \phi(x) \phi(y) \right.$$
$$\left. - \int_{\mathbb{R}^d} dx \, g(x) |\phi(x)|^\alpha + \int_{\mathbb{R}^d} dx \, j(x) \phi(x) \right), \tag{2}$$

are derived in terms of convergent perturbation series in powers of coupling constant $g$, which is a function of $x$ in the general case:

$$\mathcal{S}_\varepsilon[\varphi] = \sum_{n=0}^\infty \frac{(-1)^n}{n!} \left\{ \prod_{a=1}^n \int_{\mathbb{R}^d} dx_a \, g(x_a) \int_{\mathbb{R}} d\phi_a |\phi_a|^\alpha \right\} \frac{1}{\sqrt{(2\pi)^n \det(G_n + \varepsilon 1_n)}}$$
$$\times \exp\left\{ -\frac{1}{2} \sum_{a,b=1}^n (G_n + \varepsilon 1_n)_{ab}^{-1} [\phi_a - \varphi(x_a)][\phi_b - \varphi(x_b)] \right\}. \tag{3}$$

In Expression (2) we use the notation "big differential" under the integral sign only formally: it is well known that there is no Lebesgue measure on infinite-dimensional separable HS and BS. However, together with the part of the integrand, the Gaussian exponent, this notation (up to a constant) means an integral over the Gaussian measure $\gamma_G$ with zero mean and covariance operator (propagator) $G$ in the corresponding space. This is how Expression (2) should be understood in the framework of a rigorous mathematical theory. Details are provided in the following sections.

In Expression (3) the following notations are introduced:

1.  $\mathcal{S}_\varepsilon[\varphi] = \mathcal{Z}[j]/\mathcal{Z}_0[j]$ is the interaction scattering matrix (hereafter—$\mathcal{S}_\varepsilon$-matrix) and $\mathcal{Z}_0[j]$ is the Gaussian GF;
2.  $G_n$ is $n \times n$ matrix with elements $(G_n)_{ab} = G(x_a, x_b)$ is the discretization of the propagator with values in the finite-dimensional space $\mathbb{R}^{n^2}$;
3.  $\varphi(x) = Gj(x) = \int_{\mathbb{R}^d} dy \, G(x,y)j(y)$ is the classical field ($\mathcal{S}_\varepsilon$-matrix argument) corresponding to the source $j$;
4.  $\varepsilon > 0$ is the parameter for defining integrands on null sets (in final results $\varepsilon \to +0$) and $1_n$ is $n \times n$ identity matrix with elements $\delta_{ab}$ (Kronecker delta).

Further in the paper we have proved the existence of the connected Green functions GF $\mathcal{G}_{I,\varepsilon}[\varphi] = \ln(\mathcal{Z}[j]/\mathcal{Z}_0[j])$ (index $I$ means "interaction part"). At the same time, it is convenient to consider this functional depending on $\varphi$, but not on $j$. Let us note that another definition of the connected Green functions GF is also introduced in the literature $\mathcal{G}[j] = \ln(\mathcal{Z}[j]/\mathcal{Z}_0[j=0])$ [44,48]. At the same time, such a functional is considered depending on $j$. **Convention**: hereafter we choose a normalization of functional integration such that the value $\mathcal{Z}_0[j=0] = 1$. This choice corresponds to the normalization of the Gaussian measure $\gamma_G$ in $\Phi$.

Further, PT series for $\mathcal{G}_{I,\varepsilon}[\varphi] = \sum\limits_{n=0}^{\infty} \mathcal{G}_{I,\varepsilon,n}[\varphi]$, and the nth term of the series has the following form:

$$
\begin{aligned}
\mathcal{G}_{I,\varepsilon,n}[\varphi] = \frac{(-1)^n}{n!} \sum_{\Gamma \in \mathbb{G}_{C,n}} \left\{ \prod_{a<b}^n \int_0^1 ds_{ab} \, \partial_{s_{ab}}^{\nu_{ab}(\Gamma)} \right\} \left\{ \prod_{a=1}^n \int_{\mathbb{R}^d} dx_a \, g(x_a) \int_{\mathbb{R}} d\phi_a |\phi_a|^\alpha \right\} \\
\times \frac{1}{\sqrt{(2\pi)^n \det(G_{n,\Gamma})}} \exp\left\{ -\frac{1}{2} \sum_{a,b=1}^n (G_{n,\Gamma})_{ab}^{-1} [\phi_a - \varphi(x_a)][\phi_b - \varphi(x_b)] \right\}.
\end{aligned}
\tag{4}
$$

In Expression (4) the following notations are introduced:

1.  $\mathbb{G}_{C,n}$ is the set of all connected undirected graphs with no loops and multiple edges on $n$ vertices;
2.  $\nu_{ab}(\Gamma)$ is an adjacency matrix of a graph $\Gamma$ (the differential operator $\partial_{s_{ab}}$ is raised to the power of this quantity);
3.  $(G_{n,\Gamma})_{ab} = s_{ab}\nu_{ab}(\Gamma)(G_n)_{ab} + (G(0) + \varepsilon)\delta_{ab}$, where $G(0)$ is the $G(x,x)$ in the translation invariant case;
4.  $s_{ab}$ are auxiliary variables, using for extracting connected contributions to $\mathcal{G}_{I,\varepsilon,n}[\varphi]$.

Further, we have calculated the first two orders in $g(x) = g\chi_Q(x)$, where $\chi_Q$ is the indicator function of $d$-dimensional cube centered at the origin with $\text{Vol}\, Q = V$, for the vacuum energy density $w_{vac} = \mathcal{E}/V$, namely (italic $\Gamma$ is the Gamma function, $_2F_1$ is the Gauss hypergeometric function and $\mathcal{E} = -\mathcal{G}_{I,\varepsilon}[0]$ is the vacuum energy):

$$
\begin{aligned}
w_{vac} = \frac{gG(0)^{\frac{\alpha}{2}} 2^{\frac{1+\alpha}{2}} \Gamma\left(\frac{\alpha+1}{2}\right)}{(2\pi)^{1/2}} - \frac{g^2 G(0)^\alpha 2^{\alpha-1} \Gamma\left(\frac{\alpha+1}{2}\right)^2}{\pi} \\
\times \int_Q dx \left\{ \left(1 - \frac{G(x)^2}{G(0)^2}\right)^{\alpha+\frac{1}{2}} {}_2F_1\left(\frac{\alpha+1}{2}, \frac{\alpha+1}{2}; \frac{1}{2}; \left(\frac{G(x)}{G(0)}\right)^2\right) - 1 \right\} + O(g^3),
\end{aligned}
\tag{5}
$$

and applied these formulas for the cases of virton propagator [3,8] (exponential form factor, which is a smooth UV-cutoff function in momentum space with UV parameter $\mu$) and Euclidean Klein–Gordon propagator with mass $m$.

For the Klein–Gordon propagator with mass $m$ and sharp UV-cutoff function in momentum space with UV parameter $\mu$ when the modes are frozen at $|k| > \mu$ and the

propagator in momentum representation $G(|k| > \mu) = 0$, we have obtained for the vacuum energy density $w_{vac}$ the following expression:

$$
w_{vac}\, m^{-d} = a_1 \frac{gG(0)^{\alpha/2}}{m^d} + a_2 \left( \frac{gG(0)^{\alpha/2}}{m^d} \right)^2 \left( \frac{m^{d-2}}{G(0)} \right)^2 + O(g^3),
$$

$$
a_1 = 2^{\frac{\alpha+1}{2}} \Gamma\left( \frac{\alpha+1}{2} \right), \quad a_2 = \frac{2^{\alpha-1}}{\pi} \Gamma\left( \frac{\alpha+1}{2} \right)^2 \begin{cases} \frac{1}{4}, & d = 2; \\ \frac{1}{8}, & d = 3. \end{cases}
\tag{6}
$$

Additionally, we have provided the following polynomial formula for $\mathcal{G}_{I,\varepsilon,n}[\varphi]$, derived by Legendre polynomial $P_n$ expansion (with an arbitrary coupling constant $g$):

$$
P_n(t) = 2^n \sum_{k=0}^{n} \binom{n}{k} \binom{\frac{n+k-1}{2}}{n} t^k, \quad t \in [-1,1],
\tag{7}
$$

$$
\begin{aligned}
\mathcal{G}_{I,\varepsilon,n}[\varphi] &= \frac{(-1)^n}{n!} \sum_{\Gamma \in \mathbb{G}_{C,n}} \left\{ \prod_{a<b}^{n} \int_0^1 ds_{ab}\, \partial_{s_{ab}}^{v_{ab}(\Gamma)} \right\} \left\{ \prod_{a=1}^{n} \int_{\mathbb{R}^d} dx_a\, g(x_a) \right\} e^{-\mathcal{Q}_{n,\Gamma}(x,\varphi)} \\
&\times \sum_{k=0}^{\infty} \frac{2^{n\alpha/2} G(0)^{2k+n\alpha/2}}{4^k k!} \frac{\left( \frac{n(\alpha+1)}{2} \right)_{2k}}{(1/2)_{2k}} \sum_{v_1+\dots+v_n=2k} \binom{2k}{v_1 \dots v_n} \chi_1^{v_1} \cdots \chi_n^{v_n} \sum_{q=0}^{\infty} \sum_{i=0}^{q} \left( 2q + \frac{1}{2} \right) \\
&\times \frac{2^{4q-ni+1} \Gamma\left( \frac{n(\alpha+1)}{2} \right)}{\Gamma\left( \frac{n}{2} + ni + k \right)} \sum_{p=0}^{q} \binom{2q}{2p} \binom{q+p-\frac{1}{2}}{2q} \frac{1}{2p+\alpha+1} \binom{2q}{2i} \binom{q+i-\frac{1}{2}}{2q} \\
&\times \sum_{\{l_{ab}\}} 2^{-\sum\limits_{a=1}^{n} \frac{l_{aa}}{2}} \frac{\prod\limits_{a=1}^{n} (2i+v_a)!}{\prod\limits_{a<b}^{n} (l_{ab}!) \prod\limits_{a=1}^{n} \left( \frac{l_{aa}}{2} \right)!} \prod_{a<b}^{n} \left( \frac{(G_{n,\Gamma})_{ab}}{G(0)} \right)^{l_{ab}}.
\end{aligned}
\tag{8}
$$

In Expression (8) the following notations are introduced:

1.　$\mathcal{Q}_{n,\Gamma}(x,\varphi) = \frac{1}{2} \sum\limits_{a,b=1}^{n} (G_{n,\Gamma})_{ab}^{-1} \varphi(x_a)\varphi(x_b)$ is the *n*-particle quantum entangler—the main difficulty in calculating any GF;

2.　$\chi_a = \sum\limits_{b=1}^{n} (G_{n,\Gamma})_{ab}^{-1} \varphi(x_b)$ is the auxiliary vector quantity, introduced for the compactness of formulas;

3.　$(x)_k := x(x+1)(x+2)\dots(x+k-1)$, and $(x)_0 := 1$ is the rising factorial;

4.　$\binom{2k}{v_1 \dots v_n}$ is the multinomial coefficient, in particular, $\binom{q}{p}$ is the binomial coefficient;

5.　**Convention**: summation in the last line of Expression (8) is carried out over all $l_{ab} \geq 0$ satisfying the conditions $\sum\limits_{b=1}^{n} l_{ab} = 2i + v_a$ (where $l_{ab} = l_{ba}$) and $l_{aa}$ is even. This expression is non-zero for even $\sum\limits_{a=1}^{n} v_a$ and zero for odd.

This yields the simple approximation formula for $w_{vac}$ if one chooses the degree of approximation polynomial is equal to 2 and again $g(x) = g\chi_Q(x)$:

$$
\begin{aligned}
w_{vac} &\approx \frac{3(2\alpha+1)\Gamma\left( \frac{\alpha+1}{2} \right)}{\sqrt{\pi}(\alpha+1)(\alpha+3)} (2gG(0))^{\alpha/2} + \frac{15\alpha}{(\alpha+1)(\alpha+3)} \\
&\times \sum_{n=2}^{\infty} \frac{(-1)^{n-1} 2^{n\alpha/2}}{4n} \left[ gG(0)^{\alpha/2} \right]^n \frac{\Gamma(\alpha n)}{\Gamma(3n/2)} \left\{ \prod_{a=1}^{n-1} \int_Q dy_a \frac{G(y_a)}{G(0)} \right\} \frac{G\left( \sum\limits_{k=1}^{n-1} y_k \right)}{G(0)}.
\end{aligned}
\tag{9}
$$

Beyond that, we derived another approximation formula for $w_{vac}$, based on HSG approximation of the matrix $G_n$. This approximation is quite often used in statistical physics. Supposing for $G_n$ the ansatz:

$$(G_n)_{ab} \approx (\gamma(J_n)_{ab} + (1-\gamma)\delta_{ab})G(0)\theta(\delta - |x_a - x_b|), \tag{10}$$

with $n \times n$ matrix $J_n$ of ones, Heaviside step function $\theta$ and parameters $\delta$ and $\gamma$, we arrive at the following expression for $w_{vac}$:

$$
\begin{aligned}
w_{vac} = &- \sum_{n=1}^{\infty} \frac{(-g)^n G(0)^{n\alpha/2}}{n!} 2^{n\alpha/2} v^{n-1} \sum_{\Gamma \in \mathbb{G}_{C,n}} \left\{ \prod_{a<b}^{n} \int_0^1 ds_{ab}\, \partial_{s_{ab}}^{v_{ab}} \right\} \sum_{q=0}^{\infty} \sum_{i=0}^{q} \left( 2q + \frac{1}{2} \right) \\
&\times \frac{2^{4q-ni+1}\Gamma\left(\frac{n(\alpha+1)}{2}\right)}{\Gamma\left(\frac{n}{2}+ni\right)} \binom{2q}{2i} \binom{q+i-\frac{1}{2}}{2q} \sum_{p=0}^{q} \binom{2q}{2p} \binom{q+p-\frac{1}{2}}{2q} \frac{1}{2p+\alpha+1} \\
&\times \sum_{\{l_{ab}\}} 2^{-\sum_{a=1}^{n} \frac{l_{aa}}{2}} \frac{(2i)!^n}{\prod_{a<b}(l_{ab}!)\prod_{a=1}^{n}\left(\frac{l_{aa}}{2}\right)!} \prod_{a<b}(v_{ab}s_{ab}\gamma)^{l_{ab}},
\end{aligned}
\tag{11}
$$

where $v = \frac{\pi^{d/2}}{\Gamma(d/2+1)}\delta^d$ is the "volume of one hard-sphere particle" and parameters $\delta$ and $\gamma$ are determined by the equations:

$$G(0)\gamma v = \int_{\mathbb{R}^d} dx\, G(x), \quad G(0)^2 \gamma^2 v = \int_{\mathbb{R}^d} dx\, G(x)^2. \tag{12}$$

Let us make one remark about Equation (12). The HSG approximation is used in the statistical physics for potentials that decay rapidly at infinity in $x$. This is reflected in Equation (12) when integrating over $\mathbb{R}^d$. This usually also means that the product of a power function and the power of the propagator is again integrable. All the propagators considered in the paper satisfy the conditions of HSG approximation applicability.

The other reasonable equations for parameters $\delta$ and $\gamma$ are also possible, and these are chosen to make formulas of HSG approximation simpler for substituting the particular cases of propagators. All the obtained formulas were checked for the limiting case $\alpha = 2$ and applied to the virton propagator (exponential form factor, smooth UV-cutoff function) as well as to Euclidean Klein–Gordon propagator with mass $m$ and sharp UV-cutoff function. Finally, we present approximate plots of vacuum energy density $w_{vac}$ for all values of $g$ for both described approximations and provide corresponding approximate formulas. These formulas are valid for strong coupling $g$ as well as for weak, which is quite a rare case for quantum field models.

Using the Weierstrass M-test, we proved the convergence of such a series for $j = 0$ and we have obtained the majorizing series explicitly. As for $j \neq 0$, we found the majorant depending on the regulator $\varepsilon$. There is still an open question about finding a more accurate majorant whose dependence on epsilon is negligible (for the case of the non-zero source $j$). Therefore, we will remove $\varepsilon$ in the quantities calculated for $j = 0$, such as vacuum energy and its density. In addition, in the quantities depending on $j$ we will not remove $\varepsilon$, which will be reflected in their indices, such as $\mathcal{G}_{I,\varepsilon}$, etc.

Performed considerations and research will help in the understanding of interacting quantum fields' general properties. Moreover, they can potentially be applied for the strong coupling limit of nonlocal $\phi^4$ and $\phi^6$ theories, which will be the subject of further research and publication of the authors.

## 3. Physical Background and Motivation

### 3.1. Definitions and Notations

The starting point of our research is the complete (including disconnected parts) Green functions GF $\mathcal{Z}$ in terms of functional integral over primary field $\phi$. Traveling back, the explicit expression for $\mathcal{Z}$ is:

$$\mathcal{Z}[j] = \int_{\Phi} \mathcal{D}[\phi]\, e^{-\frac{1}{2} \int_{\mathbb{R}^d} \int_{\mathbb{R}^d} dx\, dy\, L(x,y)\phi(x)\phi(y) - \int_{\mathbb{R}^d} dx\, g(x)|\phi(x)|^{\alpha} + \int_{\mathbb{R}^d} dx\, j(x)\phi(x)}. \tag{13}$$

We will also call this functional the GCPF. Let us note that hereafter we follow the normalizing convention $\mathcal{Z}_0[j = 0] = 1$, which makes this definition of GF coherent with those via Gaussian measure in the next section.

The function (distribution) $L(x,y)$ is the integral kernel of the operator $L$, which defines the quadratic part of a nonlocal QFT (Gaussian theory) action. Nonlocal QFT can be considered as a regularization of the Euclidean Klein–Gordon theory, but it is also of independent interest. Let us note that the integral kernel $G(x,y)$ of the inverse operator $G = L^{-1}$ is the UV finite propagator of a nonlocal QFT in any space dimension $d$. $G(x,y)$ is also called the Green function of Gaussian theory. In all the final calculations, where possible, we will assume the translational invariance of the Gaussian theory, which implies $G(x,y) = G(x - y)$, but at the same time, we will consider coordinate dependent coupling constant $g$. We will often choose $g(x) = g\chi_Q(x)$, where $\chi_Q$ is the indicator function of $d$-dimensional cube centered at the origin with $\mathrm{Vol}\, Q = V$. We will also suppose that $G(x) \geq 0$ and $G(x) \leq G(0)$ for all $x$. The functional integration is understood in the sense of Gaussian measure with covariance operator (propagator) $G$. In this paper, we assume that $\hbar = c = 1$.

Generating functional $\mathcal{Z}$ is a regular functional, so it could be expanded in a functional Taylor series at $j = 0$ [44,48,52]:

$$\mathcal{Z}[j] = \sum_{n=0}^{\infty} \frac{1}{n!} \mathcal{D}^{(n)}_{1,\ldots,n} j_1 \ldots j_n \;,\quad \mathcal{D}^{(n)}_{1,\ldots,n} = \left.\frac{\delta^n \mathcal{Z}[j]}{\delta j_1 \ldots \delta j_n}\right|_{j=0}, \tag{14}$$

where the repeated index $i$ means integration over some spacetime variable $x_i$. This means the following:

$$\mathcal{D}^{(n)}_{1,\ldots,n} j_1 \ldots j_n := \int_{\mathbb{R}^d} \ldots \int_{\mathbb{R}^d} dx_1 \ldots dx_n\, \mathcal{D}^{(n)}(x_1,\ldots,x_n)\, j(x_1) \ldots j(x_n). \tag{15}$$

Coefficients $\mathcal{D}^{(n)}_{1,\ldots,n} = \mathcal{D}^{(n)}(x_1,\ldots,x_n)$ are called $n$-particle or $n$-point correlation functions of a functional $\mathcal{Z}$ since they represent $n$-point correlators of the theory:

$$\frac{1}{n!}\mathcal{D}^{(n)}(x_1,\ldots,x_n) = \langle \phi(x_1) \ldots \phi(x_n) \rangle$$
$$= \int_{\Phi} \mathcal{D}[\phi]\, \phi(x_1) \ldots \phi(x_n)\, e^{-\frac{1}{2} \int_{\mathbb{R}^d} \int_{\mathbb{R}^d} dx\, dy\, L(x,y)\phi(x)\phi(y) - \int_{\mathbb{R}^d} dx\, g(x)|\phi(x)|^{\alpha}}. \tag{16}$$

Correlators are the object of interest since they are the quantities which determine the cross-section of scattering of particles and therefore can be directly measured in particle physics as well as in statistical physics. In Gaussian theory one has a usual Gaussian integral, and we will denote the corresponding GF as $\mathcal{Z}_0$. In addition, recall the convention we follow: $\mathcal{Z}_0[j = 0] = 1$. In this case, for GF $\mathcal{Z}_0$ we have a simple expression:

$$\mathcal{Z}_0[j] = e^{\frac{1}{2} \int_{\mathbb{R}^d} \int_{\mathbb{R}^d} dx\, dy\, G(x,y)j(x)j(y)}. \tag{17}$$

It is also known that $\mathcal{Z}$ can be represented as an exponent of the other regular functional $\mathcal{G}$, which is called the connected Green functions GF:

$$\mathcal{Z}[j] = e^{\mathcal{G}[j]}, \tag{18}$$

and from statistical physics point of view $\mathcal{Z}$ is a GCPF, as well as $\mathcal{G}$, is a grand thermodynamic potential (up to a temperature factor).

The functional $\mathcal{G}$ can also be expanded in a functional Taylor series, and its coefficients are called connected $n$-particle or $n$-point Green functions:

$$\mathcal{G}[j] = \ln \mathcal{Z}[0] + \sum_{n=1}^{\infty} \frac{1}{n!} \mathcal{G}^{(n)}_{1,\dots,n} j_1 \dots j_n \ , \quad \mathcal{G}^{(n)}_{1,\dots,n} = \left. \frac{\delta^n \mathcal{G}[j]}{\delta j_1 \dots \delta j_n} \right|_{j=0}. \tag{19}$$

The terminology goes from Feynman diagrams and will be clarified in the following while considering the Meyer cluster expansion. Of course, all $\mathcal{D}^{(n)}(x_1,\dots,x_n)$ can be expressed via $\mathcal{G}^{(n)}(x_1,\dots,x_n)$ by expanding the exponent. Throughout this paper, we denote connected $n$-particle Green functions with the mathcal font $\mathcal{G}^{(n)}_{1,\dots,n}$ rather than the Green function of Gaussian theory $G(x-y)$.

In statistical physics, a finite limit $\mathcal{G}/V$ at $V \to \infty$ exists. Therefore, we can write $\mathcal{G}$ as follows:

$$\mathcal{G}[j] = V f[j], \tag{20}$$

where $f$ is a volume density of $\mathcal{G}$. For a homogeneous thermodynamic system this limit is equal to pressure divided by temperature. It can be shown that $\mathcal{G}[0] = \ln \mathcal{Z}[0] = -\mathcal{E}$, where $\mathcal{E}$ is a vacuum energy of the considering QFT. It is also convenient to define its volume density:

$$w_{vac} := \frac{\mathcal{E}}{V}. \tag{21}$$

The quantity $w_{vac}$ is a very useful quantity which in particular can tell whether the system has phase transition or not. This relation is literal.

From the Lehmann–Symanzik–Zimmermann reduction formula, one can obtain that the expression for the $\mathcal{S}$-matrix $n$-particle correlation functions in momentum representation reads (the repeated variable means integration):

$$\mathcal{S}^{(n)}_{1',\dots,n'} = \mathcal{D}^{(n)}_{1,\dots,n} \prod_{k=1}^{n} (G_{kk'})^{-1}, \tag{22}$$

where $G_{kk'}$ is the two-particle Green function of Gaussian theory in momentum representation. Of course, $G^{-1} = L$, but it is accepted to denote the corresponding operator as $G^{-1}$. For deriving the $\mathcal{S}$-matrix, that is exactly the functional which functional Taylor coefficients are $\mathcal{S}^{(n)}_{1',\dots,n'}$, one should simply change variables from sources to so-called "classical fields" ($\varphi$ instead of $\phi$), namely:

$$\varphi(x) = Gj(x) = \int_{\mathbb{R}^d} dy\, G(x,y)j(y), \tag{23}$$

so the $\mathcal{S}$-matrix satisfies the following relation (up to a certain point, it is not necessary to introduce the regulator $\varepsilon$, mentioned in the Section 2):

$$\mathcal{S}[\varphi] = \frac{1}{\mathcal{Z}[0]} \mathcal{Z}[G^{-1}\varphi], \tag{24}$$

as well as "normalized" GF $\mathcal{Z}_I$:

$$\mathcal{Z}_I[j] = \frac{\mathcal{Z}[j]}{\mathcal{Z}_0[j]}, \tag{25}$$

and normalized GF $\mathcal{G}_I$ of connected Green functions (this functional can be considered both depending on the source and depending on the classical field, which was also mentioned in the Section 2):

$$\mathcal{G}_I[j] = \ln \mathcal{Z}_I[j] = \mathcal{G}[j] - \mathcal{G}_0[j], \tag{26}$$

where $\mathcal{G}_0[j]$ is the Gaussian theory connected Green functions GF. Such a functional is equal to the quadratic:

$$\mathcal{G}_0[j] = \frac{1}{2} \int_{\mathbb{R}^d} \int_{\mathbb{R}^d} dx \, dy \, G(x,y)j(x)j(y). \tag{27}$$

Next, we are going to describe two approaches to the perturbative calculation of $n$-particle functions $\mathcal{D}^{(n)}$ and $\mathcal{G}^{(n)}$. Let us consider the expansions of GFs $\mathcal{Z}$ and $\mathcal{G}$ in series:

$$\mathcal{Z}[j] = \sum_{n=0}^{\infty} \mathcal{Z}_n[j], \quad \mathcal{G}[j] = \sum_{n=0}^{\infty} \mathcal{G}_n[j], \tag{28}$$

which could be power series in some parameters of the system such as coupling constant $g$. Then they induce the corresponding expansions of $n$-particle functions in powers of the same parameter:

$$\mathcal{D}^{(k)}_{1,\ldots,k} = \sum_{n=0}^{\infty} \mathcal{D}^{(k)}_{1,\ldots,k;\, n} \ , \quad \mathcal{G}^{(k)}_{1,\ldots,k} = \sum_{n=0}^{\infty} \mathcal{G}^{(k)}_{1,\ldots,k;\, n} \ , \tag{29}$$

as well as for the $\mathcal{S}$-matrix. In the next subsection we are going to introduce the conventional method for obtaining of such expansions.

### 3.2. Standard PT and Its Inapplicability

Usually, when considering QFT with GF $\mathcal{Z}$:

$$\mathcal{Z}[j] = \int_{\Phi} \mathcal{D}[\phi] \, e^{-\frac{1}{2} \int_{\mathbb{R}^d} \int_{\mathbb{R}^d} dx \, dy \, L(x,y)\phi(x)\phi(y) - \int_{\mathbb{R}^d} dx \, V(\phi(x)) + \int_{\mathbb{R}^d} dx \, j(x)\phi(x)}, \tag{30}$$

with analytic interaction potential $V(\phi)$, one can obtain PT series via transformation:

$$\mathcal{Z}[j] = \exp\left[-\int_{\mathbb{R}^d} dx \, V\left(\frac{\delta}{\delta j(x)}\right)\right] \int_{\Phi} \mathcal{D}[\phi] \, e^{-\frac{1}{2} \int_{\mathbb{R}^d} \int_{\mathbb{R}^d} dx \, dy \, L(x,y)\phi(x)\phi(y) + \int_{\mathbb{R}^d} dx \, j(x)\phi(x)}, \tag{31}$$

where the operator $\exp\left[-\int_{\mathbb{R}^d} dx \, V\left(\frac{\delta}{\delta j(x)}\right)\right]$ is understood in the sense of power series in operator $\frac{\delta}{\delta j}$. Consequently, one has to demand the analyticity of the potential $V(\phi)$ in $\phi$ for this method to work. Let us note that it is this operator that leads to asymptotic series in QFT. Then, after introducing classical field $\varphi$, the final formula reads:

$$\begin{aligned}
\mathcal{Z}[j] &= \exp\left\{\frac{1}{2} \int_{\mathbb{R}^d} \int_{\mathbb{R}^d} dx \, dy \, G(x,y)j(x)j(y)\right\} \\
&= \exp\left\{\frac{1}{2} \int_{\mathbb{R}^d} \int_{\mathbb{R}^d} dx \, dy \, G(x,y)\frac{\delta}{\delta\varphi(x)}\frac{\delta}{\delta\varphi(y)}\right\} \exp\left\{-\int_{\mathbb{R}^d} dx \, V(\varphi(x))\right\}\Bigg|_{\varphi = Gj}.
\end{aligned} \tag{32}$$

The operator $\exp\left[-\int_{\mathbb{R}^d} dx \, V\left(\frac{\delta}{\delta j(x)}\right)\right]$ should be properly defined since it contains multiple variational derivatives in the same point, but in the final answer all the regularizations of this expression could be removed. Then, expanding both exponents, one can obtain a series in coupling constant $g$. As already mentioned above, these series are usually asymptotic. As one can see, this way relies significantly on the analyticity of the potential $V(\phi)$. It is invalid for the case of fractional power self-interaction potential.

### 3.3. Motivation of Study

At first glance, it can appear to be nonphysical to consider models with interaction potential of the form $V(\phi) = g(x)|\phi|^\alpha$ (explicit dependence on $x$ through $g$ is not indicated in the potential). Though, there are several reasons to study it.

Firstly, it is a kind of exactly solvable model in QFT, where one can produce converging series in $g$. So it is an object of common interest because it is useful to learn the behavior of different QFT systems to:

1. understand what kind of properties they can own and, consequently, what one can look up in other models;
2. develop new methods and approaches, which can be subsequently extended and applied to other systems to gain new results;
3. since we have the converging series for such a theory, we can hope to construct their analytic continuations to points $\alpha > 2$, in particular in the vicinity of $\alpha = 4$, which is an object of common interest.

Secondly, there are reasonable expectations, that in theory with the potential $V(\phi) = g(x)|\phi|^\alpha$ it is possible to complete explicit transition in obtained formulas from weak to strong coupling regimes. Moreover, there is the **conjecture** on the following duality between scalar field theories:

$$L \leftrightarrow G, \quad g(x) \leftrightarrow \frac{1}{g(x)}, \quad |\phi(x)|^\alpha \leftrightarrow |\phi(x)|^{\frac{\alpha}{\alpha-1}}. \tag{33}$$

This duality can be very useful for the research of nonlocal $\phi^4$ theory in a strong coupling regime as well as a nonperturbative description of phase transitions.

This duality can be obtained with the use of the Parseval–Plancherel identity for the functional integral with the following substitution of the functional Fourier transform asymptotic, which turns out to be valid in any $d$, since $d$ is the dimension of the "index" of vector while the argument of Fourier operator is the vector itself. Though, for careful checking of this duality one should obtain PT series of asymptotic expansions of $\phi^4$ theory from the strong coupling regime of $|\phi|^{\frac{4}{3}}$, which demands the general studying of $|\phi|^\alpha$ theories for $\alpha \in (1, 2)$. Describing, checking and using of this duality is a subject of current studies of the authors, and will be the topic of the nearest publications.

On the one hand, it should be noted that the first of three substitutions in Expression (33) poses a danger: if the original Gaussian measure $\gamma_G$ was given by a trace class covariance operator $G$, the dual Gaussian measure $\gamma_L$ is determined by an operator $L$ with an infinite trace value. Therefore, the integral over $\gamma_L$ is ill-defined. This is consistent with the conventional point of view that $\phi^4$ is quantum trivial in $d \geq 4$.

On the other hand, from the Gaussian measure in separable HS point of view, there are no evident reasons for GF to be in any sense "trivial" for any $d$ since $x$ is the index of a vector and the field $\phi$ is a vector in the corresponding HS $\Phi$. The integration theory in $\Phi$ is insensitive to the dimension of the index and is constructed uniformly for any $\Phi$. For the exponential of a polynomial, the integral over a "good" Gaussian measure is defined in rigorous mathematical sense [46,47]. For this reason, quantum triviality may turn out to be a consequence of the incorrectness of mathematical transformations in the calculation.

Be that as it may, the authors would like to find rigorous mathematical proof of the quantum triviality or to draw the conclusion that this triviality is a PT artifact only. Moreover, the present paper is the first step to ascertain the truth in this complicated question.

## 4. Derivation of PT

### 4.1. Mathematical Buildup and Definitions

4.1.1. Gaussian Measure

Let us rewrite the expression for the complete Green functions GF $\mathcal{Z}$ as an integral over a Gaussian measure $\gamma_G$ with functions $g$ and $j$ in $\mathcal{L}^2(\mathbb{R}^d)$ (vector space of Lebesgue square-integrable functions):

$$\mathcal{Z}[j] = \int_\Phi \gamma_G(d\phi)\, e^{-\int_{\mathbb{R}^d} dx\, g(x)|\phi(x)|^\alpha + \int_{\mathbb{R}^d} dx\, j(x)\phi(x)}, \tag{34}$$

where we understand the functional integral as the integral over HS $\Phi = L^2(\mathbb{R}^d)$—linear space of equivalence classes of functions from $\mathcal{L}^2(\mathbb{R}^d)$.

By definition, Gaussian measure $\gamma_G$ is a measure in HS $\Phi$ defined by some covariance operator (propagator) $G : \Phi \to \Phi$, which is demanded to be trace-class. Let us introduce $\{e_n\}_{n=1}^\infty$—a family of eigenvectors of $G$ with corresponding eigenvalues $\{\lambda_n\}_{n=1}^\infty$, so that $Ge_n = \lambda_n e_n$, $\forall n \in \mathbb{N}$. We also denote as $\Phi_N := \langle e_i \rangle_{i=1}^N$ the linear span of first $N$ eigenvectors, and as $P_N : \Phi \to \Phi$ a projection operator on this linear span. Then, by definition, the integral of any function $f$ depending only on $P_N\phi$ over Gaussian measure $\gamma_G$ reads:

$$\int_\Phi \gamma_G(d\phi) f[P_N\phi] = \int_{\Phi_N} \prod_{k=1}^N \frac{d\phi_k}{\sqrt{2\pi\lambda_k}} e^{-\frac{1}{2}\sum_{k=1}^N \frac{\phi_k^2}{\lambda_k}} f[P_N\phi]. \tag{35}$$

Then, under certain conditions, the integral of the generic function $f : \Phi \to \mathbb{C}$ can be computed using the Dominated Convergence Theorem (DCT for short) and convergence of the sequence $f[P_N\phi] \to f[\phi]$ as $N \to \infty$ almost everywhere:

$$\int_\Phi \gamma_G(d\phi) f[\phi] = \lim_{N\to\infty} \int_{\Phi_N} \prod_{k=1}^N \frac{d\phi_k}{\sqrt{2\pi\lambda_k}} e^{-\frac{1}{2}\sum_{k=1}^N \frac{\phi_k^2}{\lambda_k}} f[P_N\phi]. \tag{36}$$

All the described results on Gaussian measure can be found with derivation in [46,47].

4.1.2. Physical and Mathematical Approaches to Measure Theory in Infinite-Dimensional Spaces of Functions

It is useful to compare the mathematical view on functional integration with the physical one. In physics, we write Expression (30). However, there is no translation invariant countably-additive Lebesgue measure on infinite-dimensional separable BS or HS. So if one wants to develop a measure theory in separable HS, one must reject either countable additivity or translational invariance. So in mathematics, there are several approaches to integration in functional spaces, which are:

1.  Gaussian countably-additive measure (the method we use in the present paper) [46];
2.  Translational invariant finitely-additive measure [55–57];
3.  Ito integral (Wiener measure, which corresponds to significant restriction or sigma-algebra to the one generated by cylindrical sets only) [46].

In this paper, as has been previously explained, we use Gaussian measures language. In the way the physicists understand the functional measure, one can write informally:

$$\gamma_G(d\phi) = \mathcal{D}[\phi]\, e^{-\frac{1}{2}\int_{\mathbb{R}^d}\int_{\mathbb{R}^d} dx\, dy\, L(x,y)\phi(x)\phi(y)}, \quad G = L^{-1}. \tag{37}$$

It means that informally, the Gaussian measure determined by some propagator $G$ is a "physical" measure multiplied by an exponential of minus quadratic part of the action with "kinetic operator" $G = L^{-1}$.

4.1.3. About Continuity of Functionals

Let us make one important remark that will further ensure the correctness of all subsequent transformations. Consider the following functional (a function defined on a function):

$$f[\phi] = e^{-\int_{\mathbb{R}^d} dx\, g(x)|\phi(x)|^\alpha + \int_{\mathbb{R}^d} dx\, j(x)\phi(x)}, \tag{38}$$

so it is positive, measurable and $f[\phi] \leq e^{\int j(x)\phi(x)dx} \leq e^{\|j\|_2 \|\phi\|_2}$ which is clearly integrable with a finite value of the integral. Moreover, for proving that the sequence $f[P_N\phi] \to f[\phi]$ as $N \to \infty$ almost everywhere it is necessary and sufficient to show that $\int_{\mathbb{R}^d} dx\, g(x)|\phi(x)|^\alpha$ is a continuous functional on $\Phi$, since exponent and $\int_{\mathbb{R}^d} dx\, j(x)\phi(x)$ are continuous.

Let the coupling constant (non-negative function) $g$ be bounded and Lebesgue integrable. Since $\alpha \in (1,2)$, the following is true (we denote the measure for which the coupling constant is the measure density by the same letter $g$):

$$\int_{\mathbb{R}^d} dx\, g(x)|\phi(x)|^\alpha = \|\phi\|_{g,\alpha}^\alpha, \quad \int_{\mathbb{R}^d} dx\, g(x) = g(\mathbb{R}^d), \quad \sup_{x \in \mathbb{R}^d} g(x) = g_s,$$

$$\left| \|P_N\phi\|_{g,\alpha} - \|\phi\|_{g,\alpha} \right| \leq \|P_N\phi - \phi\|_{g,\alpha} \leq g(\mathbb{R}^d)^{\left(\frac{1}{\alpha}-\frac{1}{2}\right)} g_s^{\frac{1}{2}} \|P_N\phi - \phi\|_2. \tag{39}$$

Thus the functional $\int_{\mathbb{R}^d} dx\, g(x)|\phi(x)|^\alpha$ is continuous for all $\phi \in \Phi$. Here, we have used a well-known consequence of Hölder inequality for the set $\mathbb{R}^d$ with finite measure $g(\mathbb{R}^d)$. We would like to underline that this functional is not continuous when $\alpha > 2$ because in this case $\|\phi\|_{g,\alpha}$ is not finite if $\|\phi\|_2$ is finite (which is equivalent to $\phi \in \Phi$).

As a result, one can apply DCT and find that:

$$\mathcal{Z}[j] = \lim_{N \to \infty} \int_{\Phi_N} \prod_{k=1}^N \frac{d\phi_k}{\sqrt{2\pi\lambda_k}} e^{-\frac{1}{2}\sum_{k=1}^N \frac{\phi_k^2}{\lambda_k} - \int_{\mathbb{R}^d} dx\, g(x)|(P_N\phi)(x)|^\alpha + \int_{\mathbb{R}^d} dx\, j(x)(P_N\phi)(x)}. \tag{40}$$

That is quite a good outlook that if one considers a functional integral from a Gaussian measure point of view, one cannot compute a functional integral with $\alpha > 2$ (e.g., $\alpha = 4$, which is currently the most interesting in the physical case) by simply substituting $P_N\phi$ instead of $\phi$ and taking a limit $N \to \infty$. At least DCT is not applicable in this case. However, it should be remembered that DCT gives only sufficient conditions. Perhaps some other theorem may be useful in the case $\alpha > 2$.

*4.2. Reduction of Functional Integral to the Series from Finite-Dimensional Ones (GCPF Expansion)*

In this section, we are going to derive PT type series, which will converge. Since the power of potential is not a natural number, the traditional method of carrying out an interaction action from integral in the form of a differential operator is not suitable. So, we follow the other way and in some sense obtain all Feynman diagrams summed into one integral, which appears to be exactly the coefficient of $g(x_1) \ldots g(x_n)$ in the obtained series.

The described approach of constructing PT repeats and develops methods described in [1–3,21]. Let us proceed to the computation itself. Since the potential $U(\phi) = |\phi|^\alpha$ is an even function, the following is true:

$$\int_{\mathbb{R}^d} dx\, g(x)|\phi(x)|^\alpha = \frac{1}{2\pi} \int_{\mathbb{R}^d} dx\, g(x) \mathcal{F}\big[\mathcal{F}[|\phi(x)|^\alpha]\big], \tag{41}$$

where two points are important:

1.  we understand Fourier transform $\mathcal{F}$ (internal transform is performed over the variable $\phi$, but not $x$) in the sense of distributions from $\mathcal{S}'(\mathbb{R})$—the space of linear continuous functionals on the Schwartz space $\mathcal{S}(\mathbb{R})$;

2.  this distributional Fourier transform is consistent with the following formula for the integrable function $f(\phi)$:

$$\mathcal{F}[f(\phi)](t) := \int_{\mathbb{R}} d\phi \, f(\phi) \, e^{it\phi}. \tag{42}$$

Unfortunately, the potential $U(\phi) = |\phi|^{\alpha}$ have Fourier transform only in the generalized sense in $\mathcal{S}'(\mathbb{R})$. As usual, careful computations with generalized Fourier transform are quite subtle, so we would like to avoid them. For that purpose, we approximate $U(\phi)$ with some smooth and integrable function $U_\Lambda(\phi)$ from the Schwartz space $\mathcal{S}(\mathbb{R})$, so that $U_\Lambda(\phi) \to |\phi|^{\alpha}$ pointwise, when $\Lambda \to \infty$. Then both $U_\Lambda(\phi)$ and $\mathcal{F}[U_\Lambda(\phi)]$ are usual functions in $\mathcal{S}(\mathbb{R})$ rather than distributions, and we are entitled to calculate their Fourier transform via the usual integral Formula (42). We will make all the transitions for smooth and integrable function $U_\Lambda(\phi)$. Additionally, we will assume that $0 \le U_\Lambda(\phi) \le |\phi|^{\alpha}$ for all $\phi$ and that all the functions $U_\Lambda(\phi)$ are even in $\phi$, which will be useful in calculation of PT series majorant. Namely, we can write:

$$\int_{\mathbb{R}^d} dx \, g(x) |\phi(x)|^{\alpha} = \lim_{\Lambda \to \infty} \int_{\mathbb{R}^d} dx \, g(x) U_\Lambda[\phi(x)], \tag{43}$$

since $0 \le U_\Lambda[\phi] \le |\phi|^{\alpha}$ and the left-hand side integral converges absolutely, so DCT justifies this transition. After that, we are able to write:

$$\int_{\mathbb{R}^d} dx \, g(x) |\phi(x)|^{\alpha} = \lim_{\Lambda \to \infty} \frac{1}{2\pi} \int_{\mathbb{R}^d} dx \, g(x) \mathcal{F}[\mathcal{F}[U_\Lambda[\phi(x)]]], \tag{44}$$

and this formula is much more convenient for the calculations. In the following calculations, we will extract this limit from functional integral also due to DCT and will consider regularized functional $\mathcal{Z}_\Lambda[j]$. At the end of our computation, we will calculate the limit $\Lambda \to \infty$. So we start by considering the "regularized" functional:

$$\mathcal{Z}_\Lambda[j] = \int_\Phi \gamma_G(d\phi) \, e^{- \int_{\mathbb{R}^d} dx \, g(x) U_\Lambda[\phi(x)] + \int_{\mathbb{R}^d} dx \, j(x)\phi(x)}, \tag{45}$$

and due to DCT described in previous section $\mathcal{Z}_\Lambda[j] \to \mathcal{Z}[j]$, when $\Lambda \to \infty$.

Further, we write Expression (45) in terms of the Fourier transform of potential and use that $\mathcal{F}[U_\Lambda(\phi)]$ is in $\mathcal{S}(\mathbb{R})$:

$$\mathcal{Z}_\Lambda[j] = \int_\Phi \gamma_G(d\phi) \exp\left\{ -\int_{\mathbb{R}^d} dx \, g(x) \int_{\mathbb{R}} \frac{dt}{2\pi} e^{it\phi(x)} \mathcal{F}[U_\Lambda(\phi)](t) + \int_{\mathbb{R}^d} dx \, j(x)\phi(x) \right\}. \tag{46}$$

Now let us expand the external exponent in a Taylor series in powers of the coupling constant $g$:

$$\mathcal{Z}_\Lambda[j] = \int_\Phi \gamma_G(d\phi) \sum_{n=0}^{\infty} \frac{(-1)^n}{n!(2\pi)^n} \left\{ \prod_{a=1}^{n} \int_{\mathbb{R}^d} \int_{\mathbb{R}} dx_a \, dt_a \, g(x_a) \mathcal{F}[U_\Lambda(\phi)](t_a) \right\}$$
$$\times \exp\left\{ i \sum_{a=1}^{n} t_a \phi(x_a) + \int_{\mathbb{R}^d} dx \, j(x)\phi(x) \right\}. \tag{47}$$

Exchanging summation and integration, we arrive at the following expression:

$$\mathcal{Z}_\Lambda[j] = \sum_{n=0}^{\infty} \frac{(-1)^n}{n!(2\pi)^n} \left\{ \prod_{a=1}^{n} \int_{\mathbb{R}^d} \int_{\mathbb{R}} dx_a \, dt_a \, g(x_a) \mathcal{F}[U_\Lambda(\phi)](t_a) \right\}$$
$$\times \int_\Phi \gamma_G(d\phi) \exp\left\{ i \sum_{a=1}^{n} t_a \phi(x_a) + \int_{\mathbb{R}^d} dx \, j(x)\phi(x) \right\}. \tag{48}$$

The correctness of this permutation will be proven independently in the next section.

Finally, we have to integrate over $\phi$. After introducing the "modified" source $\tilde{j}$ (the second term in the right-hand side contains the Dirac delta function $\delta$):

$$\tilde{j}(x) = j(x) + i \sum_{a=1}^{n} t_a \delta(x - x_a),$$

we obtain a usual Gaussian integral with linear exponent:

$$\int_{\Phi} \gamma_G(d\phi) \, e^{\int_{\mathbb{R}^d} \tilde{j}(x)\phi(x)} = e^{\frac{1}{2}\langle \tilde{j}|G|\tilde{j}\rangle} = e^{\frac{1}{2}\int_{\mathbb{R}^d}\int_{\mathbb{R}^d} dx\,dy\,G(x,y)\tilde{j}(x)\tilde{j}(y)}. \tag{49}$$

Using Expression (49), we arrive at the following equality:

$$\mathcal{Z}_\Lambda[j] = \mathcal{Z}_0[j] \sum_{n=0}^{\infty} \frac{(-1)^n}{n!(2\pi)^{n/2}} \left\{ \prod_{a=1}^{n} \int_{\mathbb{R}^d} \int_{\mathbb{R}} dx_a \, dt_a \, g(x_a) \mathcal{F}[U_\Lambda(\phi)](t_a) \right\}$$
$$\times \exp\left\{ -\frac{1}{2} \sum_{a,b=1}^{n} (G_n)_{ab} t_a t_b - i \sum_{a=1}^{n} t_a \varphi(x_a) \right\}. \tag{50}$$

Now we are going to use Parseval–Plancherel identity, but disappointingly the matrix $(G_n)_{ab} = G(x_a, x_b)$ (the discretization of the propagator with values in the finite-dimensional space $\mathbb{R}^{n^2}$) is only semi-positive, which will be proved in Section 4.2. However, $\det(G_n)_{ab} = 0$ only on null sets (sets of zero measure), which made it possible not to take care of it up to this moment. However, in the following, we are going to approximate integrand with polynomials so we would like it to be bounded. So we introduce the $\mathcal{Z}_{\Lambda,\varepsilon}$ instead of $\mathcal{Z}_\Lambda$, adding $-\varepsilon \sum_{a=1}^{n} t_a^2$ term into the series $n$th term exponent for all $n$. Then, we will calculate the limit $\varepsilon \to +0$ in a final result, using the obtained majorant and its (in)dependence of $\varepsilon$. Hence, we have:

$$\mathcal{Z}_{\Lambda,\varepsilon}[j] = \mathcal{Z}_0[j] \sum_{n=0}^{\infty} \frac{(-1)^n}{n!(2\pi)^{n/2}} \left\{ \prod_{a=1}^{n} \int_{\mathbb{R}^d} \int_{\mathbb{R}} dx_a \, dt_a \, g(x_a) \mathcal{F}[U_\Lambda(\phi)](t_a) \right\}$$
$$\times \exp\left\{ -\frac{1}{2} \sum_{a,b=1}^{n} ((G_n)_{ab} + \varepsilon\delta_{ab}) t_a t_b - i \sum_{a=1}^{n} t_a \varphi(x_a) \right\}. \tag{51}$$

At this point, it is evident that every term in the above series converges to the term with the same number in the series for $\mathcal{Z}_\Lambda$ when $\varepsilon \to +0$.

At present, we can use Parseval–Plancherel identity:

$$\mathcal{Z}_{\Lambda,\varepsilon}[j] = \mathcal{Z}_0[j] \sum_{n=0}^{\infty} \frac{(-1)^n}{n!(2\pi)^{n/2}} \left\{ \prod_{a=1}^{n} \int_{\mathbb{R}^d} dx_a \, g(x_a) \, e^{-\frac{1}{2}\sum_{a,b=1}^{n}(R_n)_{ab}\varphi(x_a)\varphi(x_b)} \right.$$
$$\left. \times \int_{\mathbb{R}} d\phi_a \, U_\Lambda(\phi_a) \right\} \frac{e^{-\frac{1}{2}\sum_{a,b=1}^{n}(R_n)_{ab}\phi_a\phi_b + \sum_{a=1}^{n}\phi_a\chi(x_a)}}{\sqrt{\det(G_n + \varepsilon 1_n)}}. \tag{52}$$

In Expression (52), the following notations are introduced:

1.  $(R_n)_{ab} = ((G_n)_{ab} + \varepsilon\delta_{ab})^{-1}$ is the inverse of $((G_n)_{ab} + \varepsilon\delta_{ab})$ in finite-dimensional space $\mathbb{R}^{2n}$ and $(1_n)_{ab} = \delta_{ab}$ is the identity matrix in $\mathbb{R}^{2n}$;

2.  $\chi(x_a) = \sum_{b=1}^{n}(R_n)_{ab}\varphi(x_b)$ is "effective" discrete source, since it was obtained from $j$ firstly by acting the "continuous" operator $G$, and then by its "discrete" inverse.

Let us note that both $R_n$ and $G_n$ are symmetric, hence diagonalizable in an orthonormal basis by orthogonal transformation. Moreover, all $G_n$ are semi-positive, so $G_n + \varepsilon 1_n$ is positive and hence invertible. This proves that all written integrals converge absolutely, so we can interchange the order of integration in $t_a$ and $x_a$ variables.

In the Section 4.3, we will show that the obtained series converges uniformly in $\Lambda$ for generic $j$ and also in $\varepsilon$ for $j = 0$. The idea of the proof is that we can choose $U_\Lambda(\phi)$ such, that $0 \leq U_\Lambda[\phi] \leq |\phi|^\alpha$. After that, we can apply the Weierstrass M-test to the series for $\mathcal{Z}_{\Lambda,\varepsilon}$ if we prove that the series above with $U_\Lambda(\phi)$, replaced by $|\phi|^\alpha$, converges. We will do it later in the paper.

Up to this moment, we finish on the expression for the "normalized" GF $\mathcal{Z}_{I,\varepsilon}$, removing the regulator $\Lambda$:

$$\mathcal{Z}_{I,\varepsilon}[j] = \sum_{n=0}^{\infty} \frac{(-1)^n}{n!(2\pi)^{n/2}} \left\{ \prod_{a=1}^{n} \int_{\mathbb{R}^d} dx_a \, g(x_a) \, e^{-\frac{1}{2}\sum_{a,b=1}^{n}(R_n)_{ab}\varphi(x_a)\varphi(x_b)} \right.$$
$$\left. \times \int_{\mathbb{R}} d\phi_a \, |\phi_a|^\alpha \right\} \frac{e^{-\frac{1}{2}\sum_{a,b=1}^{n}(R_n)_{ab}\phi_a\phi_b + \sum_{a=1}^{n}\phi_a\chi(x_a)}}{\sqrt{\det(G_n + \varepsilon 1_n)}}, \tag{53}$$

where we denoted $\mathcal{Z}_{I,\varepsilon}[j] = \mathcal{Z}_{\Lambda \to \infty,\varepsilon}[j] / \mathcal{Z}_0[j]$, in accordance with the notations, introduced in the Section 3. In particular, we can write the value $\mathcal{Z}_I[0]$ for the case of $j = 0$, also removing the regulator $\varepsilon$:

$$\mathcal{Z}_I[0] = \sum_{n=0}^{\infty} \frac{(-1)^n}{n!(2\pi)^{n/2}} \left\{ \prod_{a=1}^{n} \int_{\mathbb{R}^d} dx_a \, g(x_a) \int_{\mathbb{R}} d\phi_a \, |\phi_a|^\alpha \right\} \frac{e^{-\frac{1}{2}\sum_{a,b=1}^{n}(G_n)_{ab}^{-1}\phi_a\phi_b}}{\sqrt{\det G_n}}. \tag{54}$$

Here, the matrix $G_n$ is degenerate on null sets, but it does not affect the integral because of the existence of majorant not depending on $\varepsilon$.

This expansion of GF in perturbation series looks similar to the expansion of the GCPF of non-ideal gas with potential $G(x_a, x_b)$, but with additional inner integrals over $\phi_a$. Keeping this in mind, we will refer to $\mathcal{Z}_{I,\varepsilon,n}$ as the $n$-particle canonical partition function. These inner integrals are in fact the key difference between statistical physics and QFT, warranting the complication of the last one.

Properties of $G_N$-Matrices

It is useful to visualize the typical structure of matrix $G_n$. We will do this for the translation invariant case (the general case is performed in a similar way), since this is the case that will be considered in all the final results. The desired expression is:

$$G_n = \begin{pmatrix} G(0) & G(x_1 - x_2) & \dots & G(x_1 - x_n) \\ G(x_1 - x_2) & G(0) & \dots & \dots \\ \dots & \dots & G(0) & G(x_n - x_{n-1}) \\ G(x_1 - x_n) & \dots & G(x_n - x_{n-1}) & G(0) \end{pmatrix}. \tag{55}$$

There are two limiting cases for this matrix:

1. When all $x_a$ are equal, then:

$$G_n = \begin{pmatrix} G(0) & G(0) & \dots & G(0) \\ G(0) & G(0) & \dots & \dots \\ \dots & \dots & G(0) & G(0) \\ G(0) & \dots & G(0)) & G(0) \end{pmatrix}.$$

2. When all $x_a$ are infinitely faraway, then:

$$G_n = \begin{pmatrix} G(0) & 0 & \ldots & 0 \\ 0 & G(0) & \ldots & \ldots \\ \ldots & \ldots & G(0) & 0 \\ 0 & \ldots & 0 & G(0) \end{pmatrix}.$$

Let us prove the semi-positivity of $G_n$. Since the operator $G : \Phi \to \Phi$ is positive, the equality $\langle j | G | j \rangle \geq 0$ holds $\forall j \in \Phi$. For distributions, this inequality also holds from the reasons of continuity. Namely, choosing a smooth sequence of functions approximating some distribution of the form:

$$j(x) = \sum_{a=1}^{n} c_a \, \delta(x - x_a),$$

we obtain exactly:

$$\langle \vec{c} | G_n | \vec{c} \rangle \geq 0, \quad \forall \vec{c} = (c_1, \ldots, c_n) \in \mathbb{R}^n,$$

which completes the proof of semi-positivity of $G_n$ for all values of $x_a$. For this reason, $G_n + \varepsilon 1_n$ is a positive definite matrix for all $n$ as well as its inverse, for any $\varepsilon > 0$.

The other way to prove this fact (in the translation invariant case) is to use Bochner's theorem, which claims the semi-positivity of $G_n$ from the fact that propagator $G(x - y)$ is from Schwartz space and its Fourier transform $\mathcal{F}[G](k) \geq 0$ is non-negative. Though, this proof needs one more additional assumption that $G(x - y) \in \mathcal{S}(\mathbb{R}^d)$.

By construction, the matrix $G_n$ is symmetric, therefore, diagonalizable. Let us denote its eigenvalues as $0 \leq \lambda_1^{(n)} \leq \lambda_2^{(n)} \leq \ldots \leq \lambda_n^{(n)}$. Since $\sum_{a=1}^{n} \lambda_a^{(n)} = \operatorname{tr} G_n = nG(0)$, then the matrix sup-norm $\|G_n + \varepsilon 1_n\| \leq \lambda_n^{(n)} + n\varepsilon \leq n(G(0) + \varepsilon)$. In particular, we found the bound for maximum eigenvalue of $G_n$ (and, simultaneously, the minimal eigenvalue of $G_n^{-1}$, when $G_n$ is invertible):

$$\lambda_{max}^{(n)} = \lambda_n^{(n)} \leq nG(0). \tag{56}$$

This bound will be used in the future computation of majorizing series.

### 4.3. Majorizing Series (Majorant)
#### 4.3.1. Coordinate-Free Part

We always can exchange summation and integration when the series terms are positive due to MCT. In addition, we have to check the absolute convergence of the obtained series to use DCT. So, it is sufficient to take the absolute value of every term, exchange sum and integral and prove that this series converges. At this moment, we have to find the upper bound of the series terms:

$$J_n(\alpha) = \frac{1}{\sqrt{\det(G_n + \varepsilon 1_n)}} \int_{\mathbb{R}^n} d\phi_1 \ldots d\phi_n \, |\phi_1|^\alpha \ldots |\phi_n|^\alpha e^{-\frac{1}{2} \sum_{a,b=1}^{n} (R_n)_{ab} \phi_a \phi_b + \sum_{a=1}^{n} \phi_a \chi(x_a)}, \tag{57}$$

where the notation of $J_n(\alpha)$ is introducing for the convenience. Now we are going to estimate $|\phi_1|^\alpha \ldots |\phi_n|^\alpha$ with restriction $\|\phi\|^2 = r^2$. Our aim is to bound the integrand on every sphere in $\mathbb{R}^n$ centered at the origin, and then use spherical coordinates. We start with the Lagrangian function ($\lambda$ is the Lagrange multiplier):

$$f(\phi) = \alpha \sum_{a=1}^{n} \ln \phi_a - \lambda \left( \|\phi\|^2 - r^2 \right),$$

and find its unconditional extremum:

$$\partial_a f(\phi) = \frac{\alpha}{\phi_a} - 2\lambda\phi_a = 0,$$

hence:

$$\phi_a^2 = \frac{\alpha}{2\lambda}.$$

Taking the sum over $a$ to obtain $\lambda$ we receive:

$$r^2 = \frac{\alpha n}{2\lambda}.$$

So, finally, the maximum is reached at:

$$\phi_a = \frac{r}{\sqrt{n}}.$$

So, one can estimate (and these bounds are strict):

$$0 \le |\phi_1|^{\alpha} \dots |\phi_n|^{\alpha} \le \frac{r^{n\alpha}}{n^{\frac{n\alpha}{2}}}. \tag{58}$$

In addition to the previous bound, we can change variables by the orthogonal transformation and diagonalize the exponent in $J_n(\alpha)$ putting $\phi_a = \sum_{b=1}^{n} (R_n)_{ab}^{-1/2} \zeta_b$, plus estimate $\|\phi\| \le \|G_n + \varepsilon 1_n\|^{\frac{1}{2}} \|\zeta\|$:

$$\frac{J_n(\alpha)}{n!} \le \frac{1}{n^{\frac{n\alpha}{2}}} \|G_n + \varepsilon 1_n\|^{\frac{n\alpha}{2}} \frac{1}{n!} \int_{\mathbb{R}^n} d\zeta_1 \dots d\zeta_n \|\zeta\|^{n\alpha} e^{-\frac{\|\zeta\|^2}{2} + \sum_{a,b=1}^{n} (R_n)_{ab}^{-1/2} \zeta_a \varphi(x_b)},$$

where the factor $\sqrt{\det(G_n + \varepsilon 1_n)}$ cancels out with the Jacobian after changing of variables in the integral. Here, we also use the fact that $G_n$ is almost everywhere invertible. Hereafter we use the notation $(A_n)_{ab}^{\beta}$ for the matrix element of a power of matrix, namely:

$$(A_n)_{ab}^{\beta} = \left(A_n^{\beta}\right)_{ab}, \tag{59}$$

for $n \times n$ matrix $A_n$ and rational number $\beta$.

Now we have to transform the part with source:

$$\left| \sum_{a,b=1}^{n} (R_n)_{ab}^{1/2} \zeta_a \varphi(x_b) \right| \le \|\zeta\| \cdot \left\| \sum_{b=1}^{n} (R_n)_{ab}^{1/2} \varphi(x_b) \right\|.$$

We denote for simplicity (which will not be confused with the source due to index 0):

$$j_0 := \left\| \sum_{b=1}^{n} (R_n)_{ab}^{1/2} \varphi(x_b) \right\| = \sqrt{\sum_{a,b=1}^{n} (R_n)_{ab} \varphi(x_a) \varphi(x_b)} \ge 0,$$

where the last transition follows from the Euclidean norm definition. We underline that this quantity depends on $x$.

In this paper we will mainly consider the calculation of vacuum energy, which corresponds to the case $j = 0$. Hence, we will not think about careful estimations for the case of the non-zero source, and write the most rough estimation. Namely:

$$j_0 \le \sqrt{\|R_n\|} \|\phi(x_a)\| \le \frac{n}{\sqrt{\varepsilon}} \sup_{x \in \mathbb{R}^d} |\phi(x)|,$$

where we used the fact that $G_n \geq 0$, and hence maximal eigenvalue of $R_n$ is no more than $1/\varepsilon$. Here, $\|\phi(x_a)\|$ is a Euclidean norm of the vector $(\phi(x_a))_{a=1}^n$, which is bounded with $n \sup\limits_{x \in \mathbb{R}^d} |\phi(x)|$. Then we have:

$$\frac{J_n(\alpha)}{n!} \leq \frac{1}{n^{\frac{n\alpha}{2}}} \|G_n + \varepsilon 1_n\|^{\frac{n\alpha}{2}} \frac{1}{n!} \int_{\mathbb{R}^n} d\zeta_1 \dots d\zeta_n \|\zeta\|^{n\alpha} e^{-\frac{\|\zeta\|^2}{2} + \|\zeta\| j_0}. \tag{60}$$

Next, we recall the inequality $\|G_n + \varepsilon 1_n\| \leq n(G(0) + \varepsilon)$. The integral in the right hand side of Expression (60) can be easily calculated via series expansion in $j_0$:

$$\frac{J_n(\alpha)}{n!} \leq (G(0) + \varepsilon)^{\frac{n\alpha}{2}} \frac{1}{n!} \frac{n\pi^{\frac{n}{2}} 2^{\frac{1}{2}(n\alpha+n-2)}}{\Gamma(\frac{n}{2}+1)} \sum_{m=0}^{\infty} \frac{2^{\frac{m}{2}} \Gamma(\frac{1}{2}(m+n+n\alpha))}{\Gamma(m+1)} j_0^m.$$

As a first step for bounding the obtained series, we use log-convexity of Gamma function $\Gamma$ for $m \geq 1$:

$$\Gamma\left(\frac{n(\alpha+1)}{2} + \frac{m}{2}\right) \leq \Gamma\left(\frac{n(\alpha+1)}{2} + \frac{m+1}{2}\right) \leq \Gamma(n(\alpha+1))^{1/2} \Gamma(m+1)^{1/2},$$

and rewrite:

$$\sum_{m=1}^{\infty} \frac{2^{\frac{m}{2}} \Gamma(\frac{1}{2}(m+n+n\alpha))}{\Gamma(m+1)} j_0^m \leq \Gamma(n(\alpha+1))^{1/2} \sum_{m=1}^{\infty} \frac{2^m}{\Gamma(m+1)^{1/2}} j_0^m.$$

Moreover, for $m \to \infty$ the equivalence relation holds:

$$\frac{\Gamma(m+1)^{1/2}}{2^{m/2} \Gamma\left(\frac{m+1}{2}\right)} \sim \frac{\sqrt{12m+1}}{2\sqrt{3}\sqrt[4]{2\pi}\sqrt[4]{m}} \to \infty,$$

and it could be numerically checked, that for $m \geq 0$:

$$\frac{\Gamma(m+1)^{1/2}}{2^{m/2} \Gamma\left(\frac{m+1}{2}\right)} \geq \frac{1}{2},$$

so the following upper bound holds:

$$\sum_{m=0}^{\infty} \frac{2^{\frac{m}{2}} \Gamma(\frac{1}{2}(m+n+n\alpha))}{\Gamma(m+1)} j_0^m \leq 2\Gamma(n(\alpha+1))^{1/2} \left(1 + \sum_{m=1}^{\infty} \frac{1}{\Gamma\left(\frac{m+1}{2}\right)} j_0^m\right).$$

The last series can be calculated exactly in terms of the error function erf:

$$\sum_{m=1}^{\infty} \frac{1}{\Gamma\left(\frac{m+1}{2}\right)} j_0^m = e^{j_0^2} j_0(\text{erf}(j_0) + 1) \leq 2e^{j_0^2} j_0,$$

so we finish at:

$$\sum_{m=0}^{\infty} \frac{2^{\frac{m}{2}} \Gamma(\frac{1}{2}(m+n+n\alpha))}{\Gamma(m+1)} j_0^m \leq 2\Gamma(n(\alpha+1))^{1/2} e^{j_0^2} (1 + j_0)$$

Let us write the result:

$$\frac{J_n(\alpha)}{n!} \leq G(0)^{\frac{n\alpha}{2}} \frac{1}{n!} \frac{n\pi^{\frac{n}{2}} 2^{\frac{1}{2}(n\alpha+n)} \Gamma(n(\alpha+1))^{1/2}}{\Gamma(\frac{n}{2}+1)} e^{j_0^2} (j_0 + 1), \tag{61}$$

Finding the asymptotic of Expression (61) right hand side, we obtain:

$$\frac{J_n(\alpha)}{n!} \leq \frac{1}{n^{3/2}} n^{\frac{n\alpha}{2}-n} G(0)^{\frac{n\alpha}{2}} \pi^{\frac{n-3}{2}} 2^{\frac{1}{2}(n\alpha+3n-1)} (\alpha+1)^{n(\alpha+1)-1/2} e^{(2-\alpha)n} e^{j_0^2} (j_0+1)$$

$$= C_n n^{\frac{n\alpha}{2}-n} G(0)^{\frac{n\alpha}{2}} e^{j_0^2} (j_0+1),$$

where $C_n > 0$ is a dimensionless constant which grows no faster than exponentially.

As a result, we see that our series converges for $\alpha < 2$. For $\alpha = 2$ it also converges, but it could be proven in a much more simple way. Namely, it is easy to calculate GF $\mathcal{Z}$ for $\alpha = 2$ exactly, which will be performed further in the paper, and the corresponding expansion will converge. For $\alpha = 2$ we will consider only the case $g(x) = g\chi_Q(x)$. As for the result, we have some at least asymptotic expansion in powers of $g$. So, since even asymptotic expansions in the predetermined system of functions (power functions of the coupling constant $\{g^n\}_{n=0}^\infty$ in our case) are unique, then these two expansions must coincide. This proves that for $\alpha = 2$ our perturbation series expansion also converges. So we have to deal with the coordinate integrals to finish the proof.

### 4.3.2. Coordinate Part

The next aim is to bound the result of coordinate integration. Namely, remembering the notation $g(\mathbb{R}^d) = \int_{\mathbb{R}^d} dx \, g(x)$, we have:

$$|\mathcal{Z}_{I,\varepsilon,n}[j]| \leq \prod_{a=1}^n \int_{\mathbb{R}^d} dx_a \, g(x_a) \, e^{-\frac{1}{2} \sum_{a,b=1}^n (R_n)_{ab} \varphi(x_a)\varphi(x_b)} \frac{J_n(\alpha)}{n!} \leq C_n n^{\frac{n\alpha}{2}-n} G(0)^{\frac{n\alpha}{2}}$$

$$\times \prod_{a=1}^n \int_{\mathbb{R}^d} dx_a \, g(x_a) \, e^{j_0^2/2} (j_0+1).$$

Recalling the upper bound for $j_0$, we obtain finally:

$$|\mathcal{Z}_{I,\varepsilon,n}[j]| \leq C_n n^{\frac{n\alpha}{2}-n} G(0)^{\frac{n\alpha}{2}} g(\mathbb{R}^d)^n \exp\left\{\frac{n^2}{2\varepsilon}\left(\sup_{x\in\mathbb{R}^d}|\phi(x)|\right)^2\right\}\left(\frac{n}{\sqrt{\varepsilon}}\sup_{x\in\mathbb{R}^d}|\phi(x)|+1\right), \quad (62)$$

so the obtained series for the GF $\mathcal{Z}_{I,\varepsilon}$ converges for $j = 0$. As for $j \neq 0$, the convergence of series requires additional research and calculation of a more accurate majorant, since the obtained one increases infinitely, when $\varepsilon \to +0$, therefore, it can be argued that the series converges only for $\varepsilon \neq 0$. Though, as it was mentioned above, we mainly restrict ourselves to the case $j = 0$.

### 4.4. Exponentiation of Series Using Meyer Cluster Expansion

We have just obtained the converging PT series for the GF $\mathcal{Z}_{I,\varepsilon}$. In terms of asymptotic series, "the exponentiation of connected diagrams" is often proven, which means roughly that the GF $\mathcal{Z}_{I,\varepsilon}$ is an exponent of other regular (in the source $j$) functional. We are going to establish the same result in our case. It will be useful since after such exponentiation (for the coupling constant $g(x) = g\chi_Q(x)$) we will obtain only the first power of a volume $V = \text{Vol} \, Q$ rather than all natural powers (in general, the first power of $g(\mathbb{R}^d)$). This fact will simplify significantly the extraction of physically measurable quantities, which will be performed in the following sections of the paper.

### 4.4.1. Formulation and Applicability

We start with the following definition. Let $(X, \Sigma, \mu)$ be a measure space, that is a triple: $X$ is a set, for example, $\mathbb{R}^N$ with some natural number $N$, $\Sigma$ is a $\sigma$-algebra on $X$ and $\mu$ is a complex-valued measure (which should not be confused with the UV-cutoff parameter $\mu$) on a measurable space $(X, \Sigma)$. Complex-valued measures are also called charges. We denote as $|\mu|$ the variation of $\mu$ (which should not be confused with the total variation—the

value of $|\mu|$ on $X$). In the case, where $\mu(dy) = g(y)\nu(dy)$ and $\nu$ is a non-negative measure, $|\mu|(dy) = |g(y)|\nu(dy)$. Given a complex-valued measurable symmetric function $\zeta$ on $X \times X$, we introduce an abstract GCPF as follows:

$$\mathcal{Z}[\mu, \zeta] = \sum_{n=0}^{\infty} \frac{1}{n!} \int_X \mu(dy_1) \dots \int_X \mu(dy_n) \prod_{1 \le i < j \le n} \left(1 + \zeta(y_i, y_j)\right). \tag{63}$$

The term $n = 0$ in the sum is defined by 1. In the case of classical gas with interaction, $n$ is the number of particles.

We denote by $\mathbb{G}_n$ the set of all (undirected, no loops) graphs with $n$ vertices, and $\mathbb{G}_{C,n} \subset \mathbb{G}_n$ the set of all connected graphs with $n$ vertices. Next, let us introduce the following combinatorial function on finite sequences $(x_1, \dots, x_n)$:

$$\varphi(x_1, \dots, x_n) = \begin{cases} 1, & \text{if } n = 1; \\ \frac{1}{n!} \sum_{\Gamma \in \mathbb{G}_{C,n}} \prod_{(i,j) \in \Gamma} \zeta(x_i, x_j), & \text{if } n \ge 2. \end{cases} \tag{64}$$

The product is taken over edges of $\Gamma$. A sequence $(x_1, \dots, x_n)$ is a cluster if the graph with $n$ vertices and two vertices $i$ and $j$ whenever $\zeta(x_i, x_j) \ne 0$ are connected. The cluster expansion allows one to express the logarithm of an abstract partition function as a sum over clusters. Namely, the following theorem holds [58].

**Theorem 1** (Cluster expansion). *Assume that $|1 + \zeta(u, y)| \le 1$ for all $u, y \in X$, and that there exists a non-negative function $a$ on $X$ such that for all $u \in X$,*

$$\int_X |\zeta(u, y)| e^{a(y)} |\mu|(dy) \le a(u), \tag{65}$$

*and $\int_X e^{a(x)} |\mu|(dx) < \infty$. Then the following is true:*

$$\mathcal{Z}[\mu, \zeta] = \exp\left\{ \sum_{n=1}^{\infty} \int_X \mu(dy_1) \dots \int_X \mu(dy_n) \varphi(y_1, \dots, y_n) \right\}. \tag{66}$$

*Combined sum and integrals converge absolutely. Furthermore, we have for all $y_1 \in X$:*

$$1 + \sum_{n=2}^{\infty} n \int_X |\mu|(dy_2) \dots \int_X |\mu|(dy_n) \varphi(y_1, \dots, y_n)| \le e^{a(y_1)}. \tag{67}$$

Unfortunately, we cannot apply this theorem to the terms $\mathcal{Z}_{I,\varepsilon,n}$ of the PT series for GF $\mathcal{Z}_{I,\varepsilon}$, since there appears $\det G_n$, spoiling the form of the terms in the GCPF expansion. So, we apply it to the terms $\mathcal{Z}_{I,\Lambda,\varepsilon,n}$ of the PT series for GF $\mathcal{Z}_{I,\Lambda,\varepsilon}[j] = \mathcal{Z}_{\Lambda,\varepsilon}[j]/\mathcal{Z}_0[j]$ in the form before using Parseval–Plancherel identity. Hence, we start from ($G_\varepsilon = G(0) + \varepsilon$ for short):

$$\mathcal{Z}_{I,\Lambda,\varepsilon}[j] = \sum_{n=0}^{\infty} \frac{1}{n!} \left\{ \prod_{a=1}^{n} \underbrace{\int_{\mathbb{R}^d} dx_a (-1) g(x_a) \int_{\mathbb{R}} \frac{dt_a}{2\pi} \mathcal{F}[U_\Lambda(\phi)](t_a) \exp\left(-it_a \varphi(x_a) - \frac{1}{2} G_\varepsilon t_a^2\right)}_{\int_{\mathbb{R}^{d+1}} \mu(dx_a \, dt_a)} \right.$$

$$\left. \times \prod_{a < b}^{n} \exp(-G(x_a - x_b) t_a t_b). \right.$$

Let us note that, for clarity from statistical physics point of view, we consider the case of translation invariant propagator $G(x_a - x_b)$, but all the formulas presented below can be easily transferred to the general case of propagator $G(x_a, x_b)$.

In our case, the point $y = (x,t)$, the function $1 + \zeta(y_a, y_b) = e^{-\frac{1}{2}G(x_a - x_b)t_a t_b}$, and the measure $\mu$, appearing in the general theory, has the form:

$$\mu(dx_a \, dt_a) = -\frac{dx_a \, dt_a}{2\pi} \, g(x_a) \mathcal{F}[U_\Lambda(\phi)](t_a) \exp\left(it_a \varphi(x_a) - \frac{1}{2}G_\varepsilon t_a^2\right). \tag{68}$$

To use the theorem, one must find the function $a$ from its formulation. However, in the following we will perform all the manipulations directly and see that in our case all the transitions are valid without explicitly presenting such a function.

### 4.4.2. Rewriting Series in Terms of Exponent

Henceforth, we will keep in mind the measure Definition (68). Hereby, we start from the expansion of GF $\mathcal{Z}_{I,\Lambda,\varepsilon}$:

$$\mathcal{Z}_{I,\Lambda,\varepsilon}[j] = \sum_{n=0}^{\infty} \frac{1}{n!} \left\{ \prod_{a=1}^{n} \int_{\mathbb{R}^{d+1}} \mu(dx_a \, dt_a) \right\} \prod_{a<b}^{n} \exp(-G(x_a - x_b)t_a t_b). \tag{69}$$

Using the definition of the function $\zeta$, we rewrite Expression (69) as follows:

$$\mathcal{Z}_{I,\Lambda,\varepsilon}[j] = \sum_{n=0}^{\infty} \frac{1}{n!} \left\{ \prod_{a=1}^{n} \int_{\mathbb{R}^{d+1}} \mu(dx_a \, dt_a) \right\} \sum_{\Gamma \in \mathbb{G}_n} \prod_{(a,b) \in \Gamma} \zeta(x_a, t_a, x_b, t_b). \tag{70}$$

Further, the sum over all possible graphs $\Gamma$ in the right hand side of Expression (70) can be represented in the following way:

$$\sum_{\Gamma \in \mathbb{G}_n} \prod_{(a,b) \in \Gamma} \zeta(x_a, t_a, x_b, t_b) = \sum_{k=1}^{n} \frac{1}{k!} \sum_{V_1, \ldots, V_k} \sum_{\Gamma_1, \ldots, \Gamma_k} \prod_{l=1}^{k} \prod_{(a,b) \in \Gamma_l} \zeta(x_a, t_a, x_b, t_b). \tag{71}$$

In Expression (71) we have decomposed the graph $\Gamma$ into the connected parts (graphs) $(\Gamma_1, \ldots, \Gamma_k)$. Each $\Gamma_i \in \mathbb{G}_{C,n}$ is a connected graph with a set of vertices $V_i$ with the cardinality $n_i$. All the sets $V_i$ form a partition of the set $\{1, \ldots, n\} : V_1 \cup \ldots \cup V_k = \{1, \ldots, n\}$, and $V_i \cap V_j = \varnothing$ if $i \neq j$. There are $k!$ such sequences for each $\Gamma$, since the order of the sets $\Gamma_i$ does not matter. The sum over $\Gamma$ can thus be realized by first summing over $k$, then over the partitions $V_1, \ldots, V_k$, and then over connected graphs on the sets $V_i$. After substituting this decomposition into the GF $\mathcal{Z}_{I,\Lambda,\varepsilon}$ we can sum over the cardinalities $n_1, \ldots, n_k \geq 1$ with condition $n_1 + \cdots + n_k = n$. Additionally, the number of partitions of $n$ elements into $k$ subsets with $n_1, \ldots, n_k$ elements is given by the multinomial coefficient $\frac{n!}{n_1! \ldots n_k!}$, which therefore should be inserted. As a result, the following chain of equalities for GF $\mathcal{Z}_{I,\Lambda,\varepsilon}$ is valid:

$$
\begin{aligned}
\mathcal{Z}_{I,\Lambda,\varepsilon}[j] &= 1 + \sum_{n=1}^{\infty} \frac{1}{n!} \sum_{k=1}^{n} \frac{1}{k!} \sum_{\substack{n_1, \ldots, n_k \\ n_1 + \ldots + n_k = n}} \frac{n!}{n_1! \ldots n_k!} \\
&\times \prod_{l=1}^{k} \left\{ \sum_{\Gamma_l \in \mathbb{G}_{C,n_l}} \int_{\mathbb{R}^{d+1}} d\mu(dx_1 \, dt_1) \ldots \int_{\mathbb{R}^{d+1}} d\mu(dx_{n_l} \, dt_{n_l}) \prod_{(a,b) \in \Gamma_l} \zeta(x_a, t_a, x_b, t_b) \right\} \\
&= 1 + \sum_{k=1}^{\infty} \frac{1}{k!} \left\{ \sum_{n=1}^{\infty} \frac{1}{n!} \sum_{\Gamma \in \mathbb{G}_{C,n}} \left\{ \prod_{a=1}^{n} \int_{\mathbb{R}^{d+1}} \mu(dx_a \, dt_a) \right\} \prod_{(a,b) \in \Gamma} \zeta(x_a, t_a, x_b, t_b) \right\}^k \\
&= \exp\left\{ \sum_{n=1}^{\infty} \frac{1}{n!} \sum_{\Gamma \in \mathbb{G}_{C,n}} \left\{ \prod_{a=1}^{n} \int_{\mathbb{R}^{d+1}} \mu(dx_a \, dt_a) \right\} \prod_{(a,b) \in \Gamma} \zeta(x_a, t_a, x_b, t_b) \right\}.
\end{aligned}
\tag{72}
$$

In accordance with the notations, introduced in the Section 3, let us denote by $\mathcal{G}_{I,\Lambda,\varepsilon} = \ln \mathcal{Z}_{I,\Lambda,\varepsilon}$ the "normalized" GF of connected Green functions with regulations $\Lambda$ and $\varepsilon$. Applying the cluster expansion we made above, we have the following formula for it:

$$\mathcal{G}_{I,\Lambda,\varepsilon}[j] = \sum_{n=1}^{\infty} \frac{1}{n!} \sum_{\Gamma \in \mathbb{G}_{C,n}} \left\{ \prod_{a=1}^{n} \int_{\mathbb{R}^{d+1}} \mu(dx_a \, dt_a) \right\} \prod_{(a,b) \in \Gamma} \zeta(x_a, t_a, x_b, t_b). \tag{73}$$

To go further, recall the definition of the adjacency matrix $\nu_{ab}(\Gamma)$ for a given graph $\Gamma$:

$$\nu_{ab}(\Gamma) = \begin{cases} 1, & (a,b) \in \Gamma; \\ 0, & (a,b) \notin \Gamma. \end{cases} \tag{74}$$

We note that $\nu_{aa} = 0$, which is equivalent to our already existing requisition for the graph to have no loops. Using this definition, as well as the definition of the function $\zeta$, we can write and transform the following product:

$$\prod_{(a,b) \in \Gamma} \zeta(x_a, t_a, x_b, t_b) = \prod_{(a,b) \in \Gamma} (\exp(-G(x_a - x_b)t_a t_b) - 1)$$

$$= \prod_{(a,b) \in \Gamma} \int_0^1 ds_{ab} \, \partial_{s_{ab}} \exp(-s_{ab} G(x_a - x_b)t_a t_b) \tag{75}$$

$$= \left\{ \prod_{a<b}^{n} \int_0^1 ds_{ab} \, \partial_{s_{ab}}^{\nu_{ab}(\Gamma)} \right\} \exp\left\{ -\sum_{a<b}^{n} s_{ab} \nu_{ab}(\Gamma) G(x_a - x_b) t_a t_b \right\}$$

We simply have presented the differences as integrals of derivatives ($s_{ab}$ are the auxiliary variables), using the Fundamental Theorem of Calculus. We have also expanded the sum in the exponent to all graph edges and added $\nu_{ab}(\Gamma)$ multipliers that do not affect the terms for presenting edges and annihilate the contributions of the terms corresponding to the absent ones. Further, we understand the powers of differential operators $\partial_{s_{ab}}$ as follows:

$$\partial_{s_{ab}}^{\nu_{ab}(\Gamma)} = \begin{cases} \partial_{s_{ab}}, & (a,b) \in \Gamma; \\ 1, & (a,b) \notin \Gamma. \end{cases} \tag{76}$$

This means exactly that the edges $(a,b)$, that are absent in the graph $\Gamma$, contribute only by 1, since for them $\nu_{ab}(\Gamma) = 0$. In addition, as for the presenting edges, they give the desired difference after the action of the integral of the derivative. Such rewriting of the products provides an unification of the expressions and will be extremely useful for obtaining compact forms of the results that follow in the paper. Finally, in accordance with Section 2, we recall the convenient notation:

$$(G_{n,\Gamma})_{ab} = s_{ab} \nu_{ab}(\Gamma) G(x_a - x_b) + G_{\varepsilon} \delta_{ab}. \tag{77}$$

This quantity depends on $\varepsilon$ and $s_{ab}$, but we will not specify it in the notations for shortness.

Traveling back to the GF $\mathcal{G}_{I,\Lambda,\varepsilon}$ in terms of the PT series $\mathcal{G}_{I,\Lambda,\varepsilon} = \sum_{n=1}^{\infty} \mathcal{G}_{I,\Lambda,\varepsilon,n}$, we find for the $n$-particle contribution, which is the contribution of all connected graphs with $n$ vertexes, the following expression:

$$\mathcal{G}_{I,\Lambda,\varepsilon,n}[j] = \frac{1}{n!} \sum_{\Gamma \in \mathbb{G}_{C,n}} \left\{ \prod_{a<b}^{n} \int_0^1 ds_{ab} \, \partial_{s_{ab}}^{\nu_{ab}(\Gamma)} \right\} \left\{ \prod_{a=1}^{n} \int_{\mathbb{R}^{d+1}} \mu(dx_a \, dt_a) \right\} \exp\left\{ -\sum_{a<b}^{n} (G_{n,\Gamma})_{ab} t_a t_b \right\}.$$

We underline that in the PT series for the connected Green functions GF $\mathcal{G}_{I,\Lambda,\varepsilon}$ the index $n$ starts from one rather than zero, as in the PT series for the complete Green functions GF $\mathcal{Z}_{I,\Lambda,\varepsilon}$ in (69). It is not a mistake, and it arrives from the very Statement (66) of the cluster expansion theorem.

After substituting the measure $\mu(dx_a \, dt_a)$ into Definition (68), we arrive at the following expression for the $n$-particle contribution:

$$
\mathcal{G}_{I,\Lambda,\varepsilon,n}[j] = \frac{1}{n!} \sum_{\Gamma \in \mathbb{G}_{C,n}} \left\{ \prod_{a<b}^{n} \int_{0}^{1} ds_{ab} \, \partial_{s_{ab}}^{\nu_{ab}(\Gamma)} \right\} \left\{ \prod_{a=1}^{n} \int_{\mathbb{R}^d} dx_a \, (-1)g(x_a) \right\}
$$
$$
\times \left\{ \prod_{a=1}^{n} \int_{\mathbb{R}} \frac{dt_a}{2\pi} \mathcal{F}[U_\Lambda(\phi)](t_a) \exp\left( it_a \varphi(x_a) - \frac{1}{2} G_\varepsilon t_a^2 \right) \right\} \exp\left\{ -\sum_{a<b}^{n} (G_{n,\Gamma})_{ab} t_a t_b \right\},
\tag{78}
$$

and, after applying Parseval–Plancherel identity to integrals over $t_a$, the expression for the $n$-particle contribution becomes:

$$
\mathcal{G}_{I,\Lambda,\varepsilon,n}[j] = \frac{(-1)^n}{n!(2\pi)^{n/2}} \sum_{\Gamma \in \mathbb{G}_{C,n}} \left\{ \prod_{a<b}^{n} \int_{0}^{1} ds_{ab} \, \partial_{s_{ab}}^{\nu_{ab}(\Gamma)} \right\} \left\{ \prod_{a=1}^{n} \int_{\mathbb{R}^d} dx_a \, g(x_a) \right\} \frac{1}{\sqrt{\det(G_{n,\Gamma})}}
$$
$$
\times \left\{ \prod_{a=1}^{n} \int_{\mathbb{R}} d\phi_a \, U_\Lambda(\phi_a) \right\} \exp\left\{ -\frac{1}{2} \sum_{a,b=1}^{n} \left( (G_{n,\Gamma})^{-1} \right)_{ab} (\phi_a - \varphi(x_a))(\phi_b - \varphi(x_b)) \right\}.
\tag{79}
$$

This formula is not a very convenient one, since $s_{ab}$ variables are included in a very complicated way because of $\det(G_{n,\Gamma})$ and $G_{n,\Gamma}^{-1}$. Though, this expression is a very important one. Indeed, it explains that it is sufficient to consider the terms $\mathcal{Z}_{I,\Lambda,\varepsilon,n}$ in the initial (unexponentiated) GF $\mathcal{Z}_{I,\Lambda,\varepsilon}$, and then, in order to obtain $\mathcal{G}_{I,\Lambda,\varepsilon,n}$, it is enough to apply the operator $\mathcal{O}$ (everything that the operator depends on is omitted in the notation):

$$
\mathcal{O} = \sum_{\Gamma \in \mathbb{G}_{C,n}} \left\{ \prod_{a<b}^{n} \int_{0}^{1} ds_{ab} \, \partial_{s_{ab}}^{\nu_{ab}(\Gamma)} \right\},
\tag{80}
$$

and we will use this notation in the following.

Summarising, we can obtain the results for the connected Green functions GF $\mathcal{G}_{I,\Lambda,\varepsilon}$ from the ones for $\mathcal{Z}_{I,\Lambda,\varepsilon}$ with the following steps:

1. changing $(G_n)_{ab} \to (G_{n,\Gamma})_{ab}$ in the **quadratic** part of the action;
2. posterior applying the operator $\mathcal{O}$.

Equivalently, on the language of formulas:

$$
\mathcal{G}_{I,\Lambda,\varepsilon,n}[j, (G_n)_{ab}] = \mathcal{O} \mathcal{Z}_{I,\Lambda,\varepsilon,n}[\varphi, (G_{n,\Gamma})_{ab}].
\tag{81}
$$

We additionally declared in the arguments of GFs the matrices $G_n$ and $G_{n,\Gamma}$ we use in the obtained perturbative expansions to avoid confusions. The notation $\mathcal{Z}_{I,\Lambda,\varepsilon,n}[\varphi, (G_{n,\Gamma})_{ab}]$ means that this functional must be expressed in terms of $\phi$, which is fixed, and the replacement of $(G_{n,\Gamma})_{ab}$ is made only in explicit dependence.

Let us also notice the form of $\mathcal{G}_{I,\Lambda,\varepsilon,n}$ without introducing new variables $s_{ab}$:

$$
\mathcal{G}_{I,\Lambda,\varepsilon,n}[j] = \frac{1}{n!} \sum_{\Gamma \in \mathbb{G}_{C,n}} \left\{ \prod_{a=1}^{n} \int_{\mathbb{R}^{d+1}} \mu(dx_a \, dt_a) \right\} \prod_{a<b}^{n} (\exp(-\nu_{ab}(\Gamma) G(x_a - x_b) t_a t_b) - 1).
\tag{82}
$$

Expression (82) will be useful for the following HSG approximation.

### 4.4.3. Final Form of the Connected Green Functions GF

For the convenience, in accordance with Section 2, we recall the notation $\mathcal{Q}_{n,\Gamma}$ for the $n$-particle quantum entangler:

$$
\mathcal{Q}_{n,\Gamma}(x, \phi) = \frac{1}{2} \sum_{a,b=1}^{n} (G_{n,\Gamma})_{ab}^{-1} \phi_a \phi_b,
\tag{83}
$$

and also we introduce the notation for the linear functional in vector $\phi_a$:

$$l(x,\phi) = \sum_{a,b=1}^{n} (G_{n,\Gamma})_{ab}^{-1} \phi_a \varphi(x_b) = \sum_{a=1}^{n} \phi_a \chi_a, \quad \chi_a = \sum_{b=1}^{n} (G_{n,\Gamma})_{ab}^{-1} \varphi(x_b). \tag{84}$$

Basically, $\chi_a$ are the sources in the coordinate representation, which are acted by the operator $G$, after which we "pull them back" by the discretization of the propagator $G$ "modulated" as in Expression (77) inverse.

Now we are going to substitute the introduced notations in Expression (79) as well as remove regulator $\Lambda$ from it. We are eligible for the last action since the majorant for $\mathcal{Z}_{I,\Lambda,\varepsilon}$ does not depend on $\Lambda$ for generic $j$ as well as on $\varepsilon$ for $j = 0$. Hence, we have:

$$\begin{aligned} \mathcal{G}_{I,\varepsilon,n}[j] = {} & \frac{(-1)^n}{n!(2\pi)^{n/2}} \, \mathcal{O}\Bigg\{ \left\{ \prod_{a=1}^{n} \int_{\mathbb{R}^d} dx_a \, g(x_a) \right\} \frac{1}{\sqrt{\det(G_{n,\Gamma})}} \\ & \times \exp(-\mathcal{Q}_{n,\Gamma}(x,\varphi)) \left\{ \prod_{a=1}^{n} \int_{\mathbb{R}} d\phi_a \, |\phi_a|^\alpha \right\} \exp(-\mathcal{Q}_{n,\Gamma}(x,\phi) + l(x,\phi)) \Bigg\}, \end{aligned} \tag{85}$$

for an arbitrary source $j$. Here, exactly as in the (53), we introduced the notation:

$$\mathcal{G}_{I,\varepsilon}[j] = \mathcal{G}_{\Lambda \to \infty, \varepsilon}[j] - \mathcal{G}_0[j].$$

Additionally, for $j = 0$ we can remove all the regulators and write:

$$\mathcal{G}_{I,n}[0] = \frac{(-1)^n}{n!(2\pi)^{n/2}} \, \mathcal{O}\Bigg\{ \left\{ \prod_{a=1}^{n} \int_{\mathbb{R}^d} dx_a \, g(x_a) \right\} \left\{ \prod_{a=1}^{n} \int_{\mathbb{R}} d\phi_a \, |\phi_a|^\alpha \right\} \frac{\exp(-\mathcal{Q}_{n,\Gamma}(x,\phi))}{\sqrt{\det(G_{n,\Gamma})}} \Bigg\}. \tag{86}$$

These formulas will be used intensively in the Section 5.

### 4.4.4. A Short Way to Obtain Coefficients of Exponentiation

At the end of the section, consider again the coupling constant $g(x) = g\chi_Q(x)$. If the expression $\mathcal{Z}(V,G,g) = e^{V \cdot f(G,g)}$ is proved, then the following equality is true:

$$\mathcal{Z}(V,G,g) = 1 + Vf(G,g) + \frac{1}{2}V^2 f(G,g)^2 + \dots \tag{87}$$

So, we can extract $f(G,g)$ as the coefficient in $V$ if we obtain some expansion of $\mathcal{Z}(V,G,g)$ in powers of $V$. This follows from the uniqueness of (asymptotic) expansion in a prescribed system of functions (power functions of the volume $\{V^k\}_{k=0}^{\infty}$ in our case).

## 5. Calculation of PT Series Terms

Let us start by noting that the integrals of the form:

$$\int_{\mathbb{R}^n} d\phi_1 \dots d\phi_n \, |\phi_1|^\alpha \dots |\phi_n|^\alpha e^{-\frac{1}{2}\sum\limits_{a,b=1}^{n} R_{ab}^{(n)} \phi_a \phi_b + \sum\limits_{a=1}^{n} \phi_a \chi(x_a)}, \tag{88}$$

are the particular cases of so-called Gelfand hypergeometric functions [59]. Unfortunately, they are too generic and their properties are excessively complicated. However, if we are going to obtain some practical results it is pointless to express anything via them. Moreover, these integrals are sophisticated enough to be expressed simply in terms of even generalized Gauss and Lauricella hypergeometric functions.

### 5.1. Off-Diagonal Terms Expansion

The first and most evident way is to expand the quadratic exponent in its off-diagonal terms, and then compute the obtained integrals directly. However, this approach also fails

because of the excessive complexity of the obtained terms. Let us demonstrate it. We start with the expansion for the zero source:

$$\mathcal{Z}_I[0] = \sum_{n=0}^{\infty} \frac{(-1)^n}{n!(2\pi)^{n/2}} \left\{ \prod_{a=1}^{n} \int_{\mathbb{R}^d} dx_a\, g(x_a) \int_{\mathbb{R}} d\phi_a\, |\phi_a|^{\alpha} \right\} \times \frac{e^{-\frac{1}{2} \sum_{a,b=1}^{n} (G_n)_{ab}^{-1} \phi_a \phi_b}}{\sqrt{\det(G_n)}}. \tag{89}$$

Here, we have already removed the regulators $\Lambda$ and $\varepsilon$ in accordance with (54). Then, substituting the series:

$$e^{-\sum_{a<b}^{n} (G_n)_{ab}^{-1} \phi_a \phi_b} = \left\{ \prod_{a<b}^{n} \sum_{l_{ab}=1}^{\infty} \right\} (-1)^{\sum_{a<b}^{n} l_{ab}} \prod_{a<b}^{n} \frac{\left( (G_n)_{ab}^{-1} \right)^{l_{ab}}}{l_{ab}!} \prod_{a=1}^{n} \phi_a^{\sum_{b|b\neq a}^{n} l_{ab}}, \tag{90}$$

and calculating integrals over $\phi_a$, we arrive at the following expression:

$$\mathcal{Z}_I[0] = \sum_{n=0}^{\infty} \frac{(-1)^n}{n!(2\pi)^{n/2}} \left\{ \prod_{a<b}^{n} \sum_{l_{ab}=1}^{\infty} \right\} \left( -\sqrt{\frac{2}{G(0)}} \right)^{n+1+\sum_{a<b}^{n} l_{ab}}$$

$$\times \left\{ \prod_{a=1}^{n} \int_{\mathbb{R}^d} dx_a\, g(x_a) \right\} \frac{1}{\sqrt{\det(G_n)}} \prod_{a<b}^{n} \frac{\left( (G_n)_{ab}^{-1} \right)^{l_{ab}}}{l_{ab}!}. \tag{91}$$

There are two main obstacles for applying this formula:

1.  it is necessary to calculate the inverse of $n \times n$ matrix, depending on $x_a$;
2.  it is necessary to integrate the elements of the inverse matrix over $x_a$.

These are the main reasons we will develop other computational techniques for the practical calculation of some concrete quantities.

*5.2. Approximation of Generic Term via Polynomials*

In this section we will approximate prefactor $|\phi_1 \dots \phi_n|^{\alpha}$ as a function $f(\zeta) := |\zeta|^{\alpha}$ for $\zeta = \phi_1 \dots \phi_n$ with polynomials. There are a couple of constructive methods, and we compare several of them. We will consider three types of polynomial approximations:

1.  Legendre polynomial approximation,
2.  Chebyshev polynomial approximation,
3.  Bernstein polynomial approximation.

However, we will present the explicit formulas only for the Legendre polynomials, since they will be the most suitable for solving the problems formulated in our paper. To show this, we will consider other families of polynomials. However, they will not be required to display our final results, so they will not be explicitly given. The complete guide about all the mentioned families of polynomials can be found in [60,61].

5.2.1. Background on Constructive Approximation with Legendre Polynomials

Legendre polynomials are defined as an orthogonal system with respect to the integral scalar product generated with the Lebesgue measure on the closed interval $[-1, 1]$. That is, $P_n$ is a polynomial of degree $n$, such that (we denoted the independent variable as $t$, since the domains of $t$ and $\zeta$ do not coincide):

$$\int_{-1}^{1} dt\, P_m(t) P_n(t) = 0 \quad \text{if } n \neq m. \tag{92}$$

The standardization $P_n(1) = 1$ fixes the normalization of the Legendre polynomials with respect to the $L^2$-norm on the closed interval $[-1, 1]$. There is an explicit formula for the Legendre polynomial of degree $n$:

$$P_n(t) = 2^n \sum_{k=0}^{n} \binom{n}{k} \binom{\frac{n+k-1}{2}}{n} t^k, \quad t \in [-1, 1], \tag{93}$$

where $\binom{n}{k}$ is the binomial coefficient, and they obey:

$$\int_{-1}^{1} dt\, P_m(t) P_n(t) = \frac{2}{2n+1} \delta_{mn}. \tag{94}$$

The feature of Formula (93) is that due to the second binomial coefficient, all the terms for $k$ with the different residue modulo 2 are zero. So, for even $n$ polynomial $P_n$ consists of even monomials, and for odd $n$—of odd monomials. So, about a half of the terms in the sum in (93) are zero, and one should bear this in mind during practical calculations. The first few even Legendre polynomials according to this definition are:

$$P_0(t) = 1, \quad P_2(t) = \frac{1}{2}(3t^2 - 1), \quad P_4(t) = \frac{1}{8}(35t^4 - 30t^2 + 3). \tag{95}$$

Further, for the absolutely continuous function $f$ with derivative $f'$ of bounded variation on $[-1, 1]$, the expansion in Legendre polynomials converges uniformly and absolutely:

$$\sum_{n=0}^{\infty} c_n P_n(t) \underset{[-1,1]}{\rightrightarrows} f(t), \tag{96}$$

due to Jackson's theorem [60]. Here, the coefficients $c_n$ are determined by the formulas:

$$c_0 = \frac{1}{2} \int_{-1}^{1} dt\, f(t), \quad c_{n>0} = \frac{2n+1}{2} \int_{-1}^{1} dt\, f(t) P_n(t). \tag{97}$$

There also exists a bound for the residue of the series truncated on $N$th term, but it is not very useful for our case. It is so because it is an estimate for the absolute value of error, so the consequent estimation of GF will be excessively rough, since we have to take the sign into account.

As it will be explained in the next subsection, we will approximate a function $f(t) = |t|^\alpha$, for $\alpha \in (1, 2)$ and $t \in [-1, 1]$, choosing some "normalized" variable $t$. For the consideration of the case, the conditions of Jackson's theorem are satisfied for $\alpha \in (1, 2)$, so we have the uniform and absolute convergence of Legendre series to $f(t)$. We write for the finite-terms approximation, using that $f$ is even (for the convenience of writing some further formulas, we denote the coefficients of the Legendre polynomial as $u_i$), the following expression:

$$h_N(t) = \sum_{q=0}^{N} c_q(f) P_{2q}(t), \quad P_{2q}(t) = \sum_{i=0}^{q} u_i t^{2i}. \tag{98}$$

Let us note that because of the absence of the odd Legendre polynomials in the approximation, we will use the numeration for the approximation coefficients as in (98) rather than in (97). Ultimately, to avoid any misconceptions, we will write the coefficients with the "new" numeration $c_q(f)$ rather than $c_q$. Let us also note that the total degree of polynomial approximation $h_N$ is $\deg h_N = 2N$.

The coefficients $c_q(f)$ for $f(t) = |t|^\alpha$ have the form:

$$c_{q>0}(f) = 2\left(2q + \frac{1}{2}\right) \int_0^1 dt\, t^\alpha P_{2q}(t) = \left(2q + \frac{1}{2}\right) \sum_{p=0}^{q} \binom{2q}{2p} \binom{q+p-\frac{1}{2}}{2q} \frac{2^{2q+1}}{2p + \alpha + 1}, \tag{99}$$

and this formula is fair also for $q = 0$, as a computation shows. So for $h_N$ we obtain:

$$h_N(t) = \sum_{q=0}^{N} \left(2q + \frac{1}{2}\right) \sum_{p=0}^{q} \binom{2q}{2p} \binom{q + p - \frac{1}{2}}{2q} \frac{2^{4q+1}}{2p + \alpha + 1} \sum_{i=0}^{q} \binom{2q}{2i} \binom{q + i - \frac{1}{2}}{2q} t^{2i}. \quad (100)$$

In the following subsections we will use only Legendre polynomials. Therefore we can expand $f$ in a series, putting $N \to \infty$. However, for the comparison of the Legendre approximation with the Bernstein one (which is only approximation rather than expansion), we will keep $N$ finite until the end of the different polynomial approximations analysis.

5.2.2. Construction of Approximation for the Terms

According to theorems on polynomial approximation, e.g., the Stone–Weierstrass theorem, it is possible to approximate the function by polynomials on compacts only. Moreover, if we try to approximate (even pointwise) a function $|\zeta|^\alpha$ for all $\zeta \in \mathbb{R}$ via any kind of polynomials, e.g., Hermite or Laguerre polynomials (multiplied by the corresponding weight functions so that such expansions converge), we obtain a diverging series after term-by-term integration. Moreover, there is a problem in the radial (in terms of spherical coordinates in the $\mathbb{R}^n$) direction. As a possible solution, before approximating by polynomials, one should evaluate the integral (88) in the radial direction, introducing spherical coordinates in $\mathbb{R}^n$ (hyperspherical coordinates, if $n > 3$, but for brevity we will use this term for $n > 1$). In the remaining integral over hypersphere we can approximate $|t|^\alpha$, where $t$—some convenient function on the hypersphere with values in $[-1, 1]$, by polynomials in $t$. There will be no such a problem since on a hypersphere $t \in [-1, 1]$, so all the approximations and expansions will converge. After such an approximation one can go back to $\mathbb{R}^n$ where it is easier to calculate the integrals using the source trick. It is a brief summary of the material in the current subsection.

Let us denote the integral, which we are going to work with, as ($n > 1$):

$$I_n = \left\{ \prod_{a=1}^{n} \int_{\mathbb{R}} d\phi_a \, |\phi_a|^\alpha \right\} \exp(-\mathcal{Q}_{n,\Gamma}(x, \phi) + l(x, \phi)). \quad (101)$$

The integrand has the symmetry under $\phi \to -\phi$, since $\mathcal{Q}_{n,\Gamma}$ and $|\phi_a|^\alpha$ are even functions, hence (in terms of the hyperbolic cosine):

$$I_n = \left\{ \prod_{a=1}^{n} \int_{\mathbb{R}} d\phi_a \, |\phi_a|^\alpha \right\} \exp(-\mathcal{Q}_{n,\Gamma}(x, \phi)) \cosh(l(x, \phi)). \quad (102)$$

In this integral we can introduce hyperspherical coordinates and proceed as:

$$I_n = \int_0^\infty dr \int_{S^{n-1}} d\Omega \, r^{n-1+n\alpha} |\xi_1 \ldots \xi_n|^\alpha \exp\left(-r^2 \mathcal{Q}_{n,\Gamma}(x, \xi)\right) \cosh(r l(x, \xi)), \quad (103)$$

where we choose $\phi_a = r\xi_a$ and $\vec{\xi} \in S^{n-1}$ is the unit normal vector. Integration over $r$ gives:

$$I_n = \frac{1}{2} \int_{S^{n-1}} d\Omega \, |\xi_1 \ldots \xi_n|^\alpha \frac{\Gamma\left(\frac{n(\alpha+1)}{2}\right)}{\mathcal{Q}_{n,\Gamma}(x, \xi)^{\frac{1}{2}(\alpha+1)n}} \, {}_1F_1\left(\frac{n(\alpha+1)}{2}; \frac{1}{2}; \frac{l(x, \xi)^2}{4\mathcal{Q}_{n,\Gamma}(x, \xi)}\right). \quad (104)$$

In Expression (104) we follow the commonly accepted notation of ${}_1F_1$ for the confluent hypergeometric function. This function is defined in terms of the following series (and this series converges for any finite value of $z$ and thus defines an entire function of $z$):

$$_1F_1(a; b; z) = \sum_{n=0}^{\infty} \frac{(a)_n z^n}{(b)_n n!}, \quad (105)$$

where $(a)_n$ is a rising factorial. Then we expand $_1F_1$ and obtain the following expression:

$$I_n = \frac{1}{2}\sum_{k=0}^{\infty}\frac{1}{4^k k!}\frac{\left(\frac{n(\alpha+1)}{2}\right)_{2k}}{(1/2)_{2k}}\Gamma\left(\frac{n(\alpha+1)}{2}\right)\int_{S^{n-1}}\frac{d\Omega}{\mathcal{Q}_{n,\Gamma}(x,\xi)^{\frac{n}{2}}}\frac{|\xi_1\ldots\xi_n|^{\alpha}}{\mathcal{Q}_{n,\Gamma}(x,\xi)^{\frac{\alpha n}{2}}}\frac{l(x,\xi)^{2k}}{\mathcal{Q}_{n,\Gamma}(x,\xi)^k}. \quad (106)$$

One can easily notice that this expansion can be obtained by expanding cosh in Formula (103). As in the Section 4.3 on majorant calculation, we can bound the following relation, using (56) and (58):

$$0 \le \frac{|\xi_1\ldots\xi_n|}{\mathcal{Q}_{n,\Gamma}(x,\xi)^{\frac{n}{2}}} \le 2^{n/2}G(0)^{n/2},$$

for all $x_a$. Now we can specify the function $t$:

$$t = \frac{1}{2^{n/2}G(0)^{n/2}}\frac{|\xi_1\ldots\xi_n|}{\mathcal{Q}_{n,\Gamma}(x,\xi)^{\frac{n}{2}}}. \quad (107)$$

Having defined an appropriate function $t$, consider finite-degree even polynomial approximation (98) of the function $f$. Let us note the following: Expression (98) describes a general polynomial approximation such that the polynomials $h_N$ are defined for all integer $N > 0$ and $\deg h_N = 2N$. We will often use the same notations, as for Legendre polynomials, but for generality and simplification of formulas, we will not specify the concrete form of (real) coefficients $u_i$ and $c_q$. Here, $\{P_{2q}(t)\}_{q=0}^{\infty}$ is some general set of polynomials suitable for pointwise approximation of the function $f$, in other words, $h_N(t) \to f(t)$ for all $t \in [-1,1]$. We will also suppose for generality, that the coefficients $c_q(f)$ can also depend on $N$, which is actual for approximation with Bernstein polynomials.

In the following we will denote as $I_{n,N}$ the integrals obtained from $I_n$ by substitution polynomial approximation $h_N(t)$ instead of $f(t)$ in the integrand. Let us note that $I_{n,N} \to I_n$ due to the DCT, when $N \to \infty$. This fact follows from the uniform convergence of Legendre series to the considered function (as discussed in Section 5.2), which means the uniform boundedness of partial sums. This convergence will also be checked numerically in Section 5.2.4.

Thus we arrive at the following expression:

$$I_{n,N} = \frac{1}{2}\sum_{k=0}^{\infty}\frac{(2G(0))^{n\alpha/2}}{4^k k!}\frac{\left(\frac{n(\alpha+1)}{2}\right)_{2k}}{(1/2)_{2k}}\Gamma\left(\frac{n(\alpha+1)}{2}\right)\sum_{q=0}^{N}c_q(f)$$

$$\times \int_{S^{n-1}}\frac{d\Omega}{\mathcal{Q}_{n,\Gamma}(x,\xi)^{\frac{n}{2}}}P_{2q}\left(\frac{|\xi_1\ldots\xi_n|}{2^{n/2}G(0)^{n/2}\mathcal{Q}_{n,\Gamma}(x,\xi)^{\frac{n}{2}}}\right)\frac{l(x,\xi)^{2k}}{\mathcal{Q}_{n,\Gamma}(x,\xi)^k}.$$

Recalling the identity for $n, s > 0$:

$$\int_0^{\infty}dr\,e^{-r^2}r^{n+2s-1} = \frac{1}{2}\Gamma\left(\frac{n}{2}+s\right),$$

we can return to $\mathbb{R}^n$ due to homogeneity:

$$I_{n,N} = \sum_{k=0}^{\infty}\frac{(2G(0))^{n\alpha/2}}{4^k k!}\frac{\left(\frac{n(\alpha+1)}{2}\right)_{2k}}{(1/2)_{2k}}\sum_{q=0}^{N}\sum_{i=0}^{q}\frac{u_i c_q(f)\Gamma\left(\frac{n(\alpha+1)}{2}\right)}{2^{ni}G(0)^{ni}\Gamma\left(\frac{n}{2}+ni+k\right)}$$

$$\times \int_{\mathbb{R}^n}d\phi_1\ldots d\phi_n\,(\phi_1\ldots\phi_n)^{2i}l(x,\phi)^{2k}e^{-\mathcal{Q}_{n,\Gamma}(x,\phi)}.$$

It is worth noting that without the ratio $\Gamma\left(\frac{n(\alpha+1)}{2}\right)/\Gamma\left(\frac{n}{2}+ni+k\right)$ we would receive cosh after summation instead of confluent hypergeometric function $_1F_1$. At the same time,

we do not write out the last one, since it is convenient to continue the transformations of the series itself.

We can rewrite the integrand, using the multinomial formula (all the $\beta_a \geq 0$):

$$l(x,\phi)^{2k} = \sum_{\beta_1+\ldots+\beta_n=2k} \binom{2k}{\beta_1 \ldots \beta_n} \chi_1^{\beta_1} \cdots \chi_n^{\beta_n} \phi_1^{\beta_1} \cdots \phi_n^{\beta_n},$$

and proceed with the integrals of the form (which are no more than the moments of multidimensional Gaussian distribution with covariance matrix $G_{n,\Gamma}$):

$$\int_{\mathbb{R}^n} d\phi_1 \ldots d\phi_n \, \phi_1^{m_1} \cdots \phi_n^{m_n} e^{-\mathcal{Q}_{n,\Gamma}(x,\phi)}, \tag{108}$$

for some integer constants $m_a \geq 0$. There is a common method for their calculation consisting in introducing auxiliary variables $\eta_a$, called "sources", in similar fashion to path integrals, and further differentiation over them. Namely, we use the identity, following from DCT:

$$\int_{\mathbb{R}^n} d\phi_1 \ldots d\phi_n \, \phi_1^{m_1} \cdots \phi_n^{m_n} e^{-\mathcal{Q}_{n,\Gamma}(x,\phi)+\sum_{a=1}^{n} \eta_a \phi_a} = \partial_1^{m_1} \ldots \partial_n^{m_n} e^{\frac{1}{2}\sum_{a,b=1}^{n}(G_{n,\Gamma})_{ab}\eta_a\eta_b}, \tag{109}$$

where the partial derivatives $\partial_a := \frac{\partial}{\partial \eta_a}$. If all $\eta_a = 0$, we obtain the desired equality.

Keeping the previous expression in mind, we obtain:

$$I_{n,N} = \sum_{k=0}^{\infty} \frac{(2G(0))^{n\alpha/2}}{4^k k!} \frac{\left(\frac{n(\alpha+1)}{2}\right)_{2k}}{(1/2)_{2k}} \sum_{q=0}^{N} \sum_{i=0}^{q} \sum_{\beta_1+\ldots+\beta_n=2k} \binom{2k}{\beta_1 \ldots \beta_n} u_i c_q(f)$$

$$\times \frac{\Gamma\left(\frac{n(\alpha+1)}{2}\right)\sqrt{(2\pi)^n \det(G_{n,\Gamma})}}{2^{ni} G(0)^{ni} \Gamma\left(\frac{n}{2}+ni+k\right)} \chi_1^{\beta_1} \cdots \chi_n^{\beta_n} \partial_1^{\beta_1+2i} \ldots \partial_n^{\beta_n+2i}\Bigg|_{\eta_a=0} e^{\frac{1}{2}\sum_{a,b=1}^{n}(G_{n,\Gamma})_{ab}\eta_a\eta_b}. \tag{110}$$

Finally, for the connected Green functions GF $n$th term approximation $\mathcal{G}_{I,n}[j]_N$ we obtain the following expression:

$$\mathcal{G}_{I,\varepsilon,n}[j]_N = \frac{(-1)^n}{n!} \sum_{\Gamma \in \mathbb{G}_{C,n}} \left\{\prod_{a<b}^{n} \int_0^1 ds_{ab} \, \partial_{s_{ab}}^{\nu_{ab}(\Gamma)}\right\} \left\{\prod_{a=1}^{n} \int_{\mathbb{R}^d} dx_a \, g(x_a)\right\} e^{-\mathcal{Q}_{n,\Gamma}(x,\varphi)}$$

$$\times \sum_{k=0}^{\infty} \frac{(2G(0))^{n\alpha/2}}{4^k k!} \frac{\left(\frac{n(\alpha+1)}{2}\right)_{2k}}{(1/2)_{2k}} \sum_{\beta_1+\ldots+\beta_n=2k} \binom{2k}{\beta_1 \ldots \beta_n} \chi_1^{\beta_1} \cdots \chi_n^{\beta_n} \sum_{q=0}^{N} \sum_{i=0}^{q} u_i c_q(f) \tag{111}$$

$$\times \frac{\Gamma\left(\frac{n(\alpha+1)}{2}\right)}{2^{ni} G(0)^{ni} \Gamma\left(\frac{n}{2}+ni+k\right)} \partial_1^{\beta_1+2i} \ldots \partial_n^{\beta_n+2i}\Bigg|_{\eta_a=0} e^{\frac{1}{2}\sum_{a,b=1}^{n}(G_{n,\Gamma})_{ab}\eta_a\eta_b}.$$

Starting from this formula and below, we found its useful to introduce the notation $\mathcal{G}_{I,\varepsilon,n}[j]_N$. We specify the value $N$ to the right of $j$, so as not to clutter up the number of indices to the left of $j$. By definition, $\mathcal{G}_{I,\varepsilon,n}[j]_N$ is obtained from $\mathcal{G}_{I,\varepsilon,n}[j]$ by substituting the approximation $I_{n,N}$ instead of $I_n$. Owing to the convergence $I_{n,N} \to I_n$, when $N \to \infty$, it is also true that $\mathcal{G}_{I,\varepsilon,n}[j]_N \to \mathcal{G}_{I,\varepsilon,n}[j]$. Recall that in Expression (111) $s_{ab}\nu_{ab}$ and $\varepsilon$ enter in three ways:

1. $\mathcal{Q}_{n,\Gamma}(x,\varphi)$ in the exponent;
2. $\chi_a$ in the prefactor;
3. $(G_{n,\Gamma})_{ab}$ in the exponent.

It is interesting that in fact Formula (111) provides the expansion of the connected Green functions GF for fractional-power interaction theory in terms of contributions of power-interaction theories with some "weights". However, it is not the same if one would

like to expand $|\phi|^{\alpha}$ in the action (1) itself. In such an approach there would be no "damping factor" of the form $\Gamma\left(\frac{n(\alpha+1)}{2}\right)/\Gamma\left(\frac{n}{2}+ni+k\right)$, and the series would be divergent due to its absence. The reason for this factor to appear is that before the approximation we reduced the integral from non-compact $\mathbb{R}^n$ to a compact $S^{n-1}$. As a result, we will not lose the convergence on any step and therefore obtain the convergent series rather than asymptotic. This achievement was the primary aim of the present paper.

5.2.3. Combinatorial Expansion

We want to derive analytical formulas for the quantities:

$$
\partial_1^{m_1}\ldots\partial_n^{m_n}\Big|_{\eta_a=0} e^{\frac{1}{2}\sum\limits_{a,b=1}^{n}(G_{n,\Gamma})_{ab}\eta_a\eta_b}, \tag{112}
$$

for any integer $m_a \geq 0$, for $a = 1, \ldots, n$. The authors are sure that such formulas were derived a long time ago, but we could not find them in the literature in a suitable form. So let us derive them from the basics for our own usage. In fact, the formulas we will obtain are nothing else but the general formulas for the symmetry coefficients (up to some factor) of Feynman diagrams.

As already mentioned about Expression (109), in this expression it is not hard to recognize the moments of $n$-dimensional Gaussian distribution with the covariance matrix $G_{n,\Gamma}$, which we will denote by brackets of "averaging":

$$
\begin{aligned}
\langle\phi_1^{m_1}\ldots\phi_n^{m_n}\rangle_{G_{n,\Gamma}} &:= \left\{\int_{\mathbb{R}^n}\frac{d\phi_1\ldots d\phi_n}{\sqrt{(2\pi)^n\det(G_{n,\Gamma})}}\,\phi_1^{m_1}\ldots\phi_n^{m_n}\,e^{-\mathcal{Q}_{n,\Gamma}(x,\phi)+\sum\limits_{a=1}^{n}\eta_a\phi_a}\right\}\Bigg|_{\eta_a=0} \\
&= \partial_1^{m_1}\ldots\partial_n^{m_n}\Big|_{\eta_a=0}e^{\frac{1}{2}\sum\limits_{a,b=1}^{n}(G_{n,\Gamma})_{ab}\eta_a\eta_b}.
\end{aligned} \tag{113}
$$

The index of the angle brackets indicates the covariance matrix we use for the computation of this quantity. We will refer to the quantity (113) as a correlator or multidimensional Gaussian moment with integer powers. We have already introduced another notion of a correlator in the Section 3 (referring to the path integral), but it will be clear from the context what notion we mean in every particular case. We will also refer to the variables $\phi_a$ in correlators as fields.

For now, we will consider covariance matrix $G_n$ rather than $G_{n,\Gamma}$ for the simplification of notations (in this subsection we will also omit the index $n$ for the same reason). Though, we will not suppose any of its special properties except it is symmetric and has equal terms on the diagonal which we will denote as $G_{aa} = G(0)$ (a generalization without this property is straightforward). Then the result for $G_{n,\Gamma}$ can be obtained with the simple substitution $G_{ab} \mapsto (G_{n,\Gamma})_{ab}$.

There is the Isserlis–Wick theorem which claims that the multidimensional Gaussian moment with integer powers can be expressed in terms of the covariance matrix elements as the sum over all possible pairings:

$$
\partial_{i_1}\ldots\partial_{i_m}\Big|_{\eta_a=0}e^{\frac{1}{2}\sum\limits_{a,b=1}^{n}G_{ab}\eta_a\eta_b} = \sum_{\text{all pairings of }\{i_a\}}G_{i_{a_1}i_{a_2}}\ldots G_{i_{a_{m-1}}i_{a_m}}, \tag{114}
$$

for even $m$ and are equal to zero for odd $m$. Let us note that the notations in (113) and (114) differ by rearranging of derivations and $m = \sum\limits_{a=1}^{n} m_a$ is a total their number.

Now we are going to explain what do we mean under this summation. Informally, we sum over all pairings of the indices set, with account of possible repetitions of $i_a$. More

formally, consider even $m$ and denote the set of all possible pairings of $\{1, \ldots, m\}$ as $\mathcal{P}_m$. For example, for $m = 4$ the set of all possible pairings reads as follows:

$$\mathcal{P}_4 = \{ \{\{1, 2\}, \{3, 4\}\}, \{\{1, 3\}, \{2, 4\}\}, \{\{1, 4\}, \{2, 3\}\}\} \}.$$

We will denote the general element of $\mathcal{P}_m$ as $\{ \{a_1, a_2\}, \ldots \{a_{m-1}, a_m\} \}$. These indices are exactly ones from Formula (114). So one can determine the set of pairings with the array $\{a_k\}_{k=1}^m$, and we will use this array to determine the indices of $i_a$.

Now we are going to calculate the number of all equal terms in the sum over all pairings in Expression (114) assuming, that $i_1, \ldots, i_{m_1} = 1$; $i_{m_1+1}, \ldots, i_{m_2} = 2$; $\ldots$ and $i_{m_{n-1}+1}, \ldots, i_{m_n} = n$, which corresponds to our object of interest (113).

We start from the case of $n = 2$, so the index $a$ takes only values 1 and 2. Then there are only two different types of terms $G_{ij}$:

1. when $i = j$, then $G_{ij} = G(0)$;
2. when $i \neq j$, then $G_{ij} = G_{12}$.

As a result, every term in (114) has the form $G(0)^q (G_{12})^p$ for some integers $p, q \geq 0$. So we have to count the number of pairings $\{a_k\}_{k=1}^m$, corresponding to every possible $p$ and $q$. Suppose we consider $\langle \phi_1^{m_1} \phi_2^{m_2} \rangle_G$ for $m_1 > 0$ and $m_2 > 0$. From the conservation of the total points number one can deduce that $m = m_1 + m_2 = 2q + 2p$. So, for given $m_1$ and $m_2$ the number $q$ is uniquely determined with $p$.

We receive the combinatorial factor from the following considerations: there are $m_1$ fields with index 1 and $m_2$ fields with index 2. It is possible to visualize them with Figure 1. We will denote all the enterings of the fields with the points, and for every pairing we will draw the segment ending in the points constituting a pair. We will also refer to these segments as edges, being inspired by the graph theory. In the following we will mean under the notion "configuration of pairings" the set of all pairings which give the equal contributions in the sum (114). In fact, this is a set of pairings (subset of $\mathcal{P}_m$), consisting of $\{ \{a_1, a_2\}, \ldots \{a_{m-1}, a_m\} \}$ for some $a_k$ which possess the same number of pairings between different clusters of points and inside every cluster. For $n = 2$ it is equivalent that they have the same $p$, but the number of "degrees of freedom" $n(n-1)/2$ rises for greater $n$. One can draw an analogy with statistical physics: we identify the microstate with the particular set $\{ \{a_1, a_2\}, \ldots \{a_{m-1}, a_m\} \}$ and macrostate is what we call the configuration of pairs.

Then we have $m_1 + m_2$ points in total, and they all have to be connected by such segments in pairs. These points can be naturally distributed between two clusters, corresponding to the indices of the fields. For the convenience, we also introduce the notations $l_a := m_a - p$ for the number of points which will be paired with the points from the same cluster. From the Isserlis–Wick theorem (114) it follows, in particular, that for odd $m_1 + m_2$ the considering correlator is zero, so we will consider only even case. So, the term $G(0)^q (G_{12})^p$ corresponds to $p$ segments with the ends in different clusters, and $q$ segments, which have ends in the same cluster. To draw all the segments which such conditions, one must:

1. choose $p$ points from every cluster which will give rise to the segments with ends in the different clusters;
2. choose the way the lines are drawn between the chosen $p$ points in every cluster;
3. choose the way the remaining $m_1 - p$ and $m_2 - p$ points in first and second clusters correspondingly will be connected with the segments inside their clusters.

The first number is $\binom{m_1}{p}\binom{m_2}{p}$ by combinatorial definition, according to the combinatorial rule of product, where $\binom{n}{k} = \frac{n!}{k!(n-k)!}$ is a number of combinations of $k$ elements from a subset of cardinality $n$. The second one is $p!$, since for the first point (for any fixed enumeration) in the first cluster we have $p$ variants of choosing a pair, for the second one—$(p-1)$, and so on. The third number equals to $\frac{l_a!}{2^{l_a/2}(l_a/2)!}$ for each of two clusters, so in total these two contributions should be multiplied. The explanation for the last formula is that we

have to choose firstly one pair, which is possible in $l_a(l_a - 1)/2$ ways, then the another one, for which we have $(l_a - 2)(l_a - 3)/2$ ways, and so on. Finally, we have to divide the product of such ways' numbers by the permutation number of points' pairs, which is $(l_a/2)!$. We also note that the numbers $l_a$ have to be even for any possible configuration, and the number of pairs inside the cluster is $l_a/2$.

In total for the number $M_p$ of pairings with contribution $G(0)^q(G_{12})^p$, using the same combinatorial rule of product, we have the following expression:

$$M_p = \binom{m_1}{p}\binom{m_2}{p} \cdot p! \cdot \frac{l_1!}{2^{l_1/2}(l_1/2)!}\frac{l_2!}{2^{l_2/2}(l_2/2)!} = \frac{m_1!m_2!}{2^{l_1/2+l_2/2}p!(l_1/2)!(l_2/2)!}. \tag{115}$$

We have written this formula in such notations to make its generalisation for $n > 2$ more clear.

At this moment, for the two-dimensional Gaussian moment with integer powers $m_1$ and $m_2$, the expression reads:

$$\langle \phi_1^{m_1}\phi_2^{m_2}\rangle_G = \sum_{p=0}^{\min\{m_1,m_2\}} \frac{m_1!m_2!}{2^{\frac{m_1+m_2}{2}-p}p!\left(\frac{m_1-p}{2}\right)!\left(\frac{m_2-p}{2}\right)!}(G_{12})^p G(0)^{\frac{m_1+m_2}{2}-p}, \tag{116}$$

where the summation is carried out over all $p$, such that $l_a = m_a - p$ are even, since only these configurations of pairings are possible. For the considering case, when $m_1 + m_2$ is even, the two cases are realisable: $m_1$ and $m_2$ are both even or both odd. In the first case the summation condition means that the summation is carried out over only even $p$, and in the second—over only odd $p$. The sense of these facts becomes clear from Figure 1.

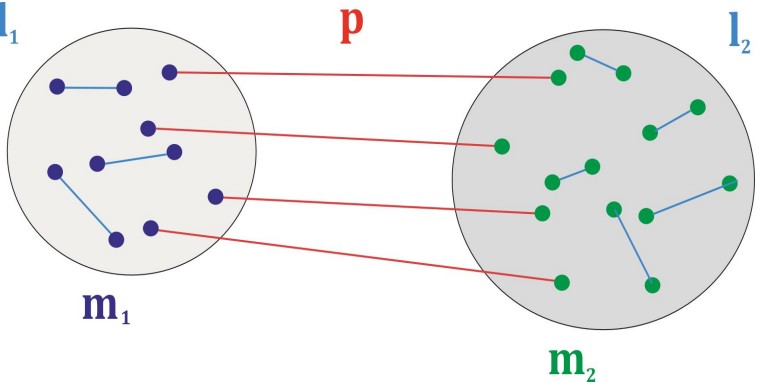

**Figure 1.** Illustration of combinatorial formula for the two-dimensional Gaussian moment with integer powers $m_1$ and $m_2$.

Let us generalize the resulting formulas to the $n$-dimensional case. Acting in the same spirit, one can verify the validity of the following expression:

$$\langle \phi_1^{m_1}\dots\phi_n^{m_n}\rangle_G = \sum_{\{l_{ab}\}} 2^{-\sum_{a=1}^{n}\frac{l_{aa}}{2}} \frac{\prod_{a=1}^{n}(m_a)!}{\prod_{a<b}^{n}(l_{ab}!)\prod_{a=1}^{n}\left(\frac{l_{aa}}{2}\right)!}\prod_{a<b}^{n}(G_{ab})^{l_{ab}}G(0)^{\frac{1}{2}\sum_{a=1}^{n}l_{aa}}. \tag{117}$$

In accordance with the notations, introduced in the Section 2, the summation in the Expression (117) is carried out over all $l_{ab} \geq 0$, satisfying the conditions $\sum_{b=1}^{n}l_{ab} = m_a$ (where $l_{ab} = l_{ba}$) and $l_{aa}$ is even. This expression is non-zero for even $\sum_{a=1}^{n}m_a$ and zero for odd. The Formula (117) generalises (116) and we will use it in our further calculations.

Let us derive Expression (117). It can be performed in the same manner as for $n = 2$, using the same graphical interpretation. We start from introducing slightly different notations.

In the general case we have $n$ clusters, consisting of $m_1, \ldots, m_n$ points, correspondingly. At present, the contribution to the sum in (114) of a given pairings configuration is prescribed with the numbers $l_{ab}$ (indices $a, b \in \{1, \ldots, n\}$):

1. the numbers $l_{ab}$ ($a \neq b$) of edges connecting $a$th and $b$th clusters;
2. the numbers $l_{aa}$ of edges connecting the vertices inside the $a$th cluster.

We will suppose that in the notation $l_{ab}$ always $a \leq b$, but for the clarity of formulas (summation conditions in (117)) we will imply $l_{ab} = l_{ba}$ each time we use both orders of indices, remembering that there are only $n(n-1)/2$ such numbers in fact. Each configuration of pairings gives the contribution $\prod\limits_{a<b}^{n} (G_{ab})^{l_{ab}} G(0)^{\frac{1}{2} \sum\limits_{a=1}^{n} l_{aa}}$. The numbers $l_{ab}$ are visualised in Figure 2.

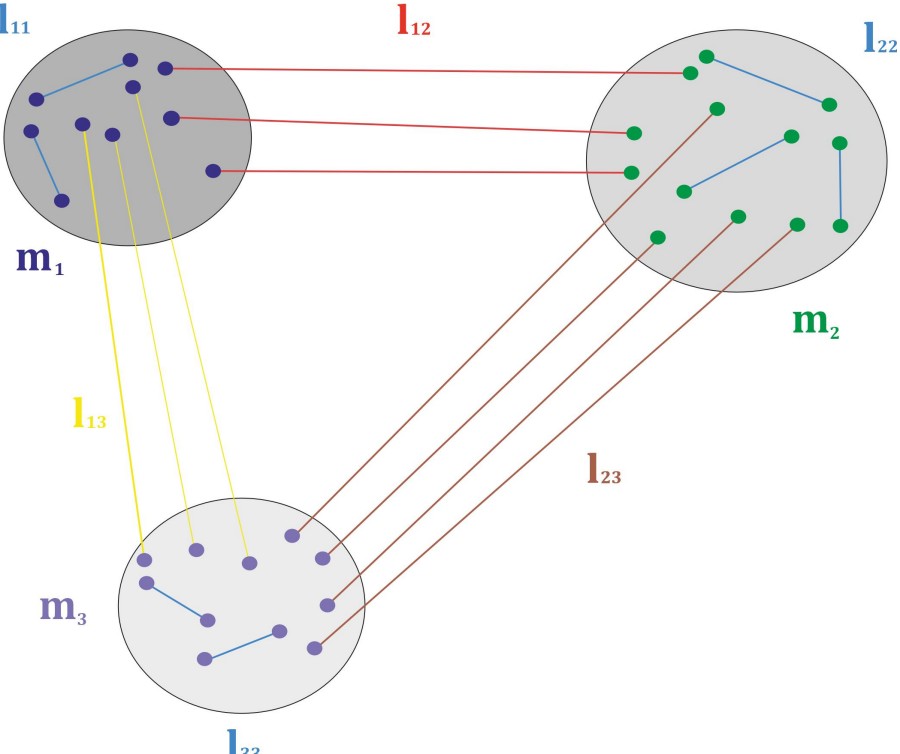

**Figure 2.** Illustration of combinatorial formula for the three-dimensional Gaussian moment with integer powers $m_1$, $m_2$ and $m_3$.

Further, it is necessary to count how many pairings configurations are described with the set of numbers $l_{ab}$ for $a < b$ and $l_{aa}$. The logic is similar to the case $n = 2$. We have to multiply:

1. the number of ways to choose from every $a$th cluster $l_{ab}$ points going to $b$th cluster for all $b$. This is a typical problem of multiple choices from $m_a$ objects firstly $l_{a1}$, then $l_{a2}$ and so on until $l_{an}$ objects (including $l_{aa}$ objects). As a result, this number equals the multinomial coefficient $\binom{m_a}{l_{a1} \, \ldots \, l_{an}}$;
2. for every $a$th cluster, the number of ways to form pairs inside of it from the remaining $l_{aa}$ points, which is $\frac{l_{aa}!}{2^{l_{aa}/2}(l_{aa}/2)!}$, with the same explanation as for $n = 2$;
3. for every pair of clusters, $a$th and $b$th, the number of ways to draw the edges from $l_{ab}$ chosen points in $a$th cluster to $l_{ab}$ chosen points in $b$th cluster. As a result, this number equals to $l_{ab}!$.

Multiplying all the described factors due to combinatorial rule of product, for the total number of ways $M_{\{l_{ab}\}}$ to realise the configuration of pairings with numbers $l_{ab}$ for $a < b$ and $l_{aa}$, we obtain the following result:

$$M_{\{l_{ab}\}} = \prod_{a=1}^{n} \binom{m_a}{l_{a1} \ldots l_{an}} \prod_{a=1}^{n} \frac{l_{aa}!}{2^{l_{aa}/2}(l_{aa}/2)!} \prod_{a<b}^{n} (l_{ab})! . \tag{118}$$

Recalling the formula for multinomial coefficients through the factorials, we arrive at the following expression for $M_{\{l_{ab}\}}$:

$$M_{\{l_{ab}\}} = 2^{-\sum_{a=1}^{n} \frac{l_{aa}}{2}} \frac{\prod_{a=1}^{n} (m_a)!}{\prod_{a<b}^{n} (l_{ab}!) \prod_{a=1}^{n} \left(\frac{l_{aa}}{2}\right)!} . \tag{119}$$

Now let us write the expression for the $n$-dimensional Gaussian moment in terms of $M_{\{l_{ab}\}}$:

$$\langle \phi_1^{m_1} \ldots \phi_n^{m_n} \rangle_G = \sum_{\{l_{ab}\}} \prod_{a<b}^{n} (G_{ab})^{l_{ab}} G(0)^{\frac{1}{2} \sum_{a=1}^{n} l_{aa}} M_{\{l_{ab}\}} . \tag{120}$$

Expression (120) gives us exactly the result (117) after substituting the explicit form of $M_{\{l_{ab}\}}$. Let us note again, that all $l_{aa}$ have to be divisible by 2, otherwise such a configuration is impossible. This condition is reflected in the summation condition in (117).

5.2.4. Numerical Analysis of the Polynomial Approximations

In this subsection, we are going to use different families of polynomials for the approximation and compare the results. Let us compare pointwise approximations of $|t|^\alpha$ by Bernstein, Chebyshev and Legendre polynomials with each other. The results are represented in Figure 3 for $\alpha = 4/3$. For other $\alpha \in (1, 2)$ there is no significant difference. From Figure 3 one can draw a conclusion that Legendre and Chebyshev polynomials are better for the approximation of integrands than Bernstein ones, since for the same degree $2N$ in the "majority of points" they give a smaller error. Moreover, errors of Legendre and Chebyshev polynomial approximations oscillate and change sign, rather than Bernstein polynomial approximation, so one should expect that there will be some "cancellation of errors" in integrals, which will additionally raise the precision of the approximation.

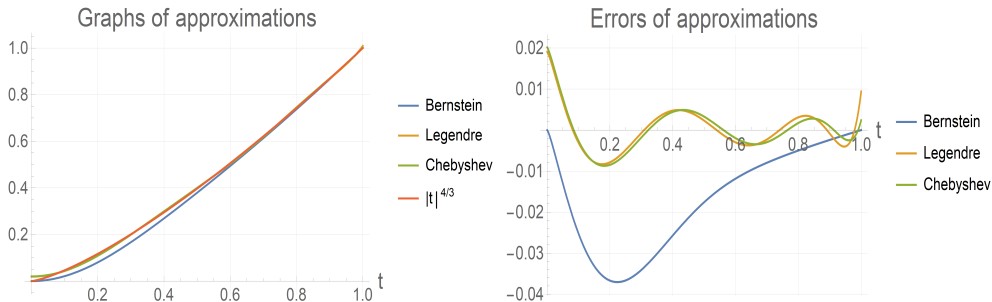

**Figure 3.** Comparative plots of approximations $h_N$ (98) by polynomials of the tenth degree ($N = 5$).

Now we are going to compare Bernstein, Chebyshev and Legendre polynomial approximations for integrals. We would like to simplify the general Formula (111), since we do not want to waste too many resources for choosing the best variant of the listed three. Namely, we will:

1.  assume that the source $j = 0$ to escape additional parameter, as well as $\varepsilon = 0$, since it is possible to remove this regulator for the zero source;

2. replace all the graphs $\Gamma$ in the sum inside the operator $\mathcal{O}$ (80) with the complete graph $K_n$, which is the graph without loops, where every two vertices are connected with an edge. This means that $\nu_{ab}(K_n) = (J_n)_{ab} - \delta_{ab}$ (the matrix $J_n$ is the matrix of ones) for all $a, b \in \{1, \ldots, n\}$, and the sum over graphs transforms into the contribution of the $K_n$, multiplied by the total number of connected graphs on $n$ vertices, which we will denote by $|\mathbb{G}_{C,n}|$ (the explicit value of this number in terms of $n$ is not needed);

3. assume that $G(x) \equiv G(0) = $ const for all $x$. From this follows that $G_n$ becomes proportional to a matrix of ones, i.e., $G_n = G(0)J_n$. Moreover, $(G_{n,K_n})_{aa} = G(0)$ and $(G_{n,K_n})_{ab} = s_{ab}G(0)$ for $a \neq b$.

This gives us the following rough approximation of connected Green functions GF, keeping in mind the definition (113):

$$
\mathcal{G}_{I,n}[0]_N \approx \frac{(-1)^n (2G(0))^{n\alpha/2}}{n!} |\mathbb{G}_{C,n}| \left\{ \prod_{a<b}^n \int_0^1 ds_{ab}\, \partial_{s_{ab}} \right\} \left\{ \prod_{a=1}^n \int_{\mathbb{R}^d} dx_a\, g(x_a) \right\}
$$
$$
\times \sum_{q=0}^N \sum_{i=0}^q \frac{u_i c_q(f)}{2^{ni} G(0)^{ni}} \frac{\Gamma\left(\frac{n(\alpha+1)}{2}\right)}{\Gamma\left(\frac{n}{2} + ni\right)} \left\langle \phi_1^{2i} \cdots \phi_n^{2i} \right\rangle_{G_{n,K_n}}. \tag{121}
$$

Now we can calculate coordinate integrals, since all the dependence of $x_a$ is left only in coupling constants $g(x_a)$ as well as the result of the integro-differential operator over the variables $s_{ab}$ action. To that end, we should substitute the combinatorial Formula (117). We arrive at the following expression:

$$
\left\{ \prod_{a<b}^n \int_0^1 ds_{ab}\, \partial_{s_{ab}} \right\} \left\langle \phi_1^{2i} \cdots \phi_n^{2i} \right\rangle_{\frac{G_{n,K_n}}{G(0)}} = \sum_{\{l_{ab}\}} \frac{(2i)!^n\, 2^{-\sum_{a=1}^n \frac{l_{aa}}{2}}}{\prod_{a<b}^n (l_{ab}!) \prod_{a=1}^n \left(\frac{l_{aa}}{2}\right)!} \left\{ \prod_{a<b}^n \int_0^1 ds_{ab}\, \partial_{s_{ab}} s_{ab}^{l_{ab}} \right\}, \tag{122}
$$

and the last factor gives nothing else but the additional restriction of the summation condition $l_{ab} > 0$ for $a < b$. However, we ignore this condition, despite the fact that this will lead to a worse approximation, since this will also lead to a decrease in the resources expended. Folding combinatorial sum back into correlator for the dimensionless covariance matrix (which is exactly the matrix of ones $J_n$), we obtain the approximation:

$$
\mathcal{G}_{I,n}[0]_N \approx \frac{(-1)^n (2G(0))^{n\alpha/2}}{n!} |\mathbb{G}_{C,n}| g\left(\mathbb{R}^d\right)^n \sum_{q=0}^N \sum_{i=0}^q \frac{u_i c_q(f)}{2^{ni+1}} \frac{\Gamma\left(\frac{n(\alpha+1)}{2}\right)}{\Gamma\left(\frac{n}{2} + ni\right)} \left\langle \phi_1^{2i} \cdots \phi_n^{2i} \right\rangle_{J_n}. \tag{123}
$$

From Formula (123), it is clear that for the comparison of different approximations it is useful to consider the following quantities $H(n, N)$:

$$
H(n, N) = \frac{(-1)^n}{(2G(0))^{n\alpha/2} |\mathbb{G}_{C,n}| g\left(\mathbb{R}^d\right)^n} \mathcal{G}_{I,n}[0]_N, \tag{124}
$$

where we have removed the common (depending on $n$ only) factors for all polynomial approximations from (123), but left the factor $1/n!$ to avoid factorial growth. Recall that the degree of polynomial approximation in all the cases is equal to $2N$. So in fact all the parameters of connected Green functions GF polynomial approximations are roughly encoded in function $H$, the explicit formula for which is:

$$
H(n, N) = \sum_{q=0}^N \sum_{i=0}^q \frac{u_i c_q(f)}{2^{ni}} \frac{\Gamma\left(\frac{n(\alpha+1)}{2}\right)}{\Gamma\left(\frac{n}{2} + ni\right)} \left\langle \phi_1^{2i} \cdots \phi_n^{2i} \right\rangle_{J_n}. \tag{125}
$$

Further, for different kinds of polynomial approximations there will be different coefficients $u_i$ and $c_q(f)$. In the following, we will mark the family of polynomials we use for the approximation in the index of $H$. Thus, we have:

$$H_L(n,N) := \sum_{q=0}^{N} \sum_{i=0}^{q} \left(2q+\frac{1}{2}\right) \sum_{p=0}^{q} \binom{2q}{2p} \binom{q+p-\frac{1}{2}}{2q} \frac{2^{4q-ni+1}}{2p+\alpha+1}$$

$$\times \binom{2q}{2i} \binom{q+i-\frac{1}{2}}{2q} \frac{\Gamma\left(\frac{n(\alpha+1)}{2}\right)}{G(0)^{ni}\Gamma\left(\frac{n}{2}+ni\right)n!} \left\langle \phi_1^{2i} \cdots \phi_n^{2i} \right\rangle_{J_n},$$

for Legendre polynomial approximation, and:

$$H_B(n,N) := \sum_{q=0}^{N} \sum_{i=q}^{N} \left(\frac{q}{N}\right)^{\alpha/2} \frac{(-1)^{i+q}N!}{q!(i-q)!(N-i)!} \frac{\Gamma\left(\frac{n(\alpha+1)}{2}\right)}{2^{ni}G(0)^{ni}\Gamma\left(\frac{n}{2}+ni\right)n!} \left\langle \phi_1^{2i} \cdots \phi_n^{2i} \right\rangle_{J_n},$$

for Bernstein polynomial approximation, as well as:

$$H_{Ch}(n,N) := \frac{\Gamma\left(\frac{\alpha}{2}+\frac{1}{2}\right)}{\Gamma\left(\frac{\alpha}{2}+1\right)} \frac{\Gamma\left(\frac{n(\alpha+1)}{2}\right)}{\Gamma\left(\frac{n}{2}\right)} + 2 \sum_{q=1}^{N} \sum_{i=0}^{q} \frac{(-1)^i q^2 \Gamma\left(\frac{n(\alpha+1)}{2}\right)}{2^{(n+2)i-4q}G(0)^{ni}n!\Gamma\left(\frac{n}{2}+ni\right)}$$

$$\times \frac{(2q-i-1)!}{i!(2q-2i)!} \left( \sum_{p=0}^{q} (-1)^p \frac{(2q-p-1)!}{p!(2q-2p)!} 2^{-2p} \frac{\Gamma\left(q-p+\frac{\alpha}{2}+\frac{1}{2}\right)}{\Gamma\left(q-p+\frac{\alpha}{2}+1\right)} \right) \left\langle \phi_1^{2i} \cdots \phi_n^{2i} \right\rangle_{J_n},$$

for Chebyshev polynomial approximation. The plots of $H_{L,B,Ch}(n,N)$ for $n=2,\ldots,5$ and $N=1,2,4$ and $\alpha=4/3$ are presented in Figure 4. The picture for other values of $\alpha \in (1,2)$ is qualitatively the same.

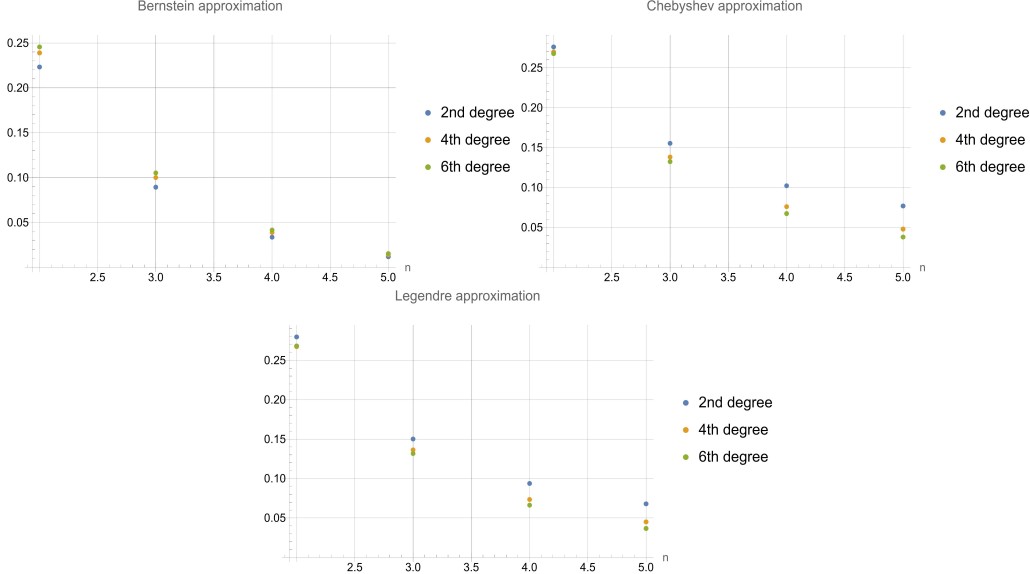

**Figure 4.** Comparative plots of approximations by polynomials of integrals $H_{L,B,Ch}(n,N)$.

From the plots one can see that Legendre and Chebyshev approximations for integrals also converge much better than Bernstein approximation. At the same time, formulas for approximation by Legendre polynomials are simpler than for approximation by Chebyshev polynomials. This, in particular, explains our choice of the Legendre polynomials. Let us also note that using the second-degree approximation gives an error which is about 20% and the error of fourth-degree is about 10%. However, it is important to note that this error

occurs in the model system considered in this subsection. As the results of the following subsections show, in a more realistic case, the errors can be larger.

5.2.5. Final Results of Polynomial Approximations for Connected Green Functions GF

According to all points described above, we will focus on Legendre polynomial approximation. Exactly this kind of approximation we will mean in the following under the term "polynomial approximation". We will write these expressions in compact forms using the already introduced notations as well as few new ones. Namely, we will additionally introduce the notations for the arguments of Gamma functions and a new operator $\mathcal{O}_g$:

$$\alpha_n := \frac{n(\alpha+1)}{2}, \quad n_i := \frac{n}{2} + i, \quad \mathcal{O}_g = \mathcal{O}\left\{\prod_{a=1}^{n} \int_{\mathbb{R}^d} dx_a \, g(x_a)\right\}. \tag{126}$$

Then Expression (111) will take a form:

$$\begin{aligned}
\mathcal{G}_{I,\varepsilon,n}[j]_N &= \frac{(-1)^n}{n!} \mathcal{O}_g \, e^{-\mathcal{Q}_{n,\Gamma}(x,\varphi)} \sum_{k=0}^{\infty} \frac{(2)^{n\alpha/2} G(0)^{2k+n\alpha/2}}{4^k k!} \frac{(\alpha_n)_{2k}}{(1/2)_{2k}} \sum_{q=0}^{N} \sum_{i=0}^{q} \frac{u_i c_q(f)}{2^{ni+1}} \\
&\times \sum_{\beta_1+\ldots+\beta_n=2k} \binom{2k}{\beta_1 \ldots \beta_n} \chi_1^{\beta_1} \cdots \chi_n^{\beta_n} \frac{\Gamma(\alpha_n)}{\Gamma(n_i+k)} \left\langle \phi_1^{2i+\beta_1} \cdots \phi_n^{2i+\beta_n} \right\rangle_{\frac{G_{n,\Gamma}}{G(0)}},
\end{aligned} \tag{127}$$

where we have rewritten correlators, using dimensionless covariance matrix $G_{n,\Gamma}/G(0)$ for the convenience. For the zero source $j$, similarly:

$$\mathcal{G}_{I,n}[0]_N = \frac{(-1)^n (2G(0))^{n\alpha/2}}{n!} \mathcal{O}_g \sum_{q=0}^{N} \sum_{i=0}^{q} \frac{u_i c_q(f)}{2^{ni+1}} \frac{\Gamma(\alpha_n)}{\Gamma(n_i)} \left\langle \phi_1^{2i} \cdots \phi_n^{2i} \right\rangle_{\frac{G_{n,\Gamma}}{G(0)}}. \tag{128}$$

In this form the structure of expressions is much more clear, so in the following we will use these formulas for representing connected Green functions GF.

For the reference we also write down the extended form of the Expressions (127) and (128). Substituting all the introduced notation, we receive very large formulas:

$$\begin{aligned}
\mathcal{G}_{I,\varepsilon,n}[j]_N &= \frac{(-1)^n}{n!} \sum_{\Gamma \in \mathbb{G}_{C,n}} \left\{\prod_{a<b}^{n} \int_0^1 ds_{ab} \, \partial_{s_{ab}}^{\nu_{ab}(\Gamma)}\right\} \left\{\prod_{a=1}^{n} \int_{\mathbb{R}^d} dx_a \, g(x_a)\right\} e^{-\mathcal{Q}_{n,\Gamma}(x,\varphi)} \\
&\times \sum_{k=0}^{\infty} \frac{(2G(0))^{n\alpha/2}}{4^k k!} \frac{\left(\frac{n(\alpha+1)}{2}\right)_{2k}}{(1/2)_{2k}} \sum_{\beta_1+\ldots+\beta_n=2k} \binom{2k}{\beta_1 \ldots \beta_n} \chi_1^{\beta_1} \cdots \chi_n^{\beta_n} \\
&\times \sum_{q=0}^{N} \sum_{i=0}^{q} \left(2q + \frac{1}{2}\right) 2^{4q-ni+1} G(0)^{2k} \frac{\Gamma\left(\frac{n(\alpha+1)}{2}\right)}{\Gamma\left(\frac{n}{2} + ni + k\right)} \\
&\times \sum_{p=0}^{q} \binom{2q}{2p} \binom{q+p-\frac{1}{2}}{2q} \frac{1}{2p+\alpha+1} \binom{2q}{2i} \binom{q+i-\frac{1}{2}}{2q} \\
&\times \sum_{\{l_{ab}\}} 2^{-\sum\limits_{a=1}^{n} \frac{l_{aa}}{2}} \frac{\prod\limits_{a=1}^{n} (2i+\beta_a)!}{\prod\limits_{a<b}^{n} (l_{ab}!) \prod\limits_{a=1}^{n} \left(\frac{l_{aa}}{2}\right)!} \prod_{a<b}^{n} \left(\frac{s_{ab} \nu_{ab}(\Gamma) G(x_a - x_b)}{G(0)}\right)^{l_{ab}}.
\end{aligned} \tag{129}$$

In accordance with the notations, introduced in the Section 2, the summation in the last line of Expression (129) is carried out over all $l_{ab} \geq 0$, satisfying the conditions $\sum\limits_{b=1}^{n} l_{ab} = 2i + \beta_a$ (where $l_{ab} = l_{ba}$) and $l_{aa}$ is even. Expression (129) is the main polynomial

approximation formula for the following computations. For $j = 0$ we can also remove the regulator $\varepsilon$ and write:

$$\mathcal{G}_{I,n}[0]_N = \frac{(-1)^n (2G(0))^{n\alpha/2}}{n!} \sum_{\Gamma \in \mathbb{G}_{C,n}} \left\{ \prod_{a<b}^n \int_0^1 ds_{ab}\, \partial_{s_{ab}}^{\nu_{ab}(\Gamma)} \right\} \left\{ \prod_{a=1}^n \int_{\mathbb{R}^d} dx_a\, g(x_a) \right\}$$

$$\times \sum_{q=0}^N \sum_{i=0}^q \binom{2q + \tfrac{1}{2}}{} \sum_{p=0}^q \binom{2q}{2p}\binom{q+p-\tfrac{1}{2}}{2q} \frac{2^{4q-ni+1}}{2p+\alpha+1} \frac{\Gamma\left(\frac{n(\alpha+1)}{2}\right)}{\Gamma\left(\frac{n}{2}+ni\right)} \binom{2q}{2i}\binom{q+i-\tfrac{1}{2}}{2q} \quad (130)$$

$$\times \sum_{\{l_{ab}\}} 2^{-\sum\limits_{a=1}^n \frac{l_{aa}}{2}} \frac{\prod\limits_{a=1}^n (2i)!}{\prod\limits_{a<b}^n (l_{ab}!) \prod\limits_{a=1}^n \left(\frac{l_{aa}}{2}\right)!} \prod_{a<b}^n \left( \frac{s_{ab}\nu_{ab}(\Gamma)G(x_a - x_b)}{G(0)} \right)^{l_{ab}}.$$

As it follows from the discussion in the Section 3, vacuum energy $\mathcal{E} = -\mathcal{G}[0]$. So in the following computations of vacuum energy density this formula will be our starting point.

Finally, let us note that due to the convergence of Legendre series we have obtained in fact the expression for $\mathcal{G}_{I,\varepsilon,n}[j]$ in terms of repeated series. In other words, in Formulas (127)–(130) one can proceed to the limit $N \to \infty$:

$$\mathcal{G}_{I,\varepsilon,n}[j] = \lim_{N\to\infty} \mathcal{G}_{I,\varepsilon,n}[j]_N. \quad (131)$$

Moreover, at least for $j = 0$ the sums over $n$ and $q$ can be permuted because of the absolute convergence of Legendre series. Hence, at least for $\mathcal{G}_I[0]$ we have obtained the expression in terms of double series which is very convenient for calculations. Therefore:

$$\mathcal{G}_I[0] = \sum_{n=1}^\infty \sum_{q=0}^\infty \sum_{i=0}^q \frac{(-1)^n (2G(0))^{n\alpha/2}}{n!} \frac{u_i c_q(f)}{2^{ni+1}} \frac{\Gamma(\alpha_n)}{\Gamma(n_i)} \mathcal{O}_g \left\langle \phi_1^{2i} \dots \phi_n^{2i} \right\rangle_{\frac{G_{n,\Gamma}}{G(0)}}, \quad (132)$$

and the same for $\mathcal{G}_{I,\varepsilon}[j]$. These formulas are the main result of the presented paper. In the next subsections we will rewrite the result of applying the operator $\mathcal{O}_g$ to the correlator to make this formula more similar to usual formulas from the Feynman diagram approach.

### 5.2.6. Two Types of Graphs in the Formulas for Connected Green Functions GF

To go further, let us provide an improvement of the graphical interpretation of the terms from the Isserlis–Wick theorem (114), developed in the Section 5.2.3. Our new graphical interpretation will be more convenient for our purposes as long as we deal with all the combinatorics.

We are going to consider the clusters (in terms of Section 5.2.3) as vertices, and to draw the edges (corresponding to pairings) between the clusters themselves (in particular, starting and ending clusters can coincide). Therefore, we are going to describe the particular pairings' configuration in terms of the unoriented graph with possible loops and multiple edges, possibly disconnected. It is illustrated in Figure 5, from which one can recognise the ordinary Feynman diagrammatics. Informally, we took the graphical interpretation from the Section 5.2.3 and "constricted" every cluster to a point. The pairings inside every cluster became loops on the corresponding vertices.

Considering a correlator $\left\langle \phi_1^{m_1} \dots \phi_n^{m_n} \right\rangle$ (where we will not specify the covariance matrix in the notation for the brevity) we will enumerate vertices with the indices of corresponding fields $\phi_a$ or with the coordinates $x_a$ (which come from the covariance matrix with the same indices). In such notations, the degree of a vertex (the number of edges that a given vertex belongs to) with the number $a$ equals to $m_a$ and $l_{ab}$ is the number of edges between the vertices $a$ and $b$. Let us note that the adjacency matrix of the described graph is $(l_{ab})_{a,b=1}^n$. In the following we will call these graphs as Isserlis–Wick graphs. In the literature they are also known as Feynman graphs.

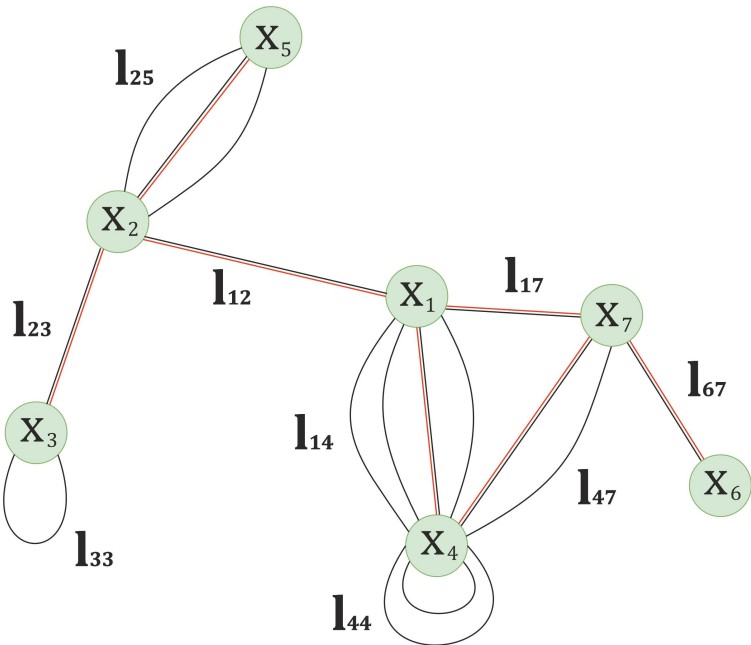

**Figure 5.** Illustration of typical Isserlis–Wick (black) and Meyer (red) graphs. The presented Isserlis–Wick graph refers to the correlator $\langle \phi_1^6 \phi_2^5 \phi_3^3 \phi_4^{10} \phi_5^3 \phi_6^1 \phi_7^4 \rangle$.

Therefore, in this paper we have two types of graphs:

1. Isserlis–Wick graphs, that are the subject of discussion in this subsection. We will denote them as $\Gamma_{IW}$;
2. Meyer graphs, which came from the Meyer cluster expansion in the Section 4.4, and still remain in the operator $\mathcal{O}$. We will denote them as $\Gamma_M$;

They are essentially different. While the Isserlis–Wick graphs can have loops, multiple edges or be disconnected, the Meyer graphs are the opposite; they have no loops, multiple edges and have to be connected. However, there is a link between these two types of graphs, explaining also why physicists have only one notion of Feynman diagrams, serving simultaneously for all the purposes. Namely, for the Meyer graphs that give a non-zero contribution to Expression (130) the following inclusion holds:

$$\Gamma_M \subset \Gamma_{IW}. \tag{133}$$

This inclusion is non-trivial for the sets of edges, since the sets of vertices are the same by definition. Let us prove this inclusion. Consider Formula (130). Next, consider concrete $\Gamma_M$. Let it contain an edge between vertices $a$ and $b$. In this case $v_{ab}(\Gamma_M) = 1$. Then differentiation with respect to $s_{ab}$ is non-trivial if and only if the corresponding value $l_{ab} > 0$. However, this means that there is such an edge in $\Gamma_{IW}$ defined by the given set $l_{ab}$. Repeating this argument for all the edges in $\Gamma_M$, we obtain the required inclusion.

In the same manner, if $v_{ab}(\Gamma_M) = 0$ for distinct $a$ and $b$, then every non-zero contribution has to be $l_{ab} = 0$ for the same $a$ and $b$. This means that if there is at least one edge in $\Gamma_{IW}$, then there is an edge in $\Gamma_M$.

The following useful statement is also true: the summation over Meyer and Isserlis–Wick graphs reduces to the summation over connected Isserlis–Wick graphs, at least for the zero source. Let us prove this statement. With the direct considering of cases, it can be shown that:

$$\int_0^1 ds\, \partial_s^v (sv)^l = v(1 - \delta_{l,0}) + (1 - v)\delta_{l,0} = \begin{cases} \delta_{l,0}, & v = 0; \\ 1 - \delta_{l,0}, & v = 1; \end{cases}, \tag{134}$$

for $v \in \{0, 1\}$ and $l \geq 0$. From Formula (134) it follows that we have the following product in Expression (130):

$$\left\{ \prod_{a<b}^{n} \int_0^1 ds_{ab} \, \partial_{s_{ab}}^{v_{ab}(\Gamma)} \right\} \prod_{a<b}^{n} (s_{ab} v_{ab}(\Gamma))^{l_{ab}} = \prod_{a<b}^{n} \left\{ v_{ab}(\Gamma)(1 - \delta_{l_{ab},0}) + (1 - v_{ab}(\Gamma))\delta_{l_{ab},0} \right\}. \quad (135)$$

However, for this product to be non-zero each of its terms have to be non-zero. For every set $\{l_{ab}\}$ as follows from (134) there is only one possible adjacency matrix $v_{ab}(\Gamma)$, satisfying this condition, and it has the following form:

$$v_{ab}(\Gamma) = \begin{cases} 0, & l_{ab} = 0; \\ 1, & l_{ab} > 0; \end{cases}, \quad (136)$$

for all nonequal $a$ and $b$. This expression can also be interpreted as the condition for Meyer graph to have the edge if and only if there exists at least one edge between these vertices in the Isserlis–Wick graph.

Therefore, for every $\{l_{ab}\}$ (every Isserlis–Wick graph) in the sum over connected (Meyer) graphs $\sum_{\Gamma \in \mathbb{G}_{C,n}}$, no more than one term stays alive, and it is completely determined with Condition (136). Though in the sum over the Meyer graphs there could be no such a graph, since Meyer graphs are always connected, but Condition (136) can be satisfied for disconnected graphs also. So the connectivity of the Isserlis–Wick graph is the necessary and sufficient condition for the term to give the non-zero contribution.

Summarising, the operator $\mathcal{O}$ leaves only connected Isserlis–Wick graphs, and we finish at the expression:

$$\mathcal{G}_{I,n}[0]_N = \frac{(-1)^n (2G(0))^{n\alpha/2}}{n!} \sum_{q=0}^{N} \sum_{i=0}^{q} \frac{u_i c_q(f)}{2^{ni+1}} \frac{\Gamma(\alpha_n)}{\Gamma(n_i)}$$
$$\times \left\{ \prod_{a=1}^{n} \int_{\mathbb{R}^d} dx_a \, g(x_a) \right\} \left\langle \phi_1^{2i} \dots \phi_n^{2i} \right\rangle_{\frac{G_{n,\Gamma}}{G(0)}}^C, \quad (137)$$

where the index "$C$" means that the summation in the correlator is carried out only over the connected Isserlis–Wick graphs. Equivalently, in the combinatorial sum (117) only those $\{l_{ab}\}$ should be taken into account, that correspond to the connected graphs. In calculations using computer algebra systems, one can check the connectivity of an Isserlis–Wick graph using built-in functions, recalling that $(l_{ab})_{a,b=1}^{n}$ gives the adjacency matrix.

In conclusion, Expression (137) is an explicit representation of (connected Green functions) GF for fractional power interaction as a weighted sum of integer-power interactions contributions. However, all our efforts were for deriving the explicit formulas for weights and obtaining the expression in terms of the converging series rather than asymptotic, which turned out to be quite a complicated task.

### 5.2.7. Simple Approximate Formula for General Term

In this subsection we will not use the general results on Meyer and Isserlis–Wick graphs, derived in Section 5.2.6 since in the particular case that we are going to consider in this subsection, it is more convenient to carry out all the calculations in a different, simpler and more intuitive way. However, for the higher degrees of polynomials, their use is unavoidable because of the complexity of the calculation.

As one can see from the model system, considered in Section 5.2.4, approximation with the Legendre polynomial of the second degree gives an error which is about 20%, and this is quite an accurate result. Remarkably, in this case one can finish up with a not very

complicated formula for the vacuum energy. To obtain such a formula we will start from Expression (128) and assume $g(x) = g\chi_Q(x)$:

$$\mathcal{G}_{I,n}[0]_N = \frac{(-g)^n (2G(0))^{n\alpha/2}}{n!} \mathcal{O}\left\{\prod_{a=1}^{n} \int_Q dx_a\right\} \sum_{q=0}^{N} \sum_{i=0}^{q} \frac{u_i c_q(f)}{2^{ni+1}} \frac{\Gamma(\alpha_n)}{\Gamma(n_i)} \left\langle \phi_1^{2i} \cdots \phi_n^{2i} \right\rangle_{\frac{G_{n,\Gamma}}{G(0)}}.$$

If one writes down this formula in more detail, there will appear the following coefficients:

$$A_{n,q,\alpha,i} = \left(2q + \frac{1}{2}\right) \frac{\Gamma(\alpha_n)}{\Gamma(n_i)} \sum_{p=0}^{q} \binom{2q}{2p} \binom{q+p-\frac{1}{2}}{2q} \frac{2^{4q-ni+1}}{2p+\alpha+1} \binom{2q}{2i} \binom{q+i-\frac{1}{2}}{2q}, \quad (138)$$

which we denote for the further use.

So, we want to calculate $\mathcal{G}_{I,n}[0]_N$ analytically for $N = 1$, which corresponds to the second-degree polynomial approximation. We consider the terms with $n = 1, 2$ and $n > 2$ separately. For the first two orders the calculation is straightforward:

$$\mathcal{G}_{I,1}[0]_1 = -\frac{3(2\alpha + 1)\Gamma\left(\frac{\alpha+1}{2}\right)}{\sqrt{\pi}(\alpha+1)(\alpha+3)} (2gG(0))^{\alpha/2} V, \quad (139)$$

for $n = 1$ and:

$$\mathcal{G}_{I,2}[0]_1 = \frac{15\alpha\Gamma(\alpha+1)}{16(\alpha+1)(\alpha+3)} (2gG(0))^{\alpha} V \int_Q dy \left(\frac{G(y)}{G(0)}\right)^2, \quad (140)$$

for $n = 2$. Further it will turn out that this expression will occasionally satisfy the formula for $n > 2$, so we will put this term into the sum over $n$ in the following. One important remark: in the calculation of such coordinate integrals, one has to change variables and the integration domain will deform. Though, we are interested mainly in finding results in the thermodynamic limit (finding the leading contribution in $V$, when $V \to \infty$). In this limit the integration domain deformation is not important. We will describe the thermodynamic limit in Section 6.1, and it will be more convenient to discuss such integrals there.

Let us proceed to the case $n > 2$. Because of the operator $\mathcal{O}$, the constant terms in all $s_{ab}$ will not contribute due to the differentiation. In more detail: it is so, since the sum in (80) is carried out over the connected graphs, so they have at least one edge for $n > 1$. As a result, in $\mathcal{O}$ at least exists one derivative, which will annihilate the constant term. This means, due to (117), we can consider only $i \neq 0$, so there will remain only a single term in a sum $\sum_{q=0}^{N} \sum_{i=0}^{q}$, namely $i = q = N = 1$:

$$\mathcal{G}_{I,n}[0]_1 = \frac{(-g)^n (2G(0))^{n\alpha/2}}{n!} A_{n,1,\alpha,1} \mathcal{O}\left\{\prod_{a=1}^{n} \int_Q dx_a\right\} \left\langle \phi_1^2 \cdots \phi_n^2 \right\rangle_{\frac{G_{n,\Gamma}}{G(0)}}. \quad (141)$$

At present, for the evaluation of the Isserlis–Wick correlators for the second degrees of fields, it is useful to directly sort all the pairings rather than use the obtained combinatorial formulas. Namely, one can sort all the configurations of pairings for $\left\langle \phi_1^2 \cdots \phi_n^2 \right\rangle_{\frac{G_{n,\Gamma}}{G(0)}}$ by the number $r$ of vertices, pairing with themselves. Other $n - r$ vertices have to constitute some number of closed chains, i.e., graphs whose vertex degrees are equal to 2, in terms of recently proposed graphical interpretation. This is the case since the degree of every vertex in any Isserlis–Wick graph is two, and all such graphs are disjoint unions of loops and chains. Then we can write:

$$\left\langle \phi_1^2 \cdots \phi_n^2 \right\rangle_{\frac{G_{n,\Gamma}}{G(0)}} = \sum_{r=0}^{n} G(0)^{r-n} \left\{\sum_{i_1,\ldots,i_{n-r}} (G_{n,\Gamma})_{i_1,i_2} (G_{n,\Gamma})_{i_2,i_3} \cdots (G_{n,\Gamma})_{i_{n-r},i_1} + \ldots\right\}. \quad (142)$$

The summation indices $i_a$ in the right-hand side of Expression (142) take values in $\{1, \ldots, n\}$ and enumerate all different closed chains (one closed chain as a graph can be prescribed with different sequences of indices, what will be discussed in more details further). In Expression (142) we have written only the terms with the unique chain, and denoted the others as "...". We will not write them down explicitly, since in the following they will give only zero contribution.

Further, substituting the last expression into Expression (141), we arrive at the following result:

$$
\mathcal{G}_{I,n}[0]_1 = \frac{(-g)^n (2G(0))^{n\alpha/2}}{n!} A_{n,1,\alpha,1} \mathcal{O}\left\{ \prod_{a=1}^{n} \int_Q dx_a \right\}
$$
$$
\times \left\{ \sum_{r=0}^{n} \sum_{i_1,\ldots,i_{n-r}} \frac{s_{i_1,i_2} v_{i_1,i_2}(\Gamma) G(x_{i_1} - x_{i_2})}{G(0)} \cdots \frac{s_{i_{n-r},i_1} v_{i_{n-r},i_1}(\Gamma) G(x_{i_{n-r}} - x_{i_1})}{G(0)} + \ldots \right\}.
\tag{143}
$$

Because of the operator $\mathcal{O}$, each term in the sum $\sum_{r=0}^{n}$ survives if and only if $v_{ab}(\Gamma) = 1$ for $a$ of $b$ belonging to all the vertices which both lie in the chain. From this follows that the graph $\Gamma$ has to be contained in this chain. However, since $\Gamma$ is connected, this condition cannot be satisfied for the disconnected Isserlis–Wick graphs. This means that in the sum in (143), only the terms corresponding to the connected graphs referring to pairings survive. In particular, all the terms denoted as "..." disappear, since they have at least two non-trivial chains and hence they are disconnected. Only the term with $r = 0$ in the remaining sum gives the non-zero contribution in $\sum_{\Gamma \in \mathbb{G}_{C,n}}$. One can note that this term corresponds to a cyclic chain, schematically written as $i_1 - i_2 - \ldots - i_n - i_1$ for some distinct $i_a \in \{1, \ldots, n\}$.

Moreover, due to the invariance of every term under permutations of $x_a$ in coordinate integration, we can suppose that $i_a = a$. The number of different chains from $n$ vertices equals to $2^n n!/2n$, since $n!$ is a total number of permutations of $x_a$, and we have to divide it by the number of permutations, assigning the same chain, which is $2n$. The thing is, we can make cyclic permutations of finite set of all $x_a$, which do not change a chain (and there are $n$ of such transformations) as well as reflections of finite set of all $x_a$ (which should not be confused with the transformation $x_a \to -x_a$), for which we have exactly 2 ways. In total it gives us the factor $2n$. Moreover, we have to multiply the obtained number $n!/(2n)$ by $2^n$ since we have two possible choices of fields in each vertex. Though this derivation is valid only for $n > 2$, this formula also gives the right result for $n = 2$. The illustration of different enumerations of vertices in a given chain is also presented in Figure 6.

Calculating the integrals over all $s_{ab}$ except for $s_{1,2}, \ldots, s_{n,1}$, we obtain:

$$
\mathcal{G}_{I,n}[0]_1 = \frac{(-2g)^n (2G(0))^{n\alpha/2}}{2n} A_{n,1,\alpha,1} \left\{ \prod_{a=1}^{n} \int_Q dx_a \right\}
$$
$$
\times \left\{ \int_0^1 ds_{12}\, \partial_{s_{12}} \ldots \int_0^1 ds_{n1}\, \partial_{s_{n1}} \right\} \frac{s_{12} G(x_1 - x_2)}{G(0)} \cdots \frac{s_{n,1} G(x_n - x_1)}{G(0)},
$$

where we have used that after permutation of $x_a$ there will be $n!$ equal terms, so $n!$ in the numerator and denominator cancel each other out. The remaining integrals over $s_{ab}$ are easily calculated:

$$
\mathcal{G}_{I,n}[0]_1 = \frac{(-2g)^n (2G(0))^{n\alpha/2}}{2n} A_{n,1,\alpha,1} \left\{ \prod_{a=1}^{n} \int_Q dx_a \right\} \frac{G(x_1 - x_2)}{G(0)} \cdots \frac{G(x_n - x_1)}{G(0)}.
$$

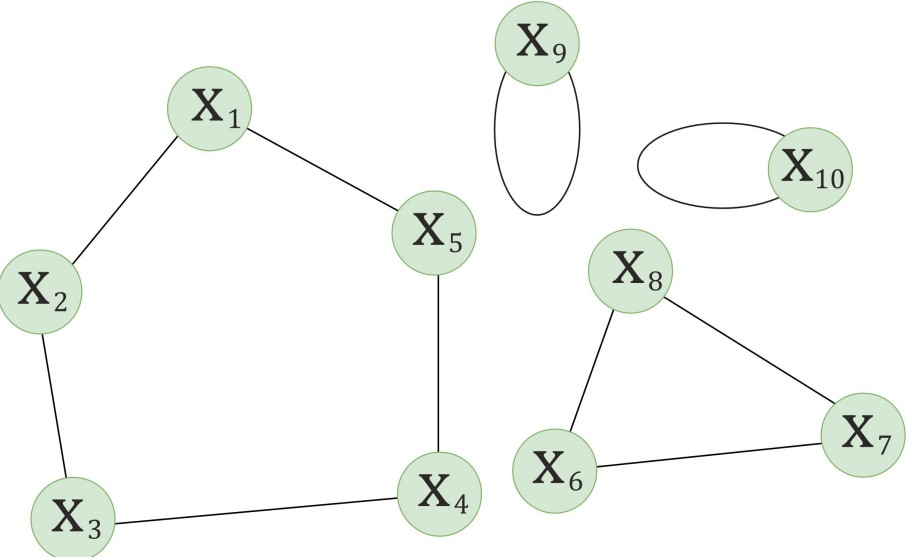

**Figure 6.** Typical possible graph from Isserlis–Wick theorem for the for ten-dimensional Gaussian moment with integer powers, equal to 2. The vertices are enumerated with the coordinates, corresponding to the field insertions. Additionally, the several possible ways of vertices enumeration are presented (clockwise and counterclockwise), illustrating the genesis of a combinatorial factor.

Let us change the variables to simplify this expression:

$$
\begin{cases} x_a - x_{a+1} = y_a, & \text{if } a \in \{1, \ldots, n-1\}, \\ x_n = y_n; \end{cases}
\Rightarrow \vec{y} = \begin{pmatrix} 1 & -1 & 0 & \ldots & 0 \\ 0 & 1 & -1 & 0 & \ldots \\ \ldots & \ldots & \ldots & \ldots & \ldots \\ 0 & \ldots & 0 & 1 & -1 \\ 0 & 0 & \ldots & 0 & 1 \end{pmatrix} \vec{x},
$$

and the last argument is the sum of all the previous:

$$
x_n - x_1 = (x_n - x_{n-1}) + (x_{n-1} - x_{n-2}) + \ldots + (x_2 - x_1).
$$

The Jacobian of such a transformation is equal to 1, since it is a linear map with upper-triangular matrix. Here, we also assume the limit $V \to \infty$, so we will not change the integration domain (since it is "big enough yet"), referring to the independent consideration of thermodynamic limit in the Section 6.1. So we receive the result for $n > 2$:

$$
\mathcal{G}_{I,n}[0]_1 = \frac{(-2g)^n (2G(0))^{n\alpha/2}}{2n} V A_{n,1,\alpha,1} \left\{ \prod_{a=1}^{n-1} \int_Q dy_a \, \frac{G(y_a)}{G(0)} \right\} \frac{G\left(\sum_{k=1}^{n-1} y_k\right)}{G(0)}.
$$

The coefficients $A_{n,1,\alpha,1}$ can also be simplified:

$$
A_{n,1,\alpha,1} = 2^{-n-1} \frac{\Gamma(\alpha n)}{\Gamma(3n/2)} \frac{15\alpha}{(\alpha+1)(\alpha+3)},
$$

thus, finally, the expression for vacuum energy density (21) reads:

$$
w_{vac,N=1} = \frac{3(2\alpha+1)\Gamma\left(\frac{\alpha+1}{2}\right)}{\sqrt{\pi}(\alpha+1)(\alpha+3)} (2gG(0))^{\alpha/2} + \frac{15\alpha}{(\alpha+1)(\alpha+3)}
$$

$$
\times \sum_{n=2}^{\infty} \frac{(-1)^{n-1} 2^{n\alpha/2}}{4n} \left[ gG(0)^{\alpha/2} \right]^n \frac{\Gamma(\alpha n)}{\Gamma(3n/2)} \left\{ \prod_{a=1}^{n-1} \int_Q dy_a \, \frac{G(y_a)}{G(0)} \right\} \frac{G\left(\sum_{k=1}^{n-1} y_k\right)}{G(0)},
$$

(144)

where we assume the thermodynamic limit $V \to \infty$.

Expression (144) is the simple approximate formula we worked for, and in the Sections 6 and 7 we will apply it for the calculations of vacuum energy density for particular cases of propagators $G$. Unfortunately, the analysis of similar formula even for fourth-degree approximation polynomial, i.e., $N = 2$, becomes much more complicated due to significant variety of Isserlis–Wick graphs with the desired properties. Let us make a remark that in this subsection we have received in a different way all the results from Section 5.2.6 for the particular case, namely we have checked that all the non-zero contributions correspond to the connected Isserlis–Wick graphs.

Let us also note that in the limiting case $\alpha = 2$:

$$w_{vac,N=1} = -\sum_{n=1}^{\infty} \frac{(-2g)^n}{2n} \left\{ \prod_{a=1}^{n-1} \int_Q dy_a \, G(y_a) \right\} G\left( \sum_{k=1}^{n-1} y_k \right), \qquad (145)$$

which coincides with the exact result for $\alpha = 2$. We will obtain this result in Section 6.3. This is a simple test of Expression (144), since the function $f(t) = t^2$ belongs to the linear span of zero-degree and second-degree Legendre polynomials.

*5.3. Hard-Sphere Gas Approximation*

The obtained general formulas from Section 5.2.5 are still difficult for analytical calculations since of the coordinate integrals. So we have to make some assumptions to obtain simpler and hence more useful formulas. Namely, the formulas become significantly simpler if we use for $G_n$ the HSG approximation, inspired by statistical physics. In this subsection, we will also consider the coupling constant in the form $g(x) = g\chi_Q(x)$, emulating a system with fixed coupling constant in finite volume $V$. In accordance with Expression (10) in the Section 2, we keep all the diagonal elements equal to $G(0)$, since they are constant, and "cut off" all the off-diagonal elements for $|x_a - x_b| > \delta$. For the graphical illustration in Figure 7, we will denote the right hand side of Expression (10) as $G_{n,\text{HSG}}$. Finally, according to the Section 2, we will denote the "volume of one hard-sphere particle" as $v$. Let us note, that all the quantities in HSG approximation will be expressed in terms of $v$ and $\gamma$, so the obtained formulas will not have an explicit dependence on the definition of these parameters.

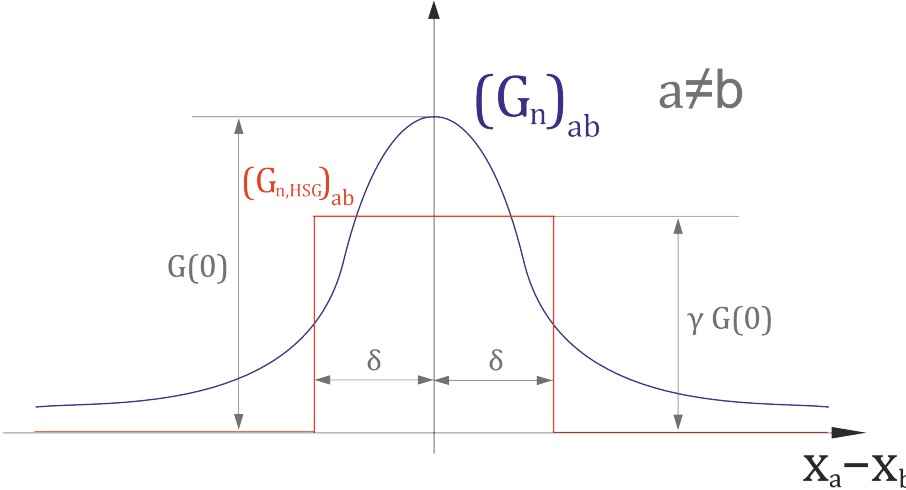

**Figure 7.** Qualitative illustration of HSG approximation.

We start from Formula (82), which precedes the Parseval–Plancherel identity, in order to avoid the appearance of $G_{n,\Gamma}^{-1}$ and $1/\sqrt{\det G_{n,\Gamma}}$. Substituting the notion of measure (68) and suggesting that $j = 0$, we arrive at the following expression:

$$
\mathcal{G}_{I,\Lambda,\varepsilon,n}[0] = \frac{(-g)^n}{n!} \sum_{\Gamma \in \mathbb{G}_{C,n}} \left\{ \prod_{a=1}^{n} \int_Q dx_a \int_{\mathbb{R}} \frac{dt_a}{2\pi} \mathcal{F}[U_\Lambda(\phi)] \exp\left( -\frac{1}{2} G_\varepsilon t_a^2 \right) \right\}
$$
$$
\times \prod_{a<b}^{n} \{ \exp[-\nu_{ab}(\Gamma)G(x_a - x_b)t_a t_b] - 1 \}.
$$

Applying for the matrix $G_n$ HSG approximation (10) and introducing auxiliary variables $s_{ab}$, we obtain the following result:

$$
\mathcal{G}_{I,\Lambda,\varepsilon,n}[0] = \frac{(-g)^n}{n!} \mathcal{O}\left\{ \prod_{a=1}^{n} \int_Q dx_a \prod_{a<b}^{n} \theta(\delta - |x_a - x_b|) \right\}
$$
$$
\times \left\{ \prod_{a=1}^{n} \int_{\mathbb{R}} \frac{dt_a}{2\pi} \mathcal{F}[U_\Lambda(\phi)] \exp\left( -\frac{1}{2} G_\varepsilon t_a^2 \right) \right\} \prod_{a<b}^{n} \exp[-\gamma G(0)s_{ab}\nu_{ab}(\Gamma)t_a t_b],
$$

(146)

where we have used the fact that:

$$
\exp[-\gamma G(0)\nu_{ab}(\Gamma)\theta(x_a - x_b)t_a t_b] - 1 = \theta(x_a - x_b)\{\exp[-\gamma G(0)\nu_{ab}(\Gamma)t_a t_b] - 1\},
$$

so the dependence of the coordinates factorises. Let us note that all the dependence of coordinates remains only in Heaviside functions.

Now we can calculate the integrals over $x_a$ approximately with a commonly accepted approximation analogical to one in the statistical physics of hard-sphere gas. We can place the first particle in the full volume which gives us $V$ and the following particle should be no further from the first than $\delta$:

$$
\left\{ \prod_{a=1}^{n} \int_Q dx_a \right\} \prod_{a<b}^{n} \theta(\delta - |x_{ab}|) \approx Vv^{n-1}.
$$

Traveling back to Expression (146), we obtain:

$$
\mathcal{G}_{I,\Lambda,\varepsilon,n}[0] = \frac{(-g)^n}{n!} Vv^{n-1} \mathcal{O}\left\{ \prod_{a=1}^{n} \int_{\mathbb{R}} \frac{dt_a}{2\pi} \mathcal{F}[U_\Lambda(\phi)] \exp\left( -\frac{1}{2} G_\varepsilon t_a^2 \right) \right\}
$$
$$
\times \prod_{a<b}^{n} \exp(-\gamma G(0)\nu_{ab}(\Gamma)s_{ab}t_a t_b).
$$

(147)

Expression (147) is much simpler than Expression (146), because the integrals over $x_a$ in (147) have already been "calculated". Well, it is still a problem to calculate directly the remaining integrals over $t_a$, so we use the developed technique of polynomial approximations. Exactly as in Sections 4.4.2 and 5.2, we find the following expression ("taking out of brackets" all the transformations that lead us to it):

$$
\mathcal{G}_{I,n}[0] = \frac{(-g)^n (2G(0))^{n\alpha/2}}{n!} Vv^{n-1} \sum_{q=0}^{\infty} \sum_{i=0}^{q} 2^{-ni-1} \frac{\Gamma(\alpha_n)}{\Gamma(n_i)} u_i c_q(f)
$$
$$
\times \mathcal{O}\left\{ \partial_1^{2i} \ldots \partial_n^{2i} \Big|_{\eta_a=0} e^{\frac{1}{2} \sum_{a,b=1}^{n} s_{ab}\nu_{ab}(\Gamma)(\gamma(J_n)_{ab} + (1-\gamma)\delta_{ab})\eta_a \eta_b} \right\},
$$

(148)

where we also used the convergence of the constructed polynomial approximations and wrote the result as a series as well as removed all the regulators.

Finally, substituting the explicit expression for the correlators (117) and using the definition of vacuum energy density $w_{vac}$ (21), we arrive at the result:

$$w_{vac} = -\sum_{n=1}^{\infty} \frac{(-g)^n (2G(0))^{n\alpha/2}}{n!} v^{n-1} \sum_{q=0}^{\infty} \sum_{i=0}^{q} \frac{(2i)!^n}{2^{ni+1}} \frac{\Gamma(\alpha_n)}{\Gamma(n_i)} u_i c_q(f)$$

$$\times \mathcal{O} \left\{ \sum_{\{l_{ab}\}} 2^{-\sum\limits_{a=1}^{n} \frac{l_{aa}}{2}} \frac{\prod\limits_{a<b}^{n} (s_{ab} \nu_{ab}(\Gamma)\gamma)^{l_{ab}}}{\prod\limits_{a<b}^{n} (l_{ab}!) \prod\limits_{a=1}^{n} \left(\frac{l_{aa}}{2}\right)!} \right\}. \tag{149}$$

One can also rewrite the last combinatorial sum, introducing $l_{aa} = 2q_a$:

$$\sum_{\{l_{ab}\}} 2^{-\sum\limits_{a=1}^{n} \frac{l_{aa}}{2}} \frac{\prod\limits_{a<b}^{n} (s_{ab} \nu_{ab}(\Gamma)\gamma)^{l_{ab}}}{\prod\limits_{a<b}^{n} (l_{ab}!) \prod\limits_{a=1}^{n} \left(\frac{l_{aa}}{2}\right)!}$$

$$= \gamma^{ni} \left\{ \prod_{a=1}^{n} \sum_{q_a=0}^{i-1} \right\} (2\gamma)^{-\sum\limits_{a=1}^{n} q_a} \frac{1}{\prod\limits_{a=1}^{n} q_a!} \sum_{\{l_{ab}\}'} \frac{1}{\prod\limits_{a<b}^{n} (l_{ab}!)} \prod_{a<b}^{n} (s_{ab} \nu_{ab}(\Gamma))^{l_{ab}}. \tag{150}$$

**Convention**: the last summation in the second line of Expression (150) is carried out over all off-diagonal ($a < b$) $l_{ab} > 0$ satisfying the conditions $\sum\limits_{\substack{b \neq a}}^{n} l_{ab} = 2i - 2q_a$ (where still $l_{ab} = l_{ba}$). Expression (150) is useful for decreasing of the computational time for numerical calculations using Formula (149).

Summarizing, we went ahead with some relatively simple approximate expression for vacuum energy density $w_{vac}$ for a generic nonlocal propagator $G$. It can be simply realized using any software suitable for symbolic computations, such as Wolfram Mathematica. Its main advantage is that unlike in general formulas for GF one does not need to compute numerically or analytically coordinate integrals. This fact makes us suppose that HSG approximation is good enough for investigating the qualitative and partly quantitative properties of nonlocal theories.

### 6. PT Calculation of System's Physical Characteristics

Using the results obtained one can calculate the following characteristics of the quantum scalar field in nonlocal theory:

1. quantum scalar field vacuum energy density;
2. quantum scalar field Green functions in terms of functional integral over primary field $\phi$, for example, two-particle Green function;
3. quantum scalar field composite operators Green functions, which are also often called "form factors".

In this paper, we will focus on the vacuum energy density computation. We will use the first few exactly calculated terms of PT series in coupling constant $g$ as well as approximate expressions for general terms and explore what physical results can be obtained from them. Derivation for the non-zero source general case remains the same but includes more technical details and large calculations. Everywhere in Sections 6 and 7 we will suppose $g(x) = g\chi_Q(x)$ in all the obtained formulas.

We start by considering special cases $n = 1, 2$ of the general Formula (81) (everywhere in this section we refer to graphs only as the Meyer graphs; see Section 5.2.6):

1. for $n = 1$ there are no $s_{ab}$ variables, since $a < b$, and the only graph is the single-point, therefore:

$$\mathcal{G}_1[0, G_1 = G(0)] = \mathcal{Z}_1[0, G_{1,\Gamma} = G(0)]; \tag{151}$$

2. for $n = 2$ there is one $s_{ab}$ variable, namely $s_{12}$, and the only one connected graph without loops $\Gamma$, for which $\nu_{12} = 1$. So the matrix $G_{2,\Gamma}$ has a form:

$$G_{2,\Gamma} = \begin{pmatrix} G(0) & s_{12}G(x_1 - x_2) \\ s_{12}G(x_1 - x_2) & G(0) \end{pmatrix},$$

and Formula (81) in this case reads as follows:

$$\mathcal{G}_2[0, (G_2)_{ab}] = \int_0^1 ds_{12}\, \partial_{s_{12}}\, \mathcal{Z}_2[0, (G_{2,\Gamma})_{ab}]. \tag{152}$$

Let us note that we consider only the zero-source case, for which the index $I$ of GFs can be omitted due to Definitions (25) and (26).

Now we are going to calculate the first orders of PT from Section 4 exactly. It appears that for $n = 1, 2$ it is possible to do without using the constructed polynomial approximations or HSG approximation. For $n = 1$, the result can be expressed in terms of elementary, and for $n = 2$ in terms of Gauss hypergeometric functions.

### 6.1. First Orders of Complete Green Functions and Connected Green Functions GFs

In this subsection we will provide exact expressions for $\mathcal{Z}_n$ for $n = 1, 2$. Then we obtain from them the corresponding $\mathcal{G}_n$. We start with Expression (54):

$$\mathcal{Z}_n[0, (G_n)_{ab}] = \frac{(-g)^n}{n!(2\pi)^{n/2}} \left\{ \prod_{a=1}^n \int_Q dx_a \int_{\mathbb{R}} d\phi_a\, |\phi_a|^\alpha \right\} \frac{e^{-\frac{1}{2}\sum_{a=1}^n (G_n)_{ab}^{-1}\phi_a\phi_b}}{\sqrt{\det(G_n)}}. \tag{153}$$

The matrix $G_n$ can be degenerate on null sets, though is does not affect the convergence of integrals because of the obtained majorant in Section 4.3.

The first order calculation is straightforward. Applying the scaling transformation of $\phi$ and calculating the integrals over $\phi$ and then over $x$, we arrive at the following result:

$$\mathcal{G}_1[0, G(0)] = \mathcal{Z}_1[0, G(0)] = -\frac{gG(0)^{\frac{\alpha}{2}}V2^{\frac{\alpha}{2}}\Gamma\left(\frac{\alpha+1}{2}\right)}{\pi^{1/2}}. \tag{154}$$

The second order calculation demands more mental and technical effort. First, we write the matrix $R_2 := (G_2)^{-1}$, inverse to the matrix $G_2$:

$$R_2(x_1, x_2) = \frac{1}{G(0)^2 - G(x_1 - x_2)^2} \begin{pmatrix} G(0) & -G(x_1 - x_2) \\ -G(x_1 - x_2) & G(0) \end{pmatrix}. \tag{155}$$

This matrix is defined almost everywhere in $x_a$ space. So for the complete Green functions GF two-particle contribution we have the multiple integral:

$$\mathcal{Z}_2[0, (G_2)_{ab}] = \frac{g^2}{4\pi} \int_Q \int_Q \frac{dx_1\, dx_2}{\sqrt{G(0)^2 - G(x_1 - x_2)^2}}$$
$$\times \int_{\mathbb{R}} \int_{\mathbb{R}} d\phi_1\, d\phi_2\, |\phi_1|^\alpha |\phi_2|^\alpha\, e^{-\frac{1}{2}R_{11}^{(2)}\phi_1^2 - \frac{1}{2}R_{11}^{(2)}\phi_2^2 - R_{12}^{(2)}\phi_1\phi_2}.$$

To calculate this integral, one should firstly rescale $\phi_1$ and $\phi_2$ variables to make the coefficients in $\phi_1^2$ and $\phi_2^2$ in the exponent equal to one. Then, calculating the obtained

integrals alternately, one can express the result in terms of Gauss hypergeometric function $_2F_1$:

$$
\mathcal{Z}_2[0, (G_2)_{ab}] = \frac{g^2 G(0)^\alpha 2^{\alpha-1} \Gamma\left(\frac{\alpha+1}{2}\right)^2}{\pi} \int_Q \int_Q dx\, dy \left(1 - \frac{G(x-y)^2}{G(0)^2}\right)^{\alpha+\frac{1}{2}}
$$
$$
\times\ _2F_1\left(\frac{\alpha+1}{2}, \frac{\alpha+1}{2}; \frac{1}{2}; \left(\frac{G(x-y)}{G(0)}\right)^2\right).
\tag{156}
$$

Here, $_2F_1$ is the Gauss hypergeometric function:

$$
_2F_1(a, b; c; z) = \sum_{n=0}^{\infty} \frac{(a)_n (b)_n}{(c)_n} \frac{z^n}{n!},
\tag{157}
$$

for $|z| < 1$ and the analytical continuation of this series for $|z| > 1$. The parameters $a$, $b$ and $c$ are positive in our paper, $(a)_n$, $(b)_n$ and $(c)_n$ are the rising factorials. Expression (156) can be simplified, but we move on to calculating $\mathcal{G}_2$.

For calculating $\mathcal{G}_2$, we have to recall (152). In the case $n = 2$ it is useful to rewrite the operator $\mathcal{O}$ as the difference of values for $s_{12} = 1$ and $s_{12} = 0$, rather than the integration of the derivative. So the following formula holds:

$$
\mathcal{G}_2[0, (G_2)_{ab}] = \mathcal{Z}_2[0, (G_{2,\Gamma})_{ab}] - \mathcal{Z}_2[0, G(0)\delta_{ab}].
\tag{158}
$$

In the result, since $_2F_1(a, b; c; 0) = 1$ for positive $a, b, c$, we obtain:

$$
\mathcal{G}_2[0, (G_2)_{ab}] = \frac{g^2 G(0)^\alpha 2^{\alpha-1} \Gamma\left(\frac{\alpha+1}{2}\right)^2}{\pi} \int_Q \int_Q dx\, dy \left\{\left(1 - \frac{G(x-y)^2}{G(0)^2}\right)^{\alpha+\frac{1}{2}}\right.
$$
$$
\left. \times\ _2F_1\left(\frac{\alpha+1}{2}, \frac{\alpha+1}{2}; \frac{1}{2}; \left(\frac{G(x-y)}{G(0)}\right)^2\right) - 1\right\}.
\tag{159}
$$

Let us note that the coordinate integrals in $\mathcal{G}_1$ and $\mathcal{G}_2$ diverge as $V \to \infty$ (in the thermodynamic limit). However, the integrals for $\mathcal{Z}_1$ and $\mathcal{Z}_2$ diverge as $V$ and $V^2$, correspondingly. Additionally, the connected Green functions GF $\mathcal{G}[0]$ is constructed so that all the terms diverging in total (with prefactors also taken into account) faster than $V$ vanish. As a result, we obtain that $\mathcal{G}[0] \sim V$, which makes correct the definition of vacuum energy density. This fact is well known in statistical physics. Thus, in the thermodynamic limit, we obtain the following result:

$$
\mathcal{G}_2[0, (G_2)_{ab}] = \frac{g^2 G(0)^\alpha V 2^{\alpha-1} \Gamma\left(\frac{\alpha+1}{2}\right)^2}{\pi} \int_{\mathbb{R}^d} dx \left\{\left(1 - \frac{G(x)^2}{G(0)^2}\right)^{\alpha+\frac{1}{2}}\right.
$$
$$
\left. \times\ _2F_1\left(\frac{\alpha+1}{2}, \frac{\alpha+1}{2}; \frac{1}{2}; \left(\frac{G(x)}{G(0)}\right)^2\right) - 1\right\}.
\tag{160}
$$

Informally, this result is expected, since it can be obtained by changing the variables $\Xi = x + y$ and $\xi = x - y$, and integration over $\Xi$ gives the factor $V$. Though, to prove it in a more strict manner we should make some effort. Namely, denoting the integrand in (159) as $f$, consider the following transformations:

$$
\int_Q \int_Q dx\, dy\, f(x-y) = \int_Q dy \left\{\int_{\mathbb{R}^d} dx\, f(x-y) - \int_{\mathbb{R}^d \setminus Q} dx\, f(x-y)\right\}.
\tag{161}
$$

In the first integral we shift the variable $x$ in the inner integration, and in the second integral we make the change of variables to $\Xi$ and $\zeta$, therefore:

$$\int_Q dy \int_{\mathbb{R}^d \setminus Q} dx\, f(x-y) = V \int_{\mathbb{R}^d \setminus 2Q} d\xi\, f(\xi) + \int_{2Q} d\xi\, \xi_1 \ldots \xi_d\, f(\xi), \tag{162}$$

where we have introduced the notation $2Q$ for the cube with the same centre, as $Q$, and doubled lengths of the edges. Further, for the applicability of the thermodynamic limit, we require that:

1. $\int_{\mathbb{R}^d \setminus 2Q} d\xi\, f(\xi) \to 0$, when $V \to \infty$;
2. $|\int_{2Q} d\xi\, \xi_1 \ldots \xi_d\, f(\xi)| < \infty$, when $V \to \infty$.

These requirements can be satisfied, for example, if the propagator $G$ satisfies the HSG approximation applicability conditions, which we assume to be satisfied throughout this paper, as already discussed after Equation (12). As a result, we have proven the expansion for $V \to \infty$:

$$\int_Q \int_Q dx\, dy\, f(x-y) = V \left\{ \int_{\mathbb{R}^d} dx\, f(x) + o(1) \right\}. \tag{163}$$

This expression finishes the proof of (160).

Finally, we also introduce the hyperspherical coordinates for rotational invariant theories, which leads us to the following result:

$$\mathcal{G}_2[0, (G_2)_{ab}] = \frac{g^2 G(0)^\alpha V 2^{\alpha-1} \Gamma^2\left(\frac{\alpha+1}{2}\right)}{\pi} \frac{d\pi^{d/2}}{\Gamma(d/2+1)}$$
$$\times \int_0^\infty dr\, r^{d-1} \left\{ \left(1 - \frac{G(r)^2}{G(0)^2}\right)^{\alpha+\frac{1}{2}} {}_2F_1\left(\frac{\alpha+1}{2}, \frac{\alpha+1}{2}; \frac{1}{2}; \left(\frac{G(r)}{G(0)}\right)^2\right) - 1 \right\}. \tag{164}$$

Because, physically, we usually consider theories with a propagator $G$ that depends only on the distance between particles. This is true, in particular, for Euclidean Klein–Gordon (with sharp cut-off) and virton propagators, which we will describe in the Section 7.

### 6.2. First Orders of Vacuum Energy Density

In this short subsection, we list the expressions for the vacuum energy density $w_{vac}$ (21). Collecting the contributions of the first two PT orders, we arrive at the final expression in thermodynamic limit:

$$w_{vac} = \frac{g G(0)^{\frac{\alpha}{2}} 2^{\frac{1+\alpha}{2}} \Gamma\left(\frac{\alpha+1}{2}\right)}{(2\pi)^{1/2}} - \frac{g^2 G(0)^\alpha 2^{\alpha-1} \Gamma\left(\frac{\alpha+1}{2}\right)^2}{\pi}$$
$$\times \int_{\mathbb{R}^d} dx \left\{ \left(1 - \frac{G(x)^2}{G(0)^2}\right)^{\alpha+\frac{1}{2}} {}_2F_1\left(\frac{\alpha+1}{2}, \frac{\alpha+1}{2}; \frac{1}{2}; \left(\frac{G(x)}{G(0)}\right)^2\right) - 1 \right\}. \tag{165}$$

Similarly for rotational invariant theory:

$$w_{vac} = \frac{g G(0)^{\frac{\alpha}{2}} 2^{\frac{1+\alpha}{2}} \Gamma\left(\frac{\alpha+1}{2}\right)}{(2\pi)^{1/2}} - \frac{g^2 G(0)^\alpha 2^{\alpha-1} \Gamma\left(\frac{\alpha+1}{2}\right)^2}{\pi} \frac{d\pi^{d/2}}{\Gamma(d/2+1)}$$
$$\times \int_0^\infty dr\, r^{d-1} \left\{ \left(1 - \frac{G(r)^2}{G(0)^2}\right)^{\alpha+\frac{1}{2}} {}_2F_1\left(\frac{\alpha+1}{2}, \frac{\alpha+1}{2}; \frac{1}{2}; \left(\frac{G(r)}{G(0)}\right)^2\right) - 1 \right\}. \tag{166}$$

We will consider important particular cases in the following.

*6.3. Verification of Pt Formulas for the First Orders for $\alpha = 2$*

We can independently calculate the GF $\mathcal{Z}$ for $\alpha = 2$, which simply yields the Gaussian integral. Using the result, one can check the formulas for the vacuum energy density, obtained above. In this subsection, we restrict ourselves to the case of a zero source and denote all the results of the presented independent calculation specific for $\alpha = 2$ with the upper index: something$^{(\alpha=2)}$.

6.3.1. Exact Result

We start from the GF $\mathcal{Z}$ in terms of path integral in physical notations of Section 3:

$$\mathcal{Z}^{(\alpha=2)}[0] = \int_{\Phi} \mathcal{D}[\phi]\, e^{-\frac{1}{2}\int_{\mathbb{R}^d}\int_{\mathbb{R}^d} dx\, dy\, L(x,y)\phi(x)\phi(y) - \int_{\mathbb{R}^d} dx\, g(x)\phi(x)^2},$$

with the same requirements to $L$ and $G = L^{-1}$, as in the Section 4.1. In terms of the Gaussian measure $\gamma_G$, this expression reads as follows:

$$\mathcal{Z}^{(\alpha=2)}[0] = \int_{\Phi} \gamma_G(d\phi)\, e^{-\int_{\mathbb{R}^d} dx\, g(x)\phi(x)^2}.$$

Such an integral is well-known and equals to [47] (hereafter, Id is the identity operator):

$$\mathcal{Z}^{(\alpha=2)}[0] = \frac{1}{\sqrt{\det(\mathrm{Id} + 2Gg)}}. \tag{167}$$

We define the product $Gg$ of operators $G$ and $g$ as the result of applying first the new operator $g$, which consists in multiplying by the function $g$ (we denote this function with the same symbol), and then the operator $G$. The operator $Gg$ is trace class as a product of trace class $G$ and bounded $g$. Therefore, it is compact and has a countable set of eigenvalues, and, possibly, zero in its spectrum, and nothing more. We denote these eigenvalues as $(Gg)_n$ for $n \in \mathbb{N}$. The infinite-dimensional determinant is defined as follows:

$$\det(\mathrm{Id} + 2Gg) := \prod_{n=1}^{\infty} \{1 + 2(Gg)_n\}. \tag{168}$$

Let us note that it is exactly the Fredholm determinant of the operator $Gg$. This infinite product converges, since converges the series $\mathrm{tr}\,(Gg) = \sum_{n=1}^{\infty} (Gg)_n$. Expression (167) can be obtained with the same methods, as PT in Section 4.2. One should apply DCT and then calculate the remaining Gaussian integrals, which gives the desired formula. For the GF $\mathcal{G}$ we therefore obtain:

$$\mathcal{G}^{(\alpha=2)}[0] = -\frac{1}{2}\mathrm{tr}\ln(\mathrm{Id} + 2Gg), \tag{169}$$

where we have used the formula $\ln\det(1 + A) = \mathrm{tr}\ln(1 + A)$ for trace class operator $A$. This formula can be proved using the definition of function of an operator through the values of this function on the eigenvalues of $A$, and we will not present its derivation here, referring to textbooks in functional analysis. Expanding the logarithm of an operator, using equivalent for analytical functions and bounded operators definition of function of an operator, we obtain the following series:

$$\mathcal{G}^{(\alpha=2)}[0] = \frac{1}{2}\sum_{n=1}^{\infty} \frac{(-2)^n}{n} \mathrm{tr}\{(Gg)^n\}. \tag{170}$$

Now we rewrite the powers of operators in terms of integrals, using the notion of operator's $G$ integral kernel $G(x - y)$ and also substitute the coupling constant $g(x) = g\chi_Q(x)$, which cuts off the integration domains:

$$\mathcal{G}^{(\alpha=2)}[0] = \frac{1}{2}\sum_{n=1}^{\infty}\frac{(-2g)^n}{n}\left\{\prod_{a=1}^{n}\int_Q dx_a\right\}G(x_n - x_1)\prod_{a=1}^{n-1}G(x_a - x_{a+1}). \tag{171}$$

Similarly to Section 5.2.7, we change variables and reduce the number of integrals by one, which leads to the appearance of the factor $V$:

$$\mathcal{G}^{(\alpha=2)}[0] = \frac{1}{2}V\sum_{n=1}^{\infty}\frac{(-2g)^n}{n}\left\{\prod_{a=1}^{n-1}\int_Q dy_a\,G(y_a)\right\}G\left(\sum_{a=1}^{n-1}y_a\right). \tag{172}$$

Finally we can write down the expression for the vacuum energy density (21):

$$w_{vac}^{(\alpha=2)} = -\frac{\mathcal{G}[0]}{V} = -\frac{1}{2}\sum_{n=1}^{\infty}\frac{(-2g)^n}{n}\left\{\prod_{a=1}^{n-1}\int_Q dy_a\,G(y_a)\right\}G\left(\sum_{a=1}^{n-1}y_a\right). \tag{173}$$

From this expression one can see that it coincides with Legendre polynomial approximation for $\alpha = 2$, obtained in the Section 5.2.7.

6.3.2. Comparison of PT Formulas for the First Orders for $\alpha = 2$ with the Exact Result

Let us recall Formulas (154) and (160) for $\mathcal{G}_1$ and $\mathcal{G}_2$ and substitute $\alpha = 2$ into them. Taking into account, that:

$$_2F_1\left(\frac{3}{2}, \frac{3}{2}; \frac{1}{2}; z^2\right) = \frac{2z^2 + 1}{(1 - z^2)^{5/2}}, \tag{174}$$

we receive the following compact expressions:

$$\mathcal{G}_1[0, G(0)] = -gG(0)V, \quad \mathcal{G}_2[0, (G_2)_{ab}] = g^2V\int_{\mathbb{R}^d}dx\,G(x)^2, \tag{175}$$

which coincides with directly obtained result for $\alpha = 2$. So, our formulas pass this simple test, which indirectly displays that they are correct.

## 7. Research of System Physical Characteristics

### 7.1. Research of First PT Terms in Vacuum Energy Density

In this subsection we are going to calculate the first PT terms for vacuum energy density. Formerly we have obtained the general Formulas (154) and (164), and now we are going to substitute different propagators into these formulas: virton propagator [3], which is purely nonlocal, and Euclidean Klein–Gordon propagator (with sharp cut-off), which admits a local limit. Then we will consider a local theory as a certain limit of a nonlocal one, being its regularization. However, let us note that the nonlocal theory is of important independent interest.

#### 7.1.1. Nonlocal Case

We start from a typical nonlocal QFT propagator, which is the virton propagator:

$$G(x) = G(0)\,e^{-\mu^2 x^2}, \tag{176}$$

where $\mu$ is an ultraviolet cut-off parameter and $G(0) > 0$ is a some constant. Let us calculate numerically the following integral:

$$\psi(d, \alpha) = \int_0^{\infty}dt\,t^{d-1}\left\{\left(1 - e^{-2t^2}\right)^{\alpha+\frac{1}{2}}{}_2F_1\left(\frac{\alpha+1}{2}, \frac{\alpha+1}{2}; \frac{1}{2}; e^{-2t^2}\right) - 1\right\}. \tag{177}$$

The plots for $\psi(d, \alpha)$ depending on $\alpha$ at different $d$ are presented in Figure 8. Let us note that the integrand above is always positive.

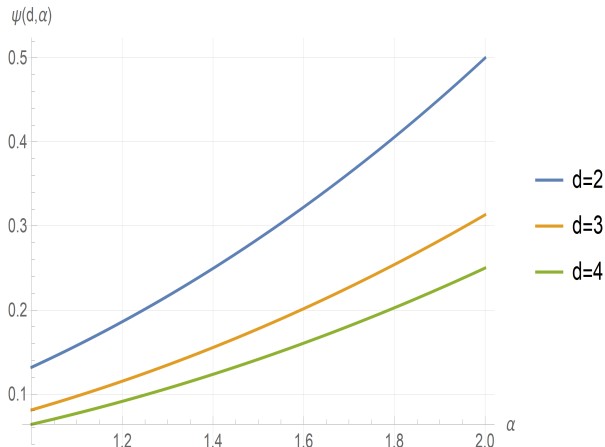

**Figure 8.** Results of numerical calculation of the integral $\psi(d, \alpha)$.

As a result, we arrive at the following expression for the vacuum energy density:

$$w_{vac} = b_1 g G(0)^{\frac{\alpha}{2}} - b_2 \frac{g^2 G(0)^\alpha}{\mu^d} + O(g^3), \tag{178}$$

where:

$$b_1 = \frac{2^{\frac{\alpha}{2}} \Gamma\left(\frac{\alpha+1}{2}\right)}{\pi^{1/2}}, \quad b_2 = \frac{2^{\alpha-1} \Gamma\left(\frac{\alpha+1}{2}\right)^2 \pi^{d/2-1}}{\Gamma\left(\frac{d}{2}+1\right)} \psi(d, \alpha).$$

In all the following calculations, we will see the qualitatively similar behaviour of $w_{vac}$ at small $g$: positive coefficient in linear term and the negative one in quadratic term. Therefore, all the results obtained in the paper are consistent with each other.

### 7.1.2. Local Case

Now let us consider the local QFT and calculate the first terms of vacuum energy density for the Klein–Gordon propagator. We consider the usual cut-off regularization of Gaussian theory Green function with Heaviside step function, i.e., sharp cut-off:

$$G(x) = \int_{\mathbb{R}^d} \frac{dk}{(2\pi)^d} \frac{e^{ikx}}{k^2 + m^2} \theta(\mu - |k|),$$

where $\mu$ is again the UV cut-off parameter. Since we want to obtain a local theory as a limiting case, it is appropriate to consider the cases of $d = 2, 3$. Taking into account, that $\mu \gg m$, for $G(0)$ for Klein–Gordon propagator in $d$ dimensions one have:

$$G(0) = \int_{\mathbb{R}^d} \frac{dk}{(2\pi)^d} \frac{\theta(\mu - |k|)}{k^2 + m^2} \simeq \begin{cases} \frac{1}{4\pi} \ln\left(\frac{\mu^2}{m^2}\right), & d = 2; \\ \frac{\mu}{2\pi^2}, & d = 3. \end{cases} \tag{179}$$

It will be convenient not to substitute $G(0)$ explicitly in the obtained formulas.

To calculate the second order perturbation theory, let us note that the integrand depends on the ratio $\frac{G(x)}{G(0)}$, where $G(0)$ is an extremely large. So one can find the value of

the integral in the dominating order in $\mu$ by expanding the integrand in powers of $\frac{G(x)}{G(0)}$. So, in principal order in $\mu$:

$$\mathcal{G}_2[0] \simeq 2^{\alpha-1}\alpha^2\Gamma\left(\frac{\alpha+1}{2}\right)^2 g^2 V G(0)^\alpha \int_{\mathbb{R}^d} dx \left(\frac{G(x)}{G(0)}\right)^2,$$

where we use the analyticity of $_2F_1$ in zero argument and take the values of derivatives from (157). We can directly calculate the remaining integral:

$$\int_{\mathbb{R}^d} dx\, G^2(x) = \int_{\mathbb{R}^d} \frac{dk}{(2\pi)^d} \frac{\theta(\mu-|k|)}{(k^2+m^2)^2} = \begin{cases} \frac{1}{2\pi}\int_0^\infty dk\, \frac{k}{(k^2+m^2)^2} = \frac{1}{4m^2\pi}, & d=2; \\ \frac{1}{2\pi^2}\int_0^\infty dk\, \frac{k^2}{(k^2+m^2)^2} = \frac{1}{8m\pi}, & d=3. \end{cases} \tag{180}$$

In the second equality of Expression (180), we removed the UV regulator, since this is possible for the corresponding values of $d$. Hence, we receive:

$$\mathcal{G}_2[0] \simeq \begin{cases} \frac{2^{\alpha-3}\alpha^2}{m^2\pi}\Gamma\left(\frac{\alpha+1}{2}\right)^2 g^2 V G(0)^{\alpha-2}, & d=2; \\ \frac{2^{\alpha-4}\alpha^2}{m\pi}\Gamma\left(\frac{\alpha+1}{2}\right)^2 g^2 V G(0)^{\alpha-2}, & d=3. \end{cases} \tag{181}$$

Putting it all together, for the first PT terms for the vacuum energy density $w_{vac}$ we obtain the expression, that was already mentioned in the Section 2:

$$w_{vac}\, m^{-d} = a_1 \frac{g G(0)^{\alpha/2}}{m^d} + a_2 \left(\frac{g G(0)^{\alpha/2}}{m^d}\right)^2 \left(\frac{m^{d-2}}{G(0)}\right)^2 + O(g^3)$$

$$a_1 = 2^{\frac{\alpha+1}{2}}\Gamma\left(\frac{\alpha+1}{2}\right), \quad a_2 = \frac{2^{\alpha-1}}{\pi}\Gamma\left(\frac{\alpha+1}{2}\right)^2 \begin{cases} \frac{1}{4}, & d=2; \\ \frac{1}{8}, & d=3. \end{cases}$$

Here, we have artificially separated some combination of parameters. In the Section 7.3 it will turn out that a necessary condition for the non-triviality of the local limit is to keep constant exactly these combinations. However, it should be noted that obtaining local theories, as limiting cases of nonlocal ones, generally requires a deeper research.

*7.2. Research of Second-Degree Legendre Polynomial Approximation Formula*

Let us start from the general Formula (144) for second-degree Legendre polynomial approximation (in the thermodynamic limit, in which the vacuum energy density is of greatest interest):

$$w_{vac,N=1} = \frac{15\alpha/4}{(\alpha+1)(\alpha+3)}\left\{ \Gamma\left(\frac{\alpha+1}{2}\right)\frac{2^{\alpha/2+2}}{\sqrt{\pi}}\frac{1+2\alpha}{5} g G(0)^{\alpha/2} + \sum_{n=2}^\infty \frac{(-1)^{n-1}}{n} \right.$$

$$\left. \times\, 2^{n\alpha/2}\frac{\Gamma\left(\frac{n(\alpha+1)}{2}\right)}{\Gamma\left(\frac{3n}{2}\right)}\left[g G(0)^{\alpha/2}\right]^n \left\{ \prod_{a=1}^{n-1}\int_{\mathbb{R}^d} dy_a\, \frac{G(y_a)}{G(0)} \right\} \frac{G\left(\sum_{a=1}^{n-1} y_a\right)}{G(0)} \right\}. \tag{182}$$

In the following, we will consider only the terms with $n \geq 2$, since the case $n=1$ is already simple enough. Now we are going to substitute different propagators into this formula.

1. **Nonlocal Case**. For the virton propagator, the integrals in Expression (182) are Gaussian, therefore, can be calculated explicitly:

$$
\left\{ \prod_{a=1}^{n-1} \int_{\mathbb{R}^d} dy_a \, \frac{G(y_a)}{G(0)} \right\} \frac{G\left( \sum_{a=1}^{n-1} y_a \right)}{G(0)} = \frac{(2\pi)^{\frac{n-1}{2}d}}{\sqrt{n}\mu^{d(n-1)}}.
$$

Thus we arrive at the following expression:

$$
w_{vac,N=1}(z,\mu) = \frac{15\alpha/4}{(\alpha+1)(\alpha+3)} \frac{\mu^d}{(2\pi)^{d/2}} h\left( (2\pi)^{d/2} 2^{\alpha/2} z \right), \tag{183}
$$

for the entire function:

$$
h(z) = \frac{4\alpha-3}{5} \frac{\Gamma\left( \frac{(\alpha+1)}{2} \right)}{\Gamma(3/2)} z + \sum_{n=1}^{\infty} \frac{(-1)^{n-1}}{n^{3/2}} z^n \frac{\Gamma\left( \frac{n(\alpha+1)}{2} \right)}{\Gamma(3n/2)}, \tag{184}
$$

and the parameter:

$$
z = gG(0)^{\alpha/2}/\mu^d. \tag{185}
$$

The summation over $n$ indeed starts from one, but it is convenient to separate the linear term into two parts.

2. **Local Case**. Again we restrict ourselves to the cases $d = 2$ and $3$. We have already calculated $G(0)$ for Klein–Gordon propagator. For integrals, that we have for $n \geq 2$, the following chain of equalities is true:

$$
\left\{ \prod_{a=1}^{n-1} \int_{\mathbb{R}^d} dy_a \, \frac{G(y_a)}{G(0)} \right\} \frac{G\left( \sum_{a=1}^{n-1} y_a \right)}{G(0)} = \int_{\mathbb{R}^d} \frac{dk}{(2\pi)^d} \frac{1}{(k^2 + m^2)^n} = \frac{(m^2)^{\frac{d}{2}-n} \Gamma\left( n - \frac{d}{2} \right)}{(4\pi)^{\frac{d}{2}} \Gamma(n)}.
$$

We removed the UV regulator again. This formula is valid for $d = 2,3$. It is convenient to introduce the following dimensionless parameters:

$$
z = \frac{gG(0)^{\alpha/2}}{m^d} = \text{const}, \quad \xi = \frac{m^{d-2}}{G(0)} = \text{const}. \tag{186}
$$

In contrast to the nonlocal case, in the local one we define the parameter $z$ in terms of $m$. In the following subsection we will explore the way to scale parameters of a nonlocal theory to obtain a non-trivial local limit. Looking ahead, we announce that keeping exactly these parameters constant is a necessary condition of the non-trivial local limit existence.

So, in these terms the expression for the vacuum energy density reads:

$$
\begin{aligned}
w_{vac,N=1} m^{-d} = \frac{15\alpha/4}{(\alpha+1)(\alpha+3)} &\left\{ \Gamma\left( \frac{\alpha+1}{2} \right) \frac{2^{\alpha/2+2}}{\sqrt{\pi}} \frac{1+2\alpha}{5} z \right. \\
&\left. + \sum_{n=2}^{\infty} \frac{(-1)^{n-1} 2^{n\alpha/2}}{(4\pi)^{d/2} n!} \frac{\Gamma\left( \frac{n(\alpha+1)}{2} \right) \Gamma\left( n - \frac{d}{2} \right)}{\Gamma\left( \frac{3n}{2} \right)} (z\xi)^n \right\}.
\end{aligned} \tag{187}
$$

The numerical plots of (183) and (187) are presented in Figure 9. Their discussion will be given in the Section 7.5. Let us note that these formulas are not very friendly for numerical calculations, though, their asymptotics may be of separate interest. In the present paper, however, we restrict ourselves to numerical results.

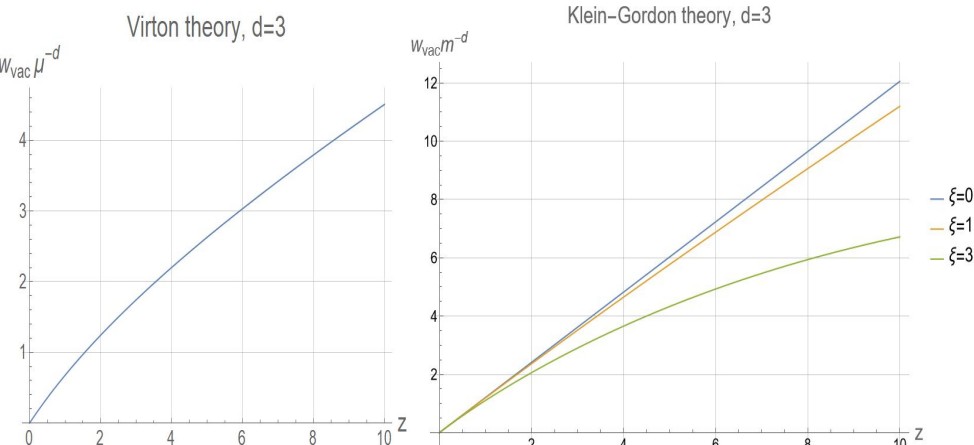

**Figure 9.** Plots of vacuum energy density $w_{vac}$ second-degree approximations (144), applied to virton (183) and Klein–Gordon (187) cases in dimension $d = 3$ and $\alpha = 4/3$. The picture is qualitatively the same for $d = 2$ and others $\alpha \in (1, 2)$.

### 7.3. Local Theory as a Limit of a Nonlocal One

In this subsection we are going to research briefly whether it is possible to scale nonlocal theory in some way to obtain a local one. We start from the writing the generic term of the polynomial approximation for $\mathcal{G}_I[0]$. Everywhere in this subsection the term "converging integral" means the converges of integral when the region of integration extends from $Q$ to $\mathbb{R}^d$. The same for the term "diverging integral". We also will use the notations from Formulas (117) and (128).

According to these formulas, we have the following $n$-particle term for the $\mathcal{G}_I[0]$ (to simplify the analysis, only one term from the combinatorial sum is written out and up to a some numerical coefficient depending on the term):

$$\left(gG(0)^{\alpha/2}\right)^n \left\{\prod_{a=1}^{n} \int_Q dx_a\right\} \prod_{a<b}^{n} \frac{G(x_a - x_b)^{l_{ab}}}{G(0)^{l_{ab}}} = Vm^d \left(\frac{gG(0)^{\alpha/2}}{m^d}\right)^n \left(\frac{m^{d-2}}{G(0)}\right)^{\sum\limits_{a<b}^{n} l_{ab}}.$$

We have written this expression from the dimension considerations and translational invariance of the considering propagator. Let us assume that all the coordinate integrals except one converge, and in this case $m$ is the only dimensional parameter in the integrals. So for the vacuum energy density we have the following expression:

$$w_{vac} m^{-d} = \Psi\left(\frac{gG(0)^{\alpha/2}}{m^d}, \frac{m^{d-2}}{G(0)}\right),$$

for some function $\Psi$ (formal power series). Therefore, in order to obtain a non-trivial local limit one should scale $m$, $g$ and $G(0)$ so that the arguments of $\Psi$ stay constant. However, it is only a **necessary** condition but not sufficient, since for the degrees of the polynomial approximation higher than two there appear diverging integrals and one have to solve the renormalization problem.

It is well-known that in $d = 2$ all the power theories are renormalizable and in $d = 3$ only the theories with power which $\leq 5$ are renormalizable. This means that for $d = 3$ it is impossible to gather all the "infinities" coming from the monomials with degrees higher than 4 into some new parameters. So, we cannot write a better approximation for $d = 3$ than the fourth-degree approximation.

The conclusion is that being applied to local theories in $d > 2$, our method has a limited precision of degree 4 in $d = 3$ and degree 2 in $d \geq 4$. Though, even the second-degree approximation can provide some tool for (rough) quantitative research.

### 7.4. Hard-Sphere Gas Approximation

The formulas for vacuum energy density (149) and (150), obtained using HSG, are convenient and simple for numerical calculations. As a practise in statistical physics and numerical simulations show, HSG approximation gives worthy results, consistent with the experiment. Unfortunately, we have to deal with the power series in system parameters, so it requires accurate work with precision of calculations to distinguish the errors from real results. We make the truncations of both series in Expression (149), namely $n_0$ for index $n$ and $N$ for index $q$, and use a numerical experiment to explore the truncations influence on the results. We assume $\alpha = 4/3$, and one can check that there will be no qualitative differences for others $\alpha \in (1, 2)$.

Recall that in the HSG approach we have approximated the matrix $G_n$ by the Heaviside step function and parameters $\gamma$ and $\delta$ in accordance with Expression (10). Let us introduce new variables:

$$z = gG(0)^{\alpha/2}v, \quad \varepsilon_{vac} = w_{vac}v, \tag{188}$$

where $\varepsilon_{vac}$ is the vacuum energy, contained in one "hard-ball" (specific energy). Further, we are going to plot the dependency $\varepsilon_{vac}(\gamma, z)$, since for HSG approximation $\gamma$ and $z$ are the most natural parameters.

The plots for different truncations $(n_0, N)$ are presented in Figure 10.

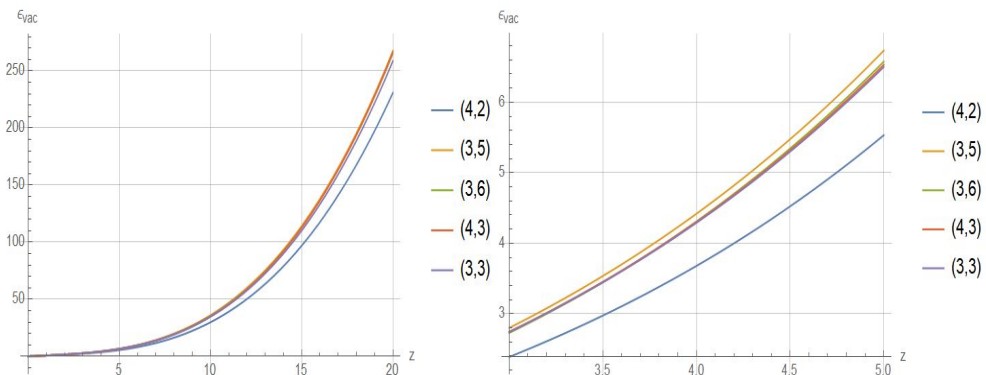

**Figure 10.** Comparative plots of specific energy $\varepsilon_{vac}$ approximations for $\gamma = 0.5$ and different values of truncations $(n_0, N)$. Only the complete graph is taken into account in the calculations.

So we argue that a quite accurate and simple approximation for complete graph contribution for $z \in [0, 20]$ and $\gamma \in [0, 1]$ is given by the following expression:

$$\varepsilon_{vac}(\gamma, z) \approx \frac{1}{\sqrt{\pi}(\alpha + 1)(\alpha + 3)(\alpha + 5)(\alpha + 7)} \left\{ \frac{88,179}{12,597} 2^{\frac{\alpha}{2}} \left(2\alpha^2 + 2\alpha + 15\right) \right.$$

$$\times (2\alpha + 1)\Gamma\left(\frac{\alpha + 1}{2}\right)z - 3^2\sqrt{\pi}2^{\alpha-14}\alpha\gamma^2 z^2 \left[\alpha^2\left(1144\gamma^4 + 3300\gamma^2 - 3645\right) - 6\alpha\right.$$

$$\times \left(1144\gamma^4 + 1540\gamma^2 - 5085\right) + 8\left(1144\gamma^4 + 660\gamma^2 - 45\right)\left]\Gamma(\alpha + 1) + \frac{2^{\frac{3\alpha}{2}-1}}{12,597}\alpha\gamma^3 z^3\right. \tag{189}$$

$$\times \left[8\left(4160\gamma^6 + 14,040\gamma^5 + 21,528\gamma^4 - 13,824\gamma^3 - 55,728\gamma^2 - 47061\gamma + 80,244\right)\right.$$

$$-6\alpha\left(4160\gamma^6 + 14,040\gamma^5 + 21,528\gamma^4 - 9948\gamma^3 - 47,976\gamma^2 - 41,247\gamma + 33,732\right)$$

$$\left.\left.+\alpha^2\left(4160\gamma^6 + 14,040\gamma^5 + 21,528\gamma^4 - 2196\gamma^3 - 32,472\gamma^2 - 29,619\gamma + 16,290\right)\right]\right\}.$$

This expression looks complicated but algebraically it is a polynomial in $z$ and $\gamma$, so it can be manipulated without significant struggles in any system of computer algebra, e.g., Wolfram Mathematica.

At the end of the subsection, it is curious to plot the 2-dimensional surface of $\varepsilon_{vac}(\gamma, z)$ for $n_0 = 3$ and $N = 3$ as well as the plot $\varepsilon_{vac}(\gamma, z)$ for some chosen $\gamma$. The results are presented in Figure 11.

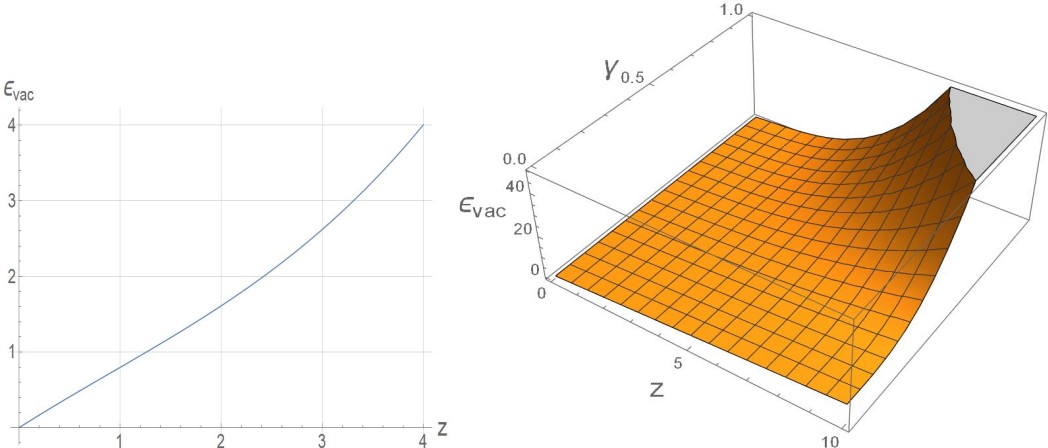

**Figure 11.** Plot of specific energy $\varepsilon_{vac}(\gamma = 0.5, z)$ **(left)** and surface of $\varepsilon_{vac}(\gamma, z)$ **(right)** for $n_0 = 3$, $N = 3$ and $\alpha = 4/3$.

The following subsection is devoted to the discussion of plots.

### 7.5. Analysis of Plots and Comparison with PT

Throughout this paper we have obtained the plots of vacuum energy density or specific energy for three different cases, namely:

1. virton vacuum energy density for the second-degree polynomial approximation (Figure 9);
2. Klein–Gordon vacuum energy density for the second-degree polynomial approximation (Figure 9);
3. specific energy in HSG approximation (Figure 11).

Throughout this subsection, for every theory from the three considered (virton, Klein–Gordon and HSG), we will use the notation $z$ for the argument, keeping in mind that its definition is the same for every theory and is given by (185), (186) and (188), correspondingly. For this reason, it is also incorrect to compare the numerical values of $w_{vac}$ for Klein–Gordon and virton theories, because the parameters $\mu$ and $m$ are fundamentally different. The parameter $\mu$ is UV parameter, while the parameter $m$ is IR one. As for the HSG formulas, they could be compared with virton and Klein–Gordon cases separately, but after the proper change of parameters to $\gamma$ and $v$ according to (12).

#### 7.5.1. Analysis of HSG Approximation

In the case of HSG approximation we have found the inflexion point. It was observed from the third-order in the coupling constant $g$ and the sixth-order polynomial approximation, which we encoded in the plot as $(3, 3)$ (three particles and sixth degree). It slightly changed by increasing the approximation order that was checked up to $(4, 3)$ (four particles and the sixth degree). Thus we expect that this result stays relevant for higher orders of approximation.

Both the virton and Klein–Gordon Gaussian theory Green functions satisfy the HSG approximation conditions. Though, the error of this approximation demands separate research, we can extract some predictions from this fact. For example, we can estimate that the next orders of polynomial approximation addition (without HSG) should result in arising of an inflexion point in virton and Klein–Gordon cases.

### 7.5.2. Analysis of PT First Orders

In the general propagator case there is no point of inflexion in orders $g$ and $g^2$ (165) and the function $w_{vac}$ is convex upwards (concave function). The existence of the inflexion point will be determined with the next PT orders.

We have also calculated the order $g^3$ for the fourth-degree approximation in the virton theory, but we will not present it here, because of this paper size. However, this contribution is positive for $g > 0$, so it produces an inflexion point referring to the virton propagator. Recalling the behaviour of polynomial approximations in the HSG approach, we expect that fourth and higher PT orders as well as higher degrees of approximations will not affect the existence of such an inflexion point. In the Klein–Gordon case, the coordinate integrals are much more complicated, so we postpone its research for future papers.

### 7.5.3. Analysis of Virton and Klein–Gordon Second-Degree Approximation Formulas

At the same time, we do not observe an inflexion point in the case of second-order polynomial approximation for the virton (183) and Klein–Gordon (187) vacuum energy densities. Numerically its absence follows from plots, and analytically from the asymptotics of the corresponding series. Namely, we have:

$$w_{vac}^{KG} = az - b(z\xi)^{3/2}, \quad w_{vac}^{v} = Az + B\ln^{3/2}z, \tag{190}$$

for some positive constants $A, B, a, b$. We do not present the derivation of these asymptotics in this paper since they are used only for reference.

From the written asymptotics it follows that both functions are convex upwards, which (combined with numerical plots) proves the absence of the inflexion points in the approximations of the second degree. Moreover, their convexity differs from the one predicted with the first three PT orders (at least, for the virton case) and the HSG consideration. Recall that we have deduced from these approaches that the correct behaviour at infinity is downward convex (convex function). Thus we come to the conclusion that these formulas are not applicable for a strong coupling limit.

Moreover, the vacuum energy density in the Klein–Gordon theory behaves nonphysically in the large coupling constants region because it becomes negative for sufficiently large $z$. It can be seen both from plots and asymptotics. Moreover, from the asymptotic, it follows that such negative values are reached for any $\xi$ if $z$ is sufficiently large.

Summarising the discussion of the second-degree polynomial approximation formulas, they fail to reach a strong coupling limit. Though, they can describe the intermediate values of the coupling constant (rather than very small only), and therefore they can be the computational alternative for PT first orders. The plots of vacuum energy density for second-degree polynomial approximation should be cut on some value of $g$.

## 8. Discussion

Let us start the discussion section from a brief overview of the work carried out in the paper. This paper starts from the theory with fractional power self-interaction $\int_{\mathbb{R}^d} dx\, g(x)|\phi(x)|^\alpha$ with $\alpha \in (1,2)$ for the nonlocal quadratic part. We rewrite the interaction potential through the successive application of straight and inverse Fourier transform (41). After that, in GF we expand the exponent with potential (50) and face the problem: we need to calculate complicated Gaussian expectation value of fractional potential Fourier transform. The solution to this problem arises from the Parseval–Plancherel identity (52), which brings us back to the original potential. Proof of the obtained series convergence for zero sources is presented in Section 4.3.

Our next goal is to introduce the connected Green functions GF (18) using the Meyer theorem (66). It generalizes the standard technique commonly used in statistical physics and results in the general expression for the connected Green functions GF in terms of coupling constant powers series (79).

The main part of our work focuses on the research of the "fractional Gaussian moments" (101) expansion in terms of integer moments. We perform careful approximation with the Legendre polynomials after restriction of the integration domain on a compact (hypersphere) $S^{n-1}$ in every $n$-particle contribution, resulting in (128), rewrite the obtained sum over graphs in a form convenient for direct calculations, and end with the expression for connected Green functions GF in terms of weighted sum of theories with integer powers contributions, obtaining this way an immediate generalisation of Feynman technique, but with converging series.

In the final part we focus on the exploration of the obtained PT applications for the cases of virton and Klein–Gordon propagators as well as for the HSG case. Thus, we provide comprehensive thorough research of various generating functionals. A brief list of the obtained results follows:

1. Construction of non-asymptotic PT series for generating functional $\mathcal{Z}$ in powers of coupling constant $g$ (53), proof of its convergence in Section 4.3 and derivation of similar formulas for GF $\mathcal{G} = \ln \mathcal{Z}$ (82). We assume that these PT series have a strong coupling limit;
2. Derivation of the general Formulas (53) and (85) for $\mathcal{Z}_n$ and $\mathcal{G}_n$ in fractional potential scalar field theory (1) with arbitrary nonlocal quadratic part of the action. These expansions are the generalization of common PT where we explicitly have a sum over Wick graphs (Feynman diagrams) with additional weights;
3. Construction of calculable and converging approximations for $\mathcal{Z}_n$ and $\mathcal{G}_n$ (111) with any precision, obtained with the polynomial approximation of function $f(t) = |t|^\alpha$ in $[-1, 1]$. It results in the fact that we can extend our attention to non-integer potential theories and rewrite it through a weighted sum of integer potential theories. As a consequence for the probability theory, we provide the series representation for multidimensional Gaussian distribution moments of fractional order, which has not been described in the literature yet, as far as we know;
4. Application of HSG approximation for $\mathcal{Z}_n[0]$ and $\mathcal{G}_n[0]$ (148), obtaining vacuum energy density (149) and clarifying its dependence on parameters of the theory (189);
5. Calculation and plotting of virton (183) and Klein–Gordon vacuum energy densities (187) for the second-degree Legendre polynomial approximation $w_{vac,N=1}$;
6. Finding the inflection point for the vacuum energy density in HSG approximation (Figure 10). We expect that this result remains if one adds the next orders of approximation and also in the exact result.

There were made multiple checks and comparisons of the obtained results.

## 9. Conclusions

Traveling back to the very beginning, to the motivation of the research, presented in this paper: the research of nonlocal field theories with interaction potential $g|\phi|^\alpha$ may shed light on the strong coupling behaviour of nonlocal $\phi^4$ theory providing such a definition that the latter is non-trivial for $d \geq 4$. Similarly for the nonlocal $\phi^6$ theory and $d \geq 3$. Moreover, our research produces new techniques for treating nonpolynomial potentials.

Additionally in the conclusion, we provide a brief list of possible further research topics that develop and generalize the material of our paper:

1. Research of the local limit of the nonlocal theories and their renormalizability. We expect that the new PT gives finite results for local QFTs with fractional potential after a suitable renormalization. That is sufficiently different from common PT, where this type lies out of applicability;
2. Research of the analytical continuation of the obtained formulas in parameter $\alpha$ to the domain $\alpha > 2$, specifically to $\alpha = 4$ and $\alpha = 6$;
3. We have carried out our research for the fractional potential, but all the general formulas can be easily generalized for other nonpolynomial potentials;
4. There are many different fields in QFT, e.g., complex-valued scalar field, fermionic field, tensor field. All these theories allow fractional generalization;

5.  The research of $\phi^4$ and $\phi^6$ theories directly in terms of integral over Gaussian measure in separable HS;

6.  Application of the Parseval–Plancherel identity for the integral over Gaussian measure in separable HS, which is very important for the strong coupling limit. It may find a lot of application in condensed matter and statistical physics;

7.  One of the central points of our paper is the polynomial approximation, where we can find error explicitly. Due to this, we have next two directions in this area:

    (a)  Research of the polynomial approximation next orders. It is a common direction for all perturbation theories;

    (b)  Careful treatment of error rate for different parameters in the theory, that can improve understanding of this approximation validity. An interesting question is "Where do we have to truncate the parameter $z$ to obtain a physically meaningful result?";

8.  In our paper we focused on vacuum energy density and did not pay enough attention to two-particle correlation function, four-particle, etc. The first one gives the Green function of theory with interaction and the second one describes the collision of particles. The research of $n$-particle Green functions is the natural next step in the study of fractional field theories. Unfortunately, this is difficult to do in one paper due to the large volume of material. It is also interesting to explore the composite operators Green functions, which were already mentioned above;

9.  Stochastic flow velocity field can be added in a multiplicative way to the theory of the scalar field with fractional interaction. In this case, we obtain a passive scalar model (a model of turbulent transport of impurity particles), which is of independent interest [44];

10. The obtained formulas are rather cumbersome, so their direct exploration is difficult enough. In such situations, asymptotic expressions are commonly used. Therefore, finding asymptotics for both HSG and high-order polynomial approximations is an independent interesting and useful problem;

11. We derived the expansion for fractional QFT in terms of integer QFTs. This situation is similar to conformal field theory, where a product of operators expands through infinite sum of other operators. However, there are such conformal theories in which this sum is finite. Therefore, we can assume that there are such nonlocal nonpolynomial theories in which we have a finite number of integer moments. Probably, such theories should contain not only bosonic but also fermionic fields;

12. It also seems natural to consider fractional field theories in a curved space-time. There are several problems that can arise here:

    (a)  In the virton propagator case, there are no solutions to the classical equations of motion for scalar field in flat space-time. However, virton quantum fluctuations in flat space-time are non-zero (which is clarified by the word "virton" itself). If in a curved space-time the correct definition of virton is still possible, then quantum fluctuations of virtons can affect the classical gravitational field. If the propagator ensures the existence of a non-zero classical solution, the interaction action value on classical fields (scalar and gravitational) is non-zero. This may be useful in astrophysics as scalar models of dark energy. However, establishing the connection between the parameters of the nonlocal field theory and those observed in astrophysics requires a separate study.

    (b)  Inspired by string theory, one can consider fractional field theory in high-dimensional spacetime, in particular $d = 26$. In this case, it is interesting to consider various compactifications of the theory, as well as the calculation of GFs after compactification. Perhaps such a problem will also be useful in astrophysical research.

    (c)  Perhaps the most general case here is the calculation of integrals over Gaussian measures specified in the space of both fields (scalar and gravitational). In physical terminology, the calculation of the path integrals over both fields. The

development of methods for calculating integrals over Gaussian measures in infinite-dimensional spaces is the necessary condition for the formulation of the quantum theory of gravitation.

Speaking about the most far-reaching research topics, one should mention the idea of the UV divergences elimination directly in path integral using some special (still unknown) "rule" and definitions of measure and integral in LCTVS. The family of seminorms defined by the Minkowski functionals and the scale hierarchy of QFT may turn out to be related to each other. However, this is quite a complicated topic and it takes a considerable amount of effort.

**Note added**. Instead of expanding path integrals in convergent PT series, one can research the issue of its approximation by absolutely convergent series finite number of terms. In other words, is it possible to approximate a path integral with any predetermined accuracy by a finite sum, depending on this accuracy? The answer to this question for scalar field theories with **polynomial** interactions turns out to be positive, and the proof is given in two papers [62,63] that deserve a special mention.

**Author Contributions:** Conceptualization, S.L.O.; Methodology, S.L.O.; Software, N.A.I. and D.V.S.; Validation, N.A.I. and D.V.S.; Formal analysis, N.A.I., S.L.O. and D.V.S.; Investigation, N.A.I., S.L.O. and D.V.S.; Writing—original draft, N.A.I., S.L.O. and D.V.S.; Writing—review and editing, N.A.I., S.L.O. and D.V.S.; Visualization, N.A.I. and D.V.S. All authors have read and agreed to the published version of the manuscript.

**Funding:** This research received no external funding.

**Data Availability Statement:** Data sharing is not applicable.

**Acknowledgments:** The authors are deeply grateful to their families for their love, wisdom and understanding. We also express special gratitude to Emil T. Akhmedov at the Department of Theoretical Physics MIPT as well as Oleg A. Zagriadskii and Stanislav S. Nikolaenko at the Department of Higher Mathematics MIPT for helpful discussions and comments. Finally, we are grateful to all the reviewers for their helpful comments, as well as to the staff of the Symmetry journal for their detailed proofreading of our paper.

**Conflicts of Interest:** The authors declare no conflict of interest.

## Abbreviations

The following abbreviations are used in this manuscript:

| | |
|---|---|
| BS | Banach Space |
| DCT | Dominated Convergence Theorem |
| FRG | Functional Renormalization Group |
| GF | Generating Functional |
| GCPF | Grand Canonical Partition Function |
| HS | Hilbert Space |
| HSG | Hard-Sphere Gas |
| IR Value | Infrared Value |
| LCTVS | Locally Convex Topological Vector Space |
| MCT | Monotone Convergence Theorem |
| PT | Perturbation Theory |
| QCD | Quantum Chromodynamics |
| QED | Quantum Electrodynamics |
| QFT | Quantum Field Theory |
| RG | Renormalization Group |
| UV Value | Ultraviolet Value |

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
