# Peer review of "Nonlocal Fractional Quantum Field Theory and Converging Perturbation Series"

_symmetry, doi:10.3390/sym15101823_

Round 1

Reviewer 1 Report

In this paper, the authors address the very important problem of fractional interaction in nonlocal field theories. They find a new perturbation theory for the interacting scalar field which is convergent. Various important results are obtained in the paper, and for the sake of clarity, they are mentioned both in the second section as well as in the discussion section. The paper is dense, as the authors present in minute details their arguments and carefully do their proofs. The proofs seem correct to me upon a first check, but it is really impossible to reproduce them in a short span of time.  

All in all, the paper represents a valuable contribution to the field of nonlocal quantum field theory. The results obtained here open up a large range of applications. It is very probable that the paper becomes a fundamental paper in the field.

I strongly recommend it for publication.

Minor misprints should be corrected, like "A section..." which should be "The section...", "Here ... " which should be "Here, ..." , etc.

Author Response

Dear Reviewer, we are of a great appreciation for your Report
on our manuscript. We have taken into account all your comments.
Please find our corrections in the attached cover letter pdf-file and
the new version of the paper.

Reviewer 2 Report

This paper examines a nonlocal quantum field theory specified
in Equation 1 involving a real scalar field in d-dimensions.
A convergent perturbation theory is developed, and the vacuum energy
density is evaluated to second order in the coupling. The paper
is very formal and mathematical, but Section 7 provides some useful
calculations with interesting plots and results.

This paper is very well written and is very detailed.  Motivations
for the work are described in detail in the introduction.  The work
is sound, the English is good, and references are plentiful and
appropriate.  On page 61, in point 8, the authors state "we focused
on vacuum energy density and did not pay enough attention to two-particle
correlation function, four-particle"...   This would be the major
deficiency of this paper, but presumably the authors will focus on
this in a future work, and given the length of this paper, this
is not a deal breaker here.

Publication is recommended after the following minor quibbles are
dealt with:

(1) In two instances, Klein-Gorgon is written instead of Klein-Gordon.

(2) At the bottom of page 1, it is stated "virton quark model and the
quark confinement model of hadrons are generally accepted".  The use
of "generally accepted" here is too strong.  It would be more accurate
to state "are qualitatively successful" or "have met with qualitative
success".

(3) There are some minor typos sprinkled throughout the text.  For
example, on page 61 on the second line from the top, there is "Besides,
out research produces..." so "out" should be "our".

The English is fine, but there are some minor mistakes as mentioned above.

Author Response

(The authors gave the same response as above.)

Reviewer 3 Report

Please see the PDF

A speed revision of the paper. 

Author Response

(The authors gave the same response as above.)
